# SPACETIME REPRESENTATION LEARNING

**Marc T. Law & James Lucas**
NVIDIA

## ABSTRACT

Much of the data we encounter in the real world can be represented as directed graphs. In this work, we introduce a general family of representations for directed graphs through connected time-oriented Lorentz manifolds, called "spacetimes" in general relativity. Spacetimes intrinsically contain a causal structure that indicates whether or not there exists a causal or even chronological order between points of the manifold, called events. This chronological order allows us to naturally represent directed edges via imposing the correct ordering when the nodes are embedded as events in the spacetime. Previous work in machine learning only considers embeddings lying on the simplest Lorentz manifold or does not exploit the connection between Lorentzian pre-length spaces and directed graphs. We introduce a well-defined approach to map data onto a general family of spacetimes. We empirically evaluate our framework in the tasks of hierarchy extraction of undirected graphs, directed link prediction and representation of directed graphs.

## 1 INTRODUCTION

Most of the machine learning literature has focused on learning representations that lie on a Riemannian manifold such as the Euclidean space, the $d$-sphere (e.g., $\ell_2$-normalized representations) (Wang et al., 2017; Tapaswi et al., 2019), hyperbolic geometry to represent graphs without cycles (Nickel & Kiela, 2017), or a statistical manifold in information geometry (Amari, 1998). Concepts of Euclidean geometry, such as distances, are naturally generalized to Riemannian geometry which remains easy to interpret. In contrast, recent approaches have considered learning representations that lie on a pseudo-Riemannian manifold to extract hierarchies in graphs with cycles (Law & Stam, 2020; Law, 2021) or represent directed graphs (Clough & Evans, 2017; Sim et al., 2021).

Pseudo-Riemannian manifolds are generalizations of Riemannian manifolds where the constraint of positive definiteness of the nondegenerate metric tensor is relaxed. The machine learning literature on pseudo-Riemannian manifolds can be divided into two categories. The first category focuses on how to optimize a given function whose domain is a pseudo-Riemannian manifold and does not take into account whether the manifold is time-oriented or not (Law & Stam, 2020; Law, 2021). The second category exploits the interpretation of a specific family of pseudo-Riemannian manifolds called "spacetimes" in general relativity (Clough & Evans, 2017; Sim et al., 2021). More specifically, spacetimes are connected time-oriented Lorentz manifolds. They intrinsically contain a *causal structure* that indicates whether or not there exists a causal order between points of the manifold, called events. This causal structure has been utilized to represent directed graphs where each node is an event and the existence of an arc (i.e., directed edge) between two nodes depends on the causal character of the curves joining them (Bombelli et al., 1987). In particular, Clough & Evans (2017) consider learning representations via the Minkowski spacetime which is the simplest such manifold. On the other hand, Sim et al. (2021) use three types of spacetimes and propose an *ad hoc* method based on the sign of some time coordinate difference function to determine the orientation of edges. The sign of such a function is not always meaningful as it for instance alternates periodically when the manifold is non-chronological and does not generalize to all spacetimes. Moreover, the distance function that they optimize is constant when two points cannot be joined by a geodesic.

**Contributions.** We propose a framework inspired by Lorentzian causality theory (Kronheimer & Penrose, 1967; Minguzzi, 2019), and in particular Lorentzian pre-length spaces (Kunzinger & Sämann, 2018), to learn directed graph representations lying on a large family of spacetimes. To this end, we present tools to account for time-orientation and exploit distances specific to Lorentz geometry. In particular, we propose to restrict the existence of edges to pairs of nodes whose representations lie in

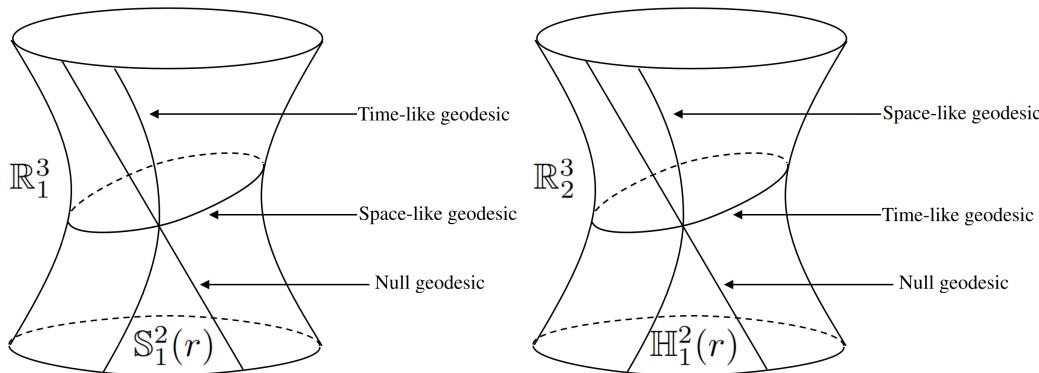

Figure 1: Geodesics of the de Sitter space $\mathbb{S}_1^d(r)$ (left) and of the anti-de Sitter space $\mathbb{H}_1^d(r)$ (right).

an open globally hyperbolic convex normal neighborhood. Such a neighborhood can be defined for any spacetime (see Theorem 2.7 of Minguzzi (2019)) and admits simple distance and time separation functions whose sign determines the direction of edges. We experimentally show that spacetimes can extract hierarchies in social networks better than standard approaches. Our framework also outperforms existing methods in link prediction on graphs with directed cycles.

## 2 SPACETIME DIFFERENTIAL GEOMETRY

We introduce some differential geometry background about spacetimes. Spacetimes have been widely studied, and we refer the reader to Hawking & Ellis (1973); Uhlenbeck (1975); O'Neill (1995); Beem et al. (1996); Wolf (2011); Gourgoulhon (2016), Chapter 5-8 & 14 of O'Neill (1983).

**Pseudo-Riemannian Manifold.** A $d$-dimensional pseudo-Riemannian manifold $(\mathcal{M}, g)$ is a smooth manifold such that every point $\mathbf{x} \in \mathcal{M}$ has a $d$-dimensional tangent space $T_\mathbf{x}\mathcal{M}$ whose metric tensor $g_\mathbf{x} : T_\mathbf{x}\mathcal{M} \times T_\mathbf{x}\mathcal{M} \to \mathbb{R}$ is a nondegenerate symmetric bilinear form (called a *scalar product*). Nondegeneracy means that $\forall \mathbf{v} \in T_\mathbf{x}\mathcal{M}, g_\mathbf{x}(\mathbf{u}, \mathbf{v}) = 0 \implies \mathbf{u} = \mathbf{0}$. When the context is clear and to simplify the notation, we write $\langle \cdot, \cdot \rangle := g_\mathbf{x}(\cdot, \cdot)$ to define the metric tensor at $\mathbf{x}$. We also write $\mathcal{M}$ instead of $(\mathcal{M}, g)$. We write points $\mathbf{x} \in \mathcal{M}$ of the manifold in bold serif font, and tangent vectors $\mathbf{u} \in T_\mathbf{x}\mathcal{M}$ in bold sans-serif font when we want to distinguish them from points.

**Lorentz manifold.** Every tangent space $T_\mathbf{x}\mathcal{M}$ of a $d$-dimensional pseudo-Riemannian manifold $\mathcal{M}$ admits an orthonormal basis $\{\mathbf{e}_1, \ldots, \mathbf{e}_d\}$ that satisfies $\forall i, \langle \mathbf{e}_i, \mathbf{e}_i \rangle = \pm 1$ and $\forall\, i \neq j, \langle \mathbf{e}_i, \mathbf{e}_j \rangle = 0$. The index $\nu \leq d$ of $\mathcal{M}$ is the number of vectors $\mathbf{e}_i$ that satisfy $\langle \mathbf{e}_i, \mathbf{e}_i \rangle = -1$. If $\nu = 0$, $\mathcal{M}$ is Riemannian and its metric tensor is positive definite (i.e., $\forall \mathbf{x} \in \mathcal{M}, \forall \mathbf{u} \in T_\mathbf{x}\mathcal{M}, \langle \mathbf{u}, \mathbf{u} \rangle \geq 0$ and $\langle \mathbf{u}, \mathbf{u} \rangle = 0 \iff \mathbf{u} = \mathbf{0}$). If $\nu = 1$, $\mathcal{M}$ is a Lorentz manifold and $T_\mathbf{x}\mathcal{M}$ is a Lorentz vector space.

**Future timecone.** A nonzero tangent vector $\mathbf{u}$ is called timelike (or *chronological*), null, spacelike or non-spacelike (or *causal*) if $\langle \mathbf{u}, \mathbf{u} \rangle$ is negative, zero, positive or nonpositive, respectively. The type into which $\mathbf{u}$ falls is called its *causal character*. If $\mathbf{u} = \mathbf{0}$, then $\mathbf{u}$ is spacelike. Every Lorentz tangent space contains two timecones. Some timelike tangent vector $\mathbf{t} \in T_\mathbf{x}\mathcal{M}$ can arbitrarily be used to define the future timecone as the following set: $\mathcal{C}_\mathbf{x}^+(\mathbf{t}) := \{\mathbf{v} \in T_\mathbf{x}\mathcal{M} : \langle \mathbf{v}, \mathbf{v} \rangle < 0, \langle \mathbf{t}, \mathbf{v} \rangle < 0\}$ whereas $-\mathbf{t}$ defines the past timecone $\mathcal{C}_\mathbf{x}^-(\mathbf{t}) := \mathcal{C}_\mathbf{x}^+(-\mathbf{t})$. Two timelike tangent vectors $\mathbf{u}$ and $\mathbf{v}$ are in the same timecone iff $\langle \mathbf{u}, \mathbf{v} \rangle < 0$. They belong to different timecones if $\mathbf{v} = -\mathbf{u}$.

**Time-orientability and time-orientation.** A continuous vector field $X$ is a function that assigns to each point $\mathbf{x} \in \mathcal{M}$ a tangent vector of $\mathcal{M}$ at $\mathbf{x}$ denoted by $X(\mathbf{x}) \in T_\mathbf{x}\mathcal{M}$. $X$ and $-X$ are *timelike* if $\forall \mathbf{x} \in \mathcal{M}, \langle X(\mathbf{x}), X(\mathbf{x}) \rangle < 0$. A Lorentz manifold is time-orientable iff there exists a timelike vector field. If $\mathcal{M}$ is assigned such a timelike vector field $X$, it is time-oriented by $X$. In this case, non-spacelike tangent vectors $\mathbf{u}$ at each point $\mathbf{x}$ can be divided into two separate classes: *future-directed* if $\langle X(\mathbf{x}), \mathbf{u} \rangle < 0$, and *past-directed* if $\langle X(\mathbf{x}), \mathbf{u} \rangle > 0$.

**A curve** $\gamma_{\mathbf{x} \to \mathbf{u}} : I \to \mathcal{M}$ where $I \subseteq \mathbb{R}$ is defined such that its initial point is $\gamma_{\mathbf{x} \to \mathbf{u}}(0) = \mathbf{x}$ and its initial velocity is $\gamma'_{\mathbf{x} \to \mathbf{u}}(0) = \mathbf{u} \in T_\mathbf{x}\mathcal{M}$. We denote it by $\gamma$ when its initial conditions are clear from

the context. If its acceleration is zero, then it is called a geodesic. The curve $\gamma$ is called timelike, null or spacelike if its velocity on the whole domain $I$ is timelike, null or spacelike, respectively. It is called future-directed (or future-pointing) if its velocity is future-directed (and causal) on $I$.

**Completeness.** The manifolds that we consider in this paper are geodesically complete (i.e., $I = \mathbb{R}$), and the exponential map $\exp_{\mathbf{x}} : T_{\mathbf{x}}\mathcal{M} \to \mathcal{M}$ of $\mathcal{M}$ at $\mathbf{x}$ is defined as $\exp_{\mathbf{x}}(\mathbf{u}) := \gamma_{\mathbf{x}\to\mathbf{u}}(1)$ where $\gamma_{\mathbf{x}\to\mathbf{u}}$ is a geodesic. The maximal normal neighborhood of $\mathbf{x}$ is the maximal subset $\mathcal{U}_{\mathbf{x}} \subseteq \mathcal{M}$ where the logarithmic map $\log_{\mathbf{x}} := \exp_{\mathbf{x}}^{-1} : \mathcal{U}_{\mathbf{x}} \to T_{\mathbf{x}}\mathcal{M}$ is a diffeomorphism, it satisfies $\mathcal{U}_{\mathbf{x}} := \{\mathbf{y} \in \mathcal{M} : \exp_{\mathbf{x}}(\exp_{\mathbf{x}}^{-1}(\mathbf{y})) = \mathbf{y}\}$. To simplify the notation, we also write $\overrightarrow{\mathbf{xy}} := \log_{\mathbf{x}}(\mathbf{y})$ where $\mathbf{y} \in \mathcal{U}_{\mathbf{x}}$. $\mathcal{U}_{\mathbf{x}}$ is *convex* if $\forall \mathbf{y} \in \mathcal{U}_{\mathbf{x}}$, there exists a unique geodesic totally contained in $\mathcal{U}_{\mathbf{x}}$ from $\mathbf{x}$ to $\mathbf{y}$.

We now present some pseudo-Riemannian manifolds that will be relevant. Their differential geometry tools (e.g., exponential/logarithmic map and parallel transport) are provided in Appendix B.

• **Pseudo-Euclidean space $\mathbb{R}_{\nu}^d$.** The flat $d$-dimensional pseudo-Riemannian manifold of index $\nu$ is denoted by $\mathbb{R}_{\nu}^d$. In particular, $\mathbb{R}_0^d$ is the Euclidean space and $\mathbb{R}_1^d$ is the Minkowski space (or Minkowski spacetime). Since it is a vector space, we can identify its tangent space to the space itself by means of the natural isomorphism $\mathbb{R}_{\nu}^d \approx T_{\mathbf{x}}\mathbb{R}_{\nu}^d$. $\mathbb{R}_{\nu}^d$ is equipped with the following scalar product:

$$\forall \mathbf{x} = (x_{1-\nu}, \ldots, x_{d-\nu})^\top, \ \mathbf{y} = (y_{1-\nu}, \ldots, y_{d-\nu})^\top, \ \langle \mathbf{x}, \mathbf{y} \rangle_{\nu} := -\sum_{i=1-\nu}^{0} x_i y_i + \sum_{j=1}^{d-\nu} x_j y_j. \quad (1)$$

The maximal normal neighborhood for all $\mathbf{x} \in \mathbb{R}_{\nu}^d$ is $\mathcal{U}_{\mathbf{x}} = \mathbb{R}_{\nu}^d$. In special relativity, the first $\nu$ elements of $\mathbf{x} \in \mathbb{R}_{\nu}^d$ are called time coordinates and the other ones are called space coordinates.

• **The pseudo-sphere $\mathbb{S}_{\nu}^d(r)$** of radius $r > 0$ is called the de Sitter space when $\nu = 1$ and defined as:

$$\mathbb{S}_{\nu}^d(r) := \{\mathbf{x} \in \mathbb{R}_{\nu}^{d+1} : \langle \mathbf{x}, \mathbf{x} \rangle_{\nu} = r^2\}. \quad (2)$$

It is not time-orientable if $d - \nu$ is odd, and $\mathcal{U}_{\mathbf{x}} = \{\mathbf{y} \in \mathbb{S}_{\nu}^d(r) : \langle \mathbf{x}, \mathbf{y} \rangle_{\nu} > -r^2\}$.

• **The pseudo-hyperbolic space $\mathbb{H}_{\nu}^d(r)$** is called the anti-de Sitter space when $\nu = 1$ and defined as:

$$\mathbb{H}_{\nu}^d(r) := \{\mathbf{x} \in \mathbb{R}_{\nu+1}^{d+1} : \langle \mathbf{x}, \mathbf{x} \rangle_{\nu+1} = -r^2\}. \quad (3)$$

The anti-de Sitter space is time-orientable for all $d$, and $\mathcal{U}_{\mathbf{x}} = \{\mathbf{y} \in \mathbb{H}_{\nu}^d(r) : \langle \mathbf{x}, \mathbf{y} \rangle_{\nu+1} < r^2\}$.

• **The cylindrical Minkowski space $\mathbb{L}_1^d(C) := \mathbb{R}_1^d/\sim$** is a quotient set defined such that $\mathbf{x} \in \mathbb{R}_1^d$ and $\mathbf{y} \in \mathbb{R}_1^d$ are equivalent (i.e., $\mathbf{x} \sim \mathbf{y}$) iff $\forall i > 0, y_i = x_i$ and $\exists k \in \mathbb{Z}, y_0 = x_0 + kC$ where $C > 0$ is a circumference hyperparameter. See page 148 of O'Neill (1983) for other types of Lorentz cylinders. We have $\mathcal{U}_{\mathbf{x}} = \{\mathbf{y} = (y_0, \ldots, y_{d-1})^\top \in \mathbb{R}_1^d : y_0 \in (x_0 - C/2, x_0 + C/2)\}$.

## 3  SPACETIME GRAPH REPRESENTATION

A **spacetime** is a connected time-oriented Lorentz manifold $\mathcal{M}$ whose points are called events. Informally, *time-oriented* is often weakened to *time-orientable*, and Hawking & Ellis (1973) ignore the time-orientability criterion to define spacetimes although it is required to study their *causal structure*. Our main contribution is to exploit this causal structure via Lorentzian pre-length spaces to represent graphs. In the following, we always assume that $\mathcal{M}$ is a spacetime unless stated otherwise. A finite directed graph can be given the structure of a Lorentzian pre-length space (Kunzinger & Sämann, 2018). To define our graphs, we then choose a special type of Lorentzian pre-length space that is easy to optimize. We now provide and follow the definitions of Kunzinger & Sämann (2018).

**Causal space.** Let $\mathcal{X}$ be a set with a reflexive and transitive relation $\leq$, and $\ll$ a transitive relation contained in $\leq$ (i.e., $\ll \subseteq \leq$, so $\mathbf{x} \ll \mathbf{y} \implies \mathbf{x} \leq \mathbf{y}$). Then $(\mathcal{X}, \ll, \leq)$ is called a causal space. This definition is more general than the one given in the seminal work of Kronheimer & Penrose (1967).

Following general relativity, the event $\mathbf{x} \in \mathcal{M}$ causally (resp. chronologically) precedes the event $\mathbf{y} \in \mathcal{M}$, and we write $\mathbf{x} < \mathbf{y}$ (resp. $\mathbf{x} \ll \mathbf{y}$) iff there exists a future-directed causal (resp. timelike) curve from $\mathbf{x}$ to $\mathbf{y}$. This condition might be difficult to verify in general. However, if $\mathbf{y}$ is in a convex normal neighborhood of $\mathbf{x}$ denoted by $\mathcal{V}_{\mathbf{x}} \subseteq \mathcal{U}_{\mathbf{x}}$ (and we always assume in the following that $\mathbf{x} \in \mathcal{V}_{\mathbf{x}}$), we have $\mathbf{x} < \mathbf{y}$ (resp. $\mathbf{x} \ll \mathbf{y}$) iff there exists a nonconstant future-directed causal (resp. timelike) geodesic from $\mathbf{x}$ to $\mathbf{y}$ (see Proposition 4.5.1 of (Hawking & Ellis, 1973)). We note

$\mathbf{x} \leq \mathbf{y} \iff \mathbf{x} = \mathbf{y}$ or $\mathbf{x} < \mathbf{y}$. Any open subset $\mathcal{W} \subseteq \mathcal{M}$ of a spacetime is a spacetime in its own right, and the *intrinsic* causal relations of $\mathcal{W}$ imply the corresponding ones in $\mathcal{M}$. In the case where $\mathcal{W} \subseteq \mathcal{M}$ is a convex open set, the intrinsic causality of $\mathcal{W}$ is as simple as that of Minkowski space (see page 403 of O'Neill (1983)). Moreover, both $(\mathcal{M}, \ll, \leq)$ and $(\mathcal{W}, \ll, \leq)$ are causal spaces.

**Lorentzian pre-length space.** Let $(\mathcal{X}, \ll, \leq)$ be a causal space and d a metric on $\mathcal{X}$. Let $\tau : \mathcal{X} \times \mathcal{X} \to [0, \infty]$ be a lower semicontinuous map (*w.r.t.* the metric topology induced by d) that satisfies the reverse triangle inequality: $\forall \mathbf{x}, \mathbf{y}, \mathbf{z} \in \mathcal{X}$ with $\mathbf{x} \leq \mathbf{y} \leq \mathbf{z}, \tau(\mathbf{x}, \mathbf{z}) \geq \tau(\mathbf{x}, \mathbf{y}) + \tau(\mathbf{y}, \mathbf{z})$. Suppose that $\tau(\mathbf{x}, \mathbf{y}) = 0$ if $\mathbf{x} \not\leq \mathbf{y}$ and $\tau(\mathbf{x}, \mathbf{y}) > 0$ iff $\mathbf{x} \ll \mathbf{y}$. Then $(\mathcal{X}, d, \leq, \ll, \tau)$ is a Lorentzian pre-length space and $\tau$ is called the *time separation function*. See details about $\tau$ in Section 3.3.

### 3.1 GRAPH CONSTRUCTION VIA LORENTZIAN PRE-LENGTH SPACES

We now explain our methodological contribution, which is the construction of directed graphs from the properties mentioned above. Let us define a directed graph $G = (V, E)$ where $V = \{v_i\}_{i=1}^{n}$ is the node set and $E$ is the set of arcs. Our goal is to represent each node $v_i$ by a point $\mathbf{x}_i \in \mathcal{M}$ so that there exists an arc from $v_i$ to $v_j$ (i.e., $(v_i, v_j) \in E$) only if $\mathbf{x}_i \ll \mathbf{x}_j$. We propose to consider subgraphs $G_i = (V, E_i)$ defined such that $E = \bigcup_{i=1}^{n} E_i$. Each subgraph $G_i$ is given by the structure of the Lorentzian pre-length space $(\mathcal{V}_{\mathbf{x}_i}, d, \leq, \ll, \tau)$ where $\mathcal{V}_{\mathbf{x}_i} \subseteq \mathcal{U}_{\mathbf{x}_i}$ is an open subset of the maximal normal neighborhood $\mathcal{U}_{\mathbf{x}_i}$. More precisely, we consider the *chronological future* $\mathcal{I}^{+}(\mathbf{x}_i, \mathcal{V}_{\mathbf{x}_i}) := \{\mathbf{y} \in \mathcal{V}_{\mathbf{x}_i} : \mathbf{x}_i \ll \mathbf{y}\}$ of the point $\mathbf{x}_i$ *relative to* some given set $\mathcal{V}_{\mathbf{x}_i} \subseteq \mathcal{U}_{\mathbf{x}_i}$ (see page 402 of O'Neill (1983)), and we draw an arc from $v_i$ to $v_j$ iff $\mathbf{x}_j \in \mathcal{I}^{+}(\mathbf{x}_i, \mathcal{V}_{\mathbf{x}_i})$. Assuming $\mathcal{V}_{\mathbf{x}_i}$ is a convex normal neighborhood and $\mathbf{x}_j \in \mathcal{V}_{\mathbf{x}_i}$, we have $\mathbf{x}_j \in \mathcal{I}^{+}(\mathbf{x}_i, \mathcal{V}_{\mathbf{x}_i})$ only if $\overrightarrow{\mathbf{x}_i \mathbf{x}_j}$ is defined and future-directed timelike (see Proposition 4.12 of Beem et al. (1996)).

Our framework produces a subgraph $G = \bigcup_{i=1}^{n} G_i$ of the graph $\mathscr{G}$ described by the chronological relations of $\mathcal{M}$, and the causal relations between events depend on the choice of $\mathcal{M}$. It is worth noting that for any spacetime, chronological order is transitive (i.e., $\mathbf{x}_i \ll \mathbf{x}_j$ and $\mathbf{x}_j \ll \mathbf{x}_k \implies \mathbf{x}_i \ll \mathbf{x}_k$ (i.e., $\mathbf{x}_k \in \mathcal{I}^{+}(\mathbf{x}_i, \mathcal{M})$), see Chapter 3.2 of Beem et al. (1996)). $v_i$ is then connected by an arc to all the successors of $v_j$ in $\mathscr{G}$. We avoid this degenerate case by drawing an arc from $v_i$ to $v_k$ in $G$ iff $\mathbf{x}_k \in \mathcal{I}^{+}(\mathbf{x}_i, \mathcal{V}_{\mathbf{x}_i})$. The choice of $\mathcal{M}$ and $\mathcal{V}_{\mathbf{x}_i} \subseteq \mathcal{M}$ for all $i$ is then crucial to construct $G \subseteq \mathscr{G}$.

**Existence of directed cycles.** Our framework can represent graphs with directed cycles only if $\mathcal{M}$ is non-chronological (i.e., there exists at least one *closed timelike curve*: $\exists \mathbf{x} \in \mathcal{M}, \mathbf{x} \ll \mathbf{x}$, see Fig. 1), which is the case if $\mathcal{M}$ is the anti-de Sitter space $\mathbb{H}_1^{d}(r)$ or the Cylindrical Minkowski space $\mathbb{L}_1^{d}(C)$. Spacetimes that do not contain closed timelike curves are called *chronological* (i.e., $\nexists \mathbf{x} \in \mathcal{M}, \mathbf{x} \ll \mathbf{x}$) and can represent only Directed Acyclic Graphs (DAGs) in our framework. Some chronological spacetimes such as $\mathbb{S}_1^{d}(r)$ (when $d \geq 3$ is odd) or $\mathbb{R}_1^{d}$ are called *globally hyperbolic* and satisfy: $\mathbf{x} \leq \mathbf{y} \implies \mathbf{x}$ and $\mathbf{y}$ can be joined by a (longest) causal geodesic that is not necessarily unique (see page 66 of Beem et al. (1996)). Nonetheless, if $\mathcal{M}$ is $\mathbb{S}_1^{d}(r)$ or $\mathbb{R}_1^{d}$, we also have $\mathbf{y} \in \mathcal{I}^{+}(\mathbf{x}, \mathcal{M})$ iff $\mathbf{y} \in \mathcal{U}_{\mathbf{x}}$ and $\overrightarrow{\mathbf{x}\mathbf{y}}$ is future-directed timelike (see page 411 of (O'Neill, 1983) for details on conditions). Although DAGs do not contain directed cycles, they can contain undirected cycles.

For simplicity, we consider that $\mathcal{V}_{\mathbf{x}} \subseteq \mathcal{U}_{\mathbf{x}}$ is the convex normal neighborhood of $\mathbf{x}$ that contains points $\mathbf{y}$ such that the arc length $d_{\gamma}$ of the geodesic $\gamma$ from $\mathbf{x} = \gamma(0)$ to $\mathbf{y} = \gamma(1)$ is smaller than some arbitrary threshold $\varepsilon \in (0, \infty]$ and can be formulated as $d_{\gamma}(\mathbf{x}, \mathbf{y}) := \sqrt{|\langle \overrightarrow{\mathbf{x}\mathbf{y}}, \overrightarrow{\mathbf{x}\mathbf{y}} \rangle|}$. We have:

$$\mathcal{I}^{+}(\mathbf{x}, \mathcal{V}_{\mathbf{x}}) = \{\mathbf{y} \in \mathcal{U}_{\mathbf{x}} : -\varepsilon^2 < \langle \overrightarrow{\mathbf{x}\mathbf{y}}, \overrightarrow{\mathbf{x}\mathbf{y}} \rangle < 0, \overrightarrow{\mathbf{x}\mathbf{y}} \in \mathcal{C}_{\mathbf{x}}^{+}(\mathbf{t})\}, \tag{4}$$

where $\mathcal{C}_{\mathbf{x}}^{+}(\mathbf{t})$ is the future timecone parametrized by some arbitrary timelike tangent vector $\mathbf{t} \in T_{\mathbf{x}}\mathcal{M}$. The motivation is to choose or learn $\varepsilon$ small enough that $v_i$ is not connected to undesired successors of $v_j$. However, in most of our experiments we simply consider that $\mathcal{V}_{\mathbf{x}} = \mathcal{U}_{\mathbf{x}}$ (see Section C.4).

### 3.2 EXAMPLES OF SPACETIMES

We first illustrate how to represent directed graphs with the simplest spacetime, which is the (flat) Minkowski space $\mathbb{R}_1^{d}$. It is used in special relativity which is a special case of the general relativity of a spacetime isometric to $\mathbb{R}_1^{4}$. It is also the geometry induced on each fixed tangent space of an arbitrary Lorentz manifold. It was used in Clough & Evans (2017) to represent DAGs due to its global hyperbolicity and the fact that any pair of points of $\mathbb{R}_1^{d}$ can be joined by a geodesic. It is worth noting that the Hopf-Rinow theorem does not hold for non-Riemannian manifolds such as spacetimes.

Therefore for many spacetimes, there exist pairs of points that cannot be joined by a geodesic even if the spacetime is geodesically complete. Working with convex normal neighborhoods $\mathcal{V}_{\mathbf{x}_i}$ allows us to constrain chronological order between points only via timelike geodesics as explained in Section 3.1.

• **Minkowski spacetime** $\mathbb{R}_1^d$. We recall that $\forall \nu, \mathbb{R}_\nu^d \approx T_{\mathbf{x}}\mathbb{R}_\nu^d$. Its geodesic $\gamma_{\mathbf{x} \to \mathbf{y}} : \mathbb{R} \to \mathbb{R}_\nu^d$ is $\gamma_{\mathbf{x} \to \mathbf{y}}(t) := \mathbf{x} + t\mathbf{y}$. The exponential map at $\mathbf{x}$ is $\exp_{\mathbf{x}}(\mathbf{y}) := \mathbf{x} + \mathbf{y}$ and its inverse is $\overrightarrow{\mathbf{xy}} := \mathbf{y} - \mathbf{x}$. $\mathbb{R}_1^d$ is time-oriented by the vector field $\partial/\partial x_0$ (e.g., $\forall \mathbf{x}, \mathbf{y}, \ \tau(\mathbf{x}, \mathbf{y}) := y_0 - x_0$, and we should in theory define $\tau(\mathbf{x}, \mathbf{y}) := 0$ if $\mathbf{x} \not\leq \mathbf{y}$ to properly follow the definition of Lorentzian pre-length spaces but we ignore this criterion during training for optimization purpose). Let us define $\mathbf{t} := (1, 0, \ldots, 0)^\top$, $\alpha = -(y_0 - x_0) + \sqrt{\sum_{i=1}^{d-1}(y_i - x_i)^2}$ and $\beta := \langle \overrightarrow{\mathbf{xy}}, \overrightarrow{\mathbf{xy}} \rangle = -(y_0 - x_0)^2 + \sum_{i=1}^{d-1}(y_i - x_i)^2$.

According to equation 4, we have $\mathbf{y} \in \mathcal{I}^+(\mathbf{x}, \mathcal{V}_{\mathbf{x}})$ iff $\overrightarrow{\mathbf{xy}}$ is future-directed timelike (i.e., $\beta < 0$ and $\langle \overrightarrow{\mathbf{xy}}, \mathbf{t} \rangle < 0$, or equivalently $\alpha < 0$) and $\mathrm{d}_\gamma(\mathbf{x}, \mathbf{y})$ is smaller than $\varepsilon$ (i.e., $-\varepsilon^2 < \beta$).

There might exist a path but no arc between $\mathbf{x} \ll \mathbf{y}$ (i.e., $\mathbf{y} \in \mathcal{I}^+(\mathbf{x}, \mathcal{M}) \setminus \mathcal{I}^+(\mathbf{x}, \mathcal{V}_{\mathbf{x}})$) iff $\langle \overrightarrow{\mathbf{xy}}, \mathbf{t} \rangle < 0$ and $\beta \leq -\varepsilon^2$. There exists no path between $\mathbf{x}$ and $\mathbf{y}$ (i.e., $\mathbf{x} \not\ll \mathbf{y}$ and $\mathbf{y} \not\ll \mathbf{x}$) iff $\beta \geq 0$.

• **de Sitter spacetime** $\mathbb{S}_1^d(r)$. The original de Sitter spacetime $\mathbb{S}_1^4(r)$ is not time-orientable (see Corollary 11.2.6 of Wolf (2011)). However, when $d \geq 3$, $\nu > 0$ and $d - \nu$ is even, $\mathbb{S}_\nu^d(r)$ is orientable and time-orientable, the projective space $\mathbb{S}_\nu^d(r)/ \pm 1$ used in Law (2021) is orientable but not time-orientable (see page 247 of O'Neill (1983)). We assume in the following that conditions hold so that $\mathbb{S}_\nu^d(r)$ is a spacetime and we refer the reader to the appendix for details.

For any pair of points $\mathbf{x} \in \mathbb{S}_\nu^d(r)$ and $\mathbf{y} \in \mathbb{S}_\nu^d(r)$, $\overrightarrow{\mathbf{xy}}$ is defined iff $\mathbf{y} \in \mathcal{U}_{\mathbf{x}}$ (i.e., $\langle \mathbf{x}, \mathbf{y} \rangle_1 > -r^2$), and $\overrightarrow{\mathbf{xy}}$ is timelike iff $\langle \mathbf{x}, \mathbf{y} \rangle_1 > r^2$. Let $\mathbf{p} := (0, \ldots, 0, r)^\top \in \mathbb{S}_1^d(r)$ denote the positive pole, and $\Gamma_{\mathbf{p}}^{\mathbf{x}} : T_{\mathbf{p}}\mathbb{S}_1^d(r) \to T_{\mathbf{x}}\mathbb{S}_1^d(r)$ denote the parallel transport from $T_{\mathbf{p}}\mathbb{S}_1^d(r)$ to $T_{\mathbf{x}}\mathbb{S}_1^d(r)$. Since the parallel transport preserves the *causal character* of any tangent vector $\mathbf{v}$, we can define the future timecone $\mathcal{C}_{\mathbf{x}}^+(\overrightarrow{\mathbf{xy}})$ with respect to the timelike tangent vector $\mathbf{t} := (1, 0, \ldots, 0)^\top \in T_{\pm \mathbf{p}}\mathbb{S}_1^d(r)$ as follows:

**Lemma 3.1.** *Assuming $\overrightarrow{\mathbf{xy}}$ is timelike, $\overrightarrow{\mathbf{xy}}$ is future-directed iff $\Gamma_{\mathbf{p}}^{\mathbf{x}}(\mathbf{t}) \in \mathcal{C}_{\mathbf{x}}^+(\overrightarrow{\mathbf{xy}})$ if $\Gamma_{\mathbf{p}}^{\mathbf{x}}$ is defined (i.e., $\mathbf{p} \in \mathcal{U}_{\mathbf{x}}$), and $\Gamma_{-\mathbf{p}}^{\mathbf{x}}(\mathbf{t}) \in \mathcal{C}_{\mathbf{x}}^+(\overrightarrow{\mathbf{xy}})$ otherwise (i.e., $-\mathbf{p} \in \mathcal{U}_{\mathbf{x}}$). See proof in Appendix C.1.2.*

• **The anti-de Sitter spacetime** $\mathbb{H}_1^d(r)$ and its projective version $\mathbb{P}_1^d(r) := \mathbb{H}_1^d(r)/ \pm 1$ are non-chronological and satisfy $\mathbf{x} \ll \mathbf{y} \implies \mathbf{y} \ll \mathbf{x}$, which is convenient to represent graphs with directed cycles. Sim et al. (2021) use $\mathbb{H}_1^d(r)$ to represent directed graphs but also promote arcs (i.e., causal relation) between pairs of nodes that are not connected by any geodesic, which makes their problem hard to optimize. In this work, we only consider the existence of arcs if there exists a timelike geodesic in the convex normal neighborhood $\mathcal{V}_{\mathbf{x}}$ joining two events. See Appendix C for details.

• **A warped product** is a manifold $(\mathcal{M}_1 \times \mathcal{M}_2, g_1 \oplus f g_2)$ denoted by $\mathcal{M}_1 \times_f \mathcal{M}_2$ where $(\mathcal{M}_1, g_1)$, $(\mathcal{M}_2, g_2)$ and $f : \mathcal{M}_1 \to (0, \infty)$ is a smooth function called the *warping function* (e.g., $f = 1$). Let $\mathcal{M} = \mathcal{B} \times_f \mathcal{F}$ be a Lorentz warped product where $\mathcal{B}$ is Lorentz and $\mathcal{F}$ is a complete Riemannian manifold. $\mathcal{M}$ is time-orientable iff $\mathcal{B}$ is (see page 417 of (O'Neill, 1983)). $\mathcal{M}$ satisfies the chronology, causality or strong causality iff $\mathcal{B}$ does. Our framework can then be extended to warped products.

### 3.3 LORENTZIAN DISTANCE AND LORENTZIAN LENGTH SPACES

The Lorentzian distance indicates chronological order (hence causality) between events when it is positive, satisfies the reverse triangle inequality and its squared function is of class $C^2$ on a normal neighborhood. These properties make it ideal for optimization, and it can be used as time separation function. We refer the reader to Chapter 4 of Beem et al. (1996) and Section 5 of Minguzzi (2019).

**Proposition 4.5.3 of Hawking & Ellis (1973):** Let $\mathbf{x}$ and $\mathbf{y}$ lie in a convex normal neighborhood $\mathcal{V}_{\mathbf{x}} \subseteq \mathcal{U}_{\mathbf{x}}$. If $\mathbf{x}$ and $\mathbf{y}$ can be joined by a causal curve in $\mathcal{V}_{\mathbf{x}}$, the longest such curve is the unique causal geodesic in $\mathcal{V}_{\mathbf{x}}$ from $\mathbf{x}$ to $\mathbf{y}$. This is in contrast with Riemannian geometry where the (spacelike) geodesic corresponds to the shortest curve joining points. Since $\mathcal{V}_{\mathbf{x}}$ is a convex normal neighborhood, we can define the arc length of such a curve by $\chi_\mathcal{V}(\mathbf{x}, \mathbf{y}) := \sqrt{-\langle \overrightarrow{\mathbf{xy}}, \overrightarrow{\mathbf{xy}} \rangle} \geq 0$. If $\mathbf{x} \ll \mathbf{y}$, $\chi_\mathcal{V}(\mathbf{x}, \mathbf{y})$ is called the Lorentzian distance from $\mathbf{x}$ to $\mathbf{y}$ on $\mathcal{V}_{\mathbf{x}}$, it corresponds to the elapsed *proper time* between the events $\mathbf{x}$ and $\mathbf{y}$ (i.e., as measured by a clock along the geodesic $\gamma_{\mathbf{x} \to \overrightarrow{\mathbf{xy}}}$) (Gourgoulhon, 2016). If we define $\tau := \chi_\mathcal{V}$, then $(\mathcal{V}_{\mathbf{x}}, \mathrm{d}, \leq, \ll, \tau)$ is called a *Lorentzian length space*. In this paper, we consider the squared Lorentzian distance which is defined as $\mathbf{y} \in \mathcal{I}^+(\mathbf{x}, \mathcal{V}_{\mathbf{x}}) \implies \chi_\mathcal{V}^2(\mathbf{x}, \mathbf{y}) = -\langle \overrightarrow{\mathbf{xy}}, \overrightarrow{\mathbf{xy}} \rangle$.

In the literature (Hawking & Ellis, 1973; Beem et al., 1996), the (squared) Lorentzian distance is defined such that $\chi^2_{\mathcal{V}}(\mathbf{x}, \mathbf{y}) = 0$ if $\mathbf{x}$ and $\mathbf{y}$ are not joined by a causal curve due to lack of causality. When evaluating learned representations (i.e., at test time), we therefore consider that $\chi^2_{\mathcal{V}}(\mathbf{x}, \mathbf{y}) = 0$ if this is the case. However, during training and for optimization purpose, we propose to consider:

$$\text{If } \mathbf{x} \not\ll \mathbf{y} \text{ and } \mathbf{y} \not\ll \mathbf{x}, \; \chi^2_{\mathcal{V}}(\mathbf{x}, \mathbf{y}) = \begin{cases} -\langle \overrightarrow{\mathbf{xy}}, \overrightarrow{\mathbf{xy}} \rangle & \text{if } \mathcal{M} = \mathbb{R}^d_\nu \\ 2(\langle \mathbf{x}, \mathbf{y} \rangle_\nu - r^2) & \text{if } \mathcal{M} = \mathbb{S}^d_\nu(r) \\ -2(|\langle \mathbf{x}, \mathbf{y} \rangle_{\nu+1}| - r^2) & \text{if } \mathcal{M} = \mathbb{P}^d_\nu(r). \end{cases} \tag{5}$$

This makes the function $\chi^2_{\mathcal{V}}$ differentiable everywhere (except when $\langle \mathbf{x}, \mathbf{y} \rangle_{\nu+1} = 0$ if $\mathcal{M} = \mathbb{P}^d_\nu(r)$), equal to $-\langle \overrightarrow{\mathbf{xy}}, \overrightarrow{\mathbf{xy}} \rangle > 0$ if $\overrightarrow{\mathbf{xy}}$ is timelike, and non-positive otherwise. $\chi^2_{\mathcal{V}}$ is defined for any pair of points $(\mathbf{x}, \mathbf{y})$ by using extrinsic geometry, whether $\overrightarrow{\mathbf{xy}}$ is defined or not (see details in Section C.3).

## 4 RELATED WORK

Exploiting spacetimes to represent directed graphs was proposed by Clough & Evans (2017) to extend MultiDimensional Scaling (MDS) (Kruskal, 1964) to the simplest spacetime $\mathbb{R}^d_1$. As in standard MDS, a target distance matrix between nodes is given as input and the goal of the method is to find the vector representations of nodes that return the best approximation of the target distance matrix when using the squared Lorentzian distance as the dissimilarity function. However, as discussed in Section 3.3, it is difficult to define target distances between pairs of nodes that are not connected. Moreover, the formulation in (Clough & Evans, 2017) considers only $\mathbb{R}^d_1$ and does not account for future or past-direction between pairs of nodes during training. We solve this problem in Section 3.2 by enforcing future timelike geodesics to have their length smaller than some threshold $\varepsilon \in (0, \infty]$.

Sim et al. (2021) extended Clough & Evans (2017) to the anti-de Sitter space and Lorentz cylinder. Although our motivation is similar, our contributions are methodological, rely on the intuitions of Lorentzian pre-length spaces, and provide an easier interpretation of the learned representations as we explain below. First, Sim et al. (2021) do not address clearly the case when there is no geodesic between pairs of points, and their optimization framework leads to a distance loss term with a zero gradient in this case, which is difficult to optimize. Moreover, their prediction of an arc between a pair of nodes is determined via a *Triple Fermi-Dirac* (TFD) probability function that accounts for the distance between the nodes, the time coordinate difference $\Delta t$ and its opposite value $-\Delta t$ (i.e., TFD accounts for both the chronological future and past of a given node). In contrast, we restrict the representation of nodes connected by an arc to belong to $\mathcal{I}^+(\mathbf{x}, \mathcal{V}_\mathbf{x})$ where $\mathcal{V}_\mathbf{x}$ is an open convex normal neighborhood, which is simple to interpret and optimize. In general, the Lorentzian distance function from $\mathbf{x}$ to $\mathbf{y}$ on $\mathcal{M}$ is defined to be infinite if $\mathcal{M}$ is non-chronological and there exists a closed timelike curve joining $\mathbf{x}$ and $\mathbf{y}$. Moreover, the Hopf-Rinow theorem does not hold for spacetimes. Working with convex normal neighborhoods allows us to restrict the existence of arcs between nodes to the existence of geodesics joining events. The definition of time separation functions also becomes straightforward and we use their sign to determine the direction of an edge. Our framework shares similarities with Sim et al. (2021) when $\mathcal{M} = \mathbb{R}^d_1 = \mathcal{V}_\mathbf{x}$ because $\mathbb{R}^d_1$ is globally hyperbolic. Otherwise, we do not formulate our time separation function in the same way and we create arcs differently. Further comparisons with Sim et al. (2021) are provided in Appendix E.

The negative of the pseudo-Riemannian gradient is generally not a descent direction. We then use the optimization tools introduced in Gao et al. (2018); Law & Stam (2020); Law (2021) as described in Appendix F to learn our representations. Our approach could be extended to time-oriented pseudo-Riemannian manifolds of index $\nu > 1$ (see pages 240-242 of (O'Neill, 1983)). We also have $\mathbf{x} \ll \mathbf{y}$ if there exists a future-directed timelike curve joining them. Lemma 3.1 that exploits parallel transport can for instance be generalized to timelike tangent vectors $\mathbf{t}_1, \ldots, \mathbf{t}_\nu$ to define different types of arcs.

Some Riemannian approaches (Bordes et al., 2013; Ganea et al., 2018; Vilnis et al., 2018) have been used to represent DAGs. Ganea et al. (2018) consider entailment relations where $\mathbf{x} \ll \mathbf{y}$ means that $\mathbf{y}$ is a subconcept of $\mathbf{x}$ by constructing cones for each node representation in hyperbolic geometry. The DAGs they consider are *directed trees* and their underlying undirected graphs are trees (i.e., graphs without cycles). We propose to consider the chronological future and past that are intrinsic to our manifolds and have been studied for decades to define the causal structure. For instance, it is known that causal relations induce partially ordered sets (called *causal sets*) on causal spacetimes and can then represent DAGs (Bombelli et al., 1987). Our framework can represent DAGs that are not directed trees (see example in Appendix G.1) and is not limited to DAGs (see examples in Appendix G.2).

Table 1: Link prediction for directed graphs. Median average precision (AP) percentages across 20 random initializations on a held-out test set.

| Dataset | DREAM 5: S. cerevisiae | | | | | DREAM5 : in silico | | | | |
|---|---|---|---|---|---|---|---|---|---|---|
| Manifold dimensionality $d$ | 3 | 5 | 10 | 50 | 100 | 3 | 5 | 10 | 50 | 100 |
| Euclidean + FD | 33.0 | 34.2 | 40.2 | 44.5 | 49.0 | 29.4 | 32.9 | 39.7 | 39.8 | 34.8 |
| Hyperboloid + FD | 29.2 | 37.9 | 46.5 | 48.8 | 47.9 | 28.8 | 46.8 | 50.8 | 50.9 | 52.5 |
| Minkowski + TFD | 34.7 | 38.6 | 46.4 | 52.7 | 54.0 | 36.3 | 43.1 | 51.2 | 57.7 | 58.0 |
| Anti de-Sitter + TFD | 37.2 | 41.3 | 44.9 | 47.5 | 49.4 | 38.1 | 45.2 | 51.9 | 55.6 | 56.0 |
| Cylindrical Minkowski + TFD | 37.4 | 42.7 | 46.8 | 53.4 | 54.6 | 41.0 | 48.4 | 56.3 | 58.9 | 61.0 |
| Cylindrical Minkowski + equation 6 | **50.0** | **52.5** | 55.2 | **56.2** | **55.7** | **52.5** | 56.5 | 59.8 | 60.4 | 60.8 |
| de Sitter + equation 6 | 44.8 | 51.6 | **55.6** | 55.3 | 55.4 | 48.5 | **57.4** | **62.0** | **60.6** | **61.1** |

## 5 EXPERIMENTS

We show how our framework can represent graphs with directed cycles and predict links effectively in Section 5.1. We also show how the causal interpretation of our model can be used to represent hierarchical graphs with cycles in Section 5.2. We provide experiments on DAGs in Appendix D.1.

### 5.1 LINK PREDICTION ON GRAPHS WITH DIRECTED CYCLES

We now consider the link prediction task on the *Saccharomyces cerevisiae*, *in silico* and *Escherichia coli* DREAM5 datasets (Marbach et al., 2012) (see values of hyperparameters, results on *Escherichia coli* with standard deviations and more discussion in Appendix D.2). We follow the experimental protocol of Sim et al. (2021) by considering only the positive-regulatory nodes from the networks mentioned above while omitting the gene-expression data itself. Each network is randomly split into train and test sets, following 85/15 splits, and a part of the training set is used for validation.

Sim et al. (2021) use the Minkowski space $\mathbb{R}_1^d$, the anti de-Sitter space $\mathbb{H}_1^d(r)$, and the Cylindrical Minkowski space $\mathbb{L}_1^d(C)$. Both $\mathbb{H}_1^d(r)$ and $\mathbb{L}_1^d(C)$ are non-chronological and can then represent graphs with directed cycles. Sim et al. (2021) design a probability function that they call *Triple Fermi-Dirac (TFD)* specifically for their Cylindrical Minkowski space.

Instead, we propose to define the chronological future $\mathcal{I}^+(\mathbf{x}, \mathcal{V}_\mathbf{x})$ of $\mathbf{x} \in \mathbb{L}_1^d(C)$ such that if $\overrightarrow{\mathbf{xy}}$ is timelike, we have $\mathbf{y} \in \mathcal{I}^+(\mathbf{x}, \mathcal{V}_\mathbf{x})$ if $\exists k \in \mathbb{Z}, y_0 + kC \in (x_0, x_0 + C/2)$. Similarly, the chronological past $\mathcal{I}^-(\mathbf{x}, \mathcal{V}_\mathbf{x})$ is defined such that if $\overrightarrow{\mathbf{xy}}$ is timelike, we have $\mathbf{y} \in \mathcal{I}^-(\mathbf{x}, \mathcal{V}_\mathbf{x})$ if $\exists k \in \mathbb{Z}, y_0 + kC \in (x_0 - C/2, x_0)$. In detail, we define the time separation function for $\mathbb{L}_1^d(C)$ as follows: $\tau(\mathbf{x}, \mathbf{y}) := \left( ((y_0 - x_0 + \frac{C}{2}) \bmod C) - \frac{C}{2} \right) \in [-\frac{C}{2}, \frac{C}{2})$ where we use the modulo operation for real values which can be written as follows: $a \bmod b := a - b \cdot \lfloor \frac{a}{b} \rfloor$, and $\lfloor \cdot \rfloor$ is the floor function. Following our spacetime interpretation, we propose to learn our representations by optimizing:

$$\min_{\{\mathbf{x}_k \in \mathcal{M}\}_{k=1}^n} - \sum_{(v_i, v_j) \in E} \log(F(\mathbf{x}_i, \mathbf{x}_j)) - \sum_{(v_a, v_b) \notin E} \log(1 - F(\mathbf{x}_a, \mathbf{x}_b)) \tag{6}$$

where $F(\mathbf{x}_i, \mathbf{x}_j) := \sigma_{\theta_1}^m(\chi_\mathcal{V}^2(\mathbf{x}_i, \mathbf{x}_j)) \cdot \sigma_{\theta_2}^m(\tau(\mathbf{x}_i, \mathbf{x}_j))$, $\theta_1, \theta_2 > 0$ are temperature hyperparameters, $m > 0$ is an exponent, and $\sigma_\theta(x) := 1/(1 + e^{-x/\theta})$ is the sigmoid function. We refer to Section C.2 and C.3 for the formulations of the time separation function $\tau$ and the squared Lorentzian distance $\chi_\mathcal{V}^2$, respectively. This promotes future-directed timelike geodesics only between nodes connected by an arc. We report in Table 1 the Average Precision scores of our method (i.e., using equation 6) that outperforms baselines reported in Sim et al. (2021). The non-chronological cylindrical Minkowski space obtains a higher performance gap in low-dimensional space. This suggests it accounts for directed cycles better than chronological spacetimes. On the other hand, the gap is reduced in higher dimension except on *Escherichia coli* where the gap increases (see Table 4).

### 5.2 HIERARCHY EXTRACTION ON A SOCIAL NETWORK DATASET

Pseudo-Riemannian manifolds were introduced in the machine learning literature to extract hierarchies in graphs with cycles (Law & Stam, 2020). Spacetimes are a special family of pseudo-Riemannian manifolds whose time-orientation was not taken into account in Law & Stam (2020). In this paper, we found that spacetimes are able to consistently outperform other pseudo-Riemannian manifold approaches by exploiting the causality interpretation implicit in the squared Lorentzian distance.

Table 2: Evaluation scores for the different learned representations (mean $\pm$ standard deviation). $\downarrow$ the lower the metric, the better. $\uparrow$ the larger the metric (in absolute value), the better.

| Manifold | Problem | $d(\mathbf{x}, \mathbf{y})$ | Rank of 1st leader ($\downarrow$) | Rank of 2nd leader ($\downarrow$) | Top 5 $\rho$ ($\uparrow$) | Top 10 $\rho$ ($\uparrow$) |
|---|---|---|---|---|---|---|
| $\mathbb{R}^2$ (Euclidean) | equation 7 | $d_\gamma(\mathbf{x}, \mathbf{y})$ | $11.4 \pm 4.3$ | $14.0 \pm 2.4$ | $-0.17 \pm 0.70$ | $-0.19 \pm 0.40$ |
| $\mathbb{P}_0^2(r)$ (hyperbolic) | equation 7 | $d_\gamma(\mathbf{x}, \mathbf{y})$ | $7.0 \pm 1.4$ | $8.8 \pm 1.1$ | $-0.47 \pm 0.31$ | $-0.26 \pm 0.13$ |
| $\mathbb{P}_2^2(r)$ (elliptic) | equation 7 | $d_\gamma(\mathbf{x}, \mathbf{y})$ | $7.5 \pm 0.5$ | $9.1 \pm 1.2$ | $-0.59 \pm 0.03$ | $-0.38 \pm 0.08$ |
| $\mathbb{P}_1^2(r)$ | equation 7 | $d_\gamma(\mathbf{x}, \mathbf{y})$ | $2.2 \pm 1.3$ | $6.4 \pm 2.2$ | $0.28 \pm 0.26$ | $0.55 \pm 0.20$ |
| $\mathbb{P}_1^2(r)$ | equation 7 | $\chi_{\mathcal{U}}^2(\mathbf{x}, \mathbf{y})$ | $1.1 \pm 0.3$ | $2.8 \pm 1.3$ | $0.60 \pm 0.23$ | $0.82 \pm 0.16$ |
| $\mathbb{R}^3$ (Euclidean) | equation 7 | $d_\gamma(\mathbf{x}, \mathbf{y})$ | $9.4 \pm 1.3$ | $11.0 \pm 0.5$ | $-0.50 \pm 0.15$ | $-0.38 \pm 0.10$ |
| $\mathbb{P}_0^3(r)$ (hyperbolic) | equation 7 | $d_\gamma(\mathbf{x}, \mathbf{y})$ | $4.3 \pm 1.1$ | $6.1 \pm 1.1$ | $0.05 \pm 0.39$ | $0.02 \pm 0.15$ |
| $\mathbb{P}_3^3(r)$ (elliptic) | equation 7 | $d_\gamma(\mathbf{x}, \mathbf{y})$ | $4.3 \pm 1.1$ | $5.9 \pm 1.4$ | $0.09 \pm 0.38$ | $0.11 \pm 0.22$ |
| $\mathbb{P}_1^3(r)$ | equation 7 | $d_\gamma(\mathbf{x}, \mathbf{y})$ | $1.2 \pm 0.4$ | $3.8 \pm 1.6$ | $0.59 \pm 0.33$ | $0.63 \pm 0.13$ |
| $\mathbb{P}_2^3(r)$ | equation 7 | $d_\gamma(\mathbf{x}, \mathbf{y})$ | $\mathbf{1.0 \pm 0.0}$ | $4.3 \pm 2.1$ | $0.46 \pm 0.20$ | $0.61 \pm 0.14$ |
| $\mathbb{S}_1^3(r)$ (de Sitter) | equation 7 | $\chi_{\mathcal{U}}^2(\mathbf{x}, \mathbf{y})$ | $1.5 \pm 1.3$ | $4.1 \pm 2.0$ | $0.41 \pm 0.59$ | $0.58 \pm 0.22$ |
| $\mathbb{P}_1^3(r)$ | equation 7 | $\chi_{\mathcal{U}}^2(\mathbf{x}, \mathbf{y})$ | $1.1 \pm 0.3$ | $2.7 \pm 0.8$ | $0.56 \pm 0.29$ | $0.85 \pm 0.07$ |
| $\mathbb{R}^4$ (Euclidean) | equation 7 | $d_\gamma(\mathbf{x}, \mathbf{y})$ | $4.6 \pm 1.0$ | $6.9 \pm 0.7$ | $0.06 \pm 0.45$ | $0.04 \pm 0.19$ |
| $\mathbb{P}_0^4(r)$ (hyperbolic) | equation 7 | $d_\gamma(\mathbf{x}, \mathbf{y})$ | $2.5 \pm 0.7$ | $3.8 \pm 1.0$ | $0.36 \pm 0.22$ | $0.38 \pm 0.18$ |
| $\mathbb{P}_4^4(r)$ (elliptic) | equation 7 | $d_\gamma(\mathbf{x}, \mathbf{y})$ | $2.5 \pm 0.8$ | $3.6 \pm 0.7$ | $0.46 \pm 0.29$ | $0.38 \pm 0.26$ |
| $\mathbb{P}_1^4(r)$ | equation 7 | $d_\gamma(\mathbf{x}, \mathbf{y})$ | $1.2 \pm 0.4$ | $2.7 \pm 0.7$ | $0.62 \pm 0.23$ | $0.73 \pm 0.12$ |
| $\mathbb{P}_2^4(r)$ | equation 7 | $d_\gamma(\mathbf{x}, \mathbf{y})$ | $1.3 \pm 0.7$ | $3.1 \pm 1.0$ | $0.61 \pm 0.28$ | $0.72 \pm 0.07$ |
| $\mathbb{P}_3^4(r)$ | equation 7 | $d_\gamma(\mathbf{x}, \mathbf{y})$ | $1.2 \pm 0.4$ | $4.4 \pm 3.0$ | $0.63 \pm 0.35$ | $0.63 \pm 0.16$ |
| $\mathbb{P}_1^4(r)$ | equation 7 | $\chi_{\mathcal{U}}^2(\mathbf{x}, \mathbf{y})$ | $1.2 \pm 0.4$ | $3.0 \pm 1.1$ | $0.56 \pm 0.34$ | $0.77 \pm 0.13$ |
| $\mathbb{R}_1^2$ (Minkowski) | equation 8 | $\chi_{\mathcal{U}}^2(\mathbf{x}, \mathbf{y})$ | $\mathbf{1.1 \pm 0.3}$ | $2.6 \pm 0.8$ | $0.71 \pm 0.19$ | $0.82 \pm 0.08$ |
| $\mathbb{P}_1^2(r)$ | equation 8 | $\chi_{\mathcal{U}}^2(\mathbf{x}, \mathbf{y})$ | $1.3 \pm 0.5$ | $\mathbf{2.5 \pm 0.5}$ | $\mathbf{0.75 \pm 0.20}$ | $\mathbf{0.90 \pm 0.07}$ |
| $\mathbb{R}_1^3$ (Minkowski) | equation 8 | $\chi_{\mathcal{U}}^2(\mathbf{x}, \mathbf{y})$ | $\mathbf{1.0 \pm 0.0}$ | $2.3 \pm 0.5$ | $0.70 \pm 0.17$ | $0.80 \pm 0.11$ |
| $\mathbb{P}_1^3(r)$ | equation 8 | $\chi_{\mathcal{U}}^2(\mathbf{x}, \mathbf{y})$ | $1.3 \pm 0.5$ | $2.4 \pm 0.5$ | $0.69 \pm 0.29$ | $\mathbf{0.90 \pm 0.07}$ |
| $\mathbb{S}_1^3(r)$ (de Sitter) | equation 8 | $\chi_{\mathcal{U}}^2(\mathbf{x}, \mathbf{y})$ | $1.1 \pm 0.3$ | $\mathbf{2.1 \pm 0.3}$ | $\mathbf{0.86 \pm 0.15}$ | $0.88 \pm 0.05$ |
| $\mathbb{R}_1^4$ (Minkowski) | equation 8 | $\chi_{\mathcal{U}}^2(\mathbf{x}, \mathbf{y})$ | $\mathbf{1.0 \pm 0.0}$ | $\mathbf{2.3 \pm 0.9}$ | $0.70 \pm 0.22$ | $0.81 \pm 0.13$ |
| $\mathbb{P}_1^4(r)$ | equation 8 | $\chi_{\mathcal{U}}^2(\mathbf{x}, \mathbf{y})$ | $1.1 \pm 0.3$ | $2.4 \pm 0.5$ | $\mathbf{0.78 \pm 0.13}$ | $\mathbf{0.91 \pm 0.07}$ |

**Graph dataset.** We consider an undirected graph $G = (V, E)$ where $V = \{v_i\}_{i=1}^n$ is the node set and $E$ is the edge set such that $(v_i, v_j) \in E$ indicates that $v_i$ and $v_j$ are connected by an edge. Zachary's karate club dataset (Zachary, 1977) is a social network of $n = 34$ members of the karate club, each represented by a node $v_i$. The club was split in two factions due to a conflict between the instructor (node $v_1$) and the administrator (node $v_{34}$) that are the two most important members (i.e., leaders) of the dataset. The other members had to decide between joining the new club created by $v_1$ or stay with $v_{34}$. Two nodes are joined by an edge $(v_i, v_j) \in E$ if the members are friends.

In the original ultrahyperbolic approaches (Law & Stam, 2020; Law, 2021), each node $v_i$ is represented by a point $\mathbf{x}_i \in \mathcal{M}$ on some pseudo-Riemannian manifold $\mathcal{M}$. The goal is to learn embeddings $\{\mathbf{x}_i\}_{i=1}^n$ so that pairs of nodes joined by an edge $(v_i, v_j) \in E$ are closer to each other than pairs of nodes not joined by an edge. To this end, the embeddings are learned in Law & Stam (2020); Law (2021) by minimizing the following problem:

$$\min_{\{\mathbf{x}_k \in \mathcal{M}\}_{k=1}^n} - \sum_{(v_i, v_j) \in E} \log \frac{e^{-d(\mathbf{x}_i, \mathbf{x}_j)/\theta}}{e^{-d(\mathbf{x}_i, \mathbf{x}_j)/\theta} + \sum_{(v_a, v_b) \notin E} e^{-d(\mathbf{x}_a, \mathbf{x}_b)/\theta}} \tag{7}$$

where the dissimilarity function $d(\mathbf{x}_i, \mathbf{x}_j)$ is the arc length $d_\gamma(\mathbf{x}_i, \mathbf{x}_j) := \sqrt{|\langle \overrightarrow{\mathbf{x}_i \mathbf{x}_j}, \overrightarrow{\mathbf{x}_i \mathbf{x}_j} \rangle|}$ of the geodesic joining $\mathbf{x}_i$ and $\mathbf{x}_j$. If $\mathcal{M}$ is Riemannian, $d_\gamma$ is the Riemannian distance. In this paper, we propose instead to consider in equation 7 that $d$ is the squared Lorentzian distance $\chi_{\mathcal{U}}^2$, as defined in Section 3.3, where $\mathcal{U}_\mathbf{x}$ is the (convex) maximal normal neighborhood of $\mathbf{x} \in \mathcal{M}$.

We also propose to simply enforce the embeddings of $v_i$ and $v_j$ to be joined by a timelike geodesic iff $(v_i, v_j) \in E$. Since $G$ is an undirected graph, the future- or past-direction is not provided so we do not constrain the future-direction of the geodesics during training. Our problem formulation is then:

$$\min_{\{\mathbf{x}_k \in \mathcal{M}\}_{k=1}^n} \sum_{(v_a, v_b) \notin E} \sigma_\theta\left(d(\mathbf{x}_a, \mathbf{x}_b)\right) + \lambda \sum_{(v_i, v_j) \in E} \sigma_\theta\left(-d(\mathbf{x}_i, \mathbf{x}_j)\right) \tag{8}$$

where $\lambda > 0$ is a regularization parameter and we set $d(\mathbf{x}_i, \mathbf{x}_j) = \chi_{\mathcal{U}}^2(\mathbf{x}_i, \mathbf{x}_j)$. The main goal of equation 8 is to enforce pairs of nodes $(v_i, v_j) \in E$ to be joined by a timelike geodesic and pairs $(v_a, v_b) \notin E$ to be joined by a spacelike geodesic since there is no causality between them.

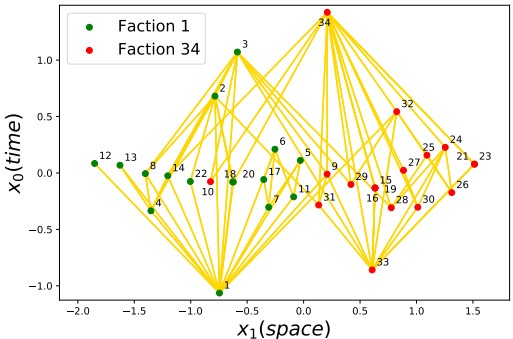 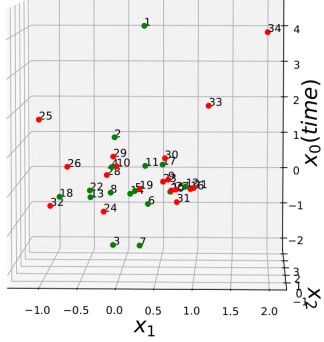

Figure 2: (left) Coordinates of 2-dimensional embeddings $\mathbf{x} = (x_0, x_1)^\top$ learned with equation 8 when $\mathcal{M} = \mathbb{R}_1^2$. (right) Coordinates of the first three coordinates of embeddings $\mathbf{x} = (x_0, x_1, x_2, x_3)^\top$ learned with equation 8 when $\mathcal{M} = \mathbb{S}_1^3(r)$. In Lorentz geometry, a timelike geodesic joining two points is the longest timelike curve in a given convex normal neighborhood. This translates in the high-level nodes $v_1$ and $v_{34}$ being the furthest from the rest of the nodes. The ground truth edges are plotted in yellow and the node color corresponds to the joined faction. A small number of spacelike edges are visible (those edges more than 45 degrees from vertical).

**Evaluation metrics.** We report in Table 2 the results obtained when optimizing equation 7 or equation 8 using different dissimilarity functions. At test time, a score $\delta_i = \sum_{j=1}^n \mathsf{d}(\mathbf{x}_i, \mathbf{x}_j)$ summing pairwise distances is assigned to each node $v_i$ and used as an indicator of importance in the hierarchical graph. Following Section 3.3, when the dissimilarity function is the arc length $\mathsf{d}_\gamma$ (resp. squared Lorentzian distance $\chi_\mathcal{U}^2$), we sort the scores $\delta_1, \ldots, \delta_n$ is ascending (resp. descending) order and report the ranks of the two leaders $v_1$ and $v_n$ (in no particular order) in the first two columns of Table 2 averaged over 10 different initializations. We also report in Table 2 the Spearmans rank correlation coefficient $\rho$ (Spearman, 1904) between the ordered $\delta_i$ scores and the order of the 5 and 10 most important nodes which are 34, 1, 33, 3, 2, 32, 24, 4, 9, 14 (in that order, see Law & Stam (2020)).

**Results.** In general, the squared Lorentzian distance $\chi_\mathcal{U}^2$ returns better performance than the arc length $\mathsf{d}_\gamma$. Moreover, the optimization problem in equation 8 enforcing timelike geodesics between connected nodes (i.e., causality) and spacelike geodesics between unconnected nodes (i.e., non-causality) extracts the most important nodes in low-dimensional spacetimes more effectively. The predicted $\rho$ also shows higher rank correlation with $\chi_\mathcal{U}^2$. It is worth noting that hyperbolic geometry was shown relevant to represent graphs without cycles (Gromov, 1987) but our hierarchical graph contains cycles, which explains why geometries other than hyperbolic may be more relevant. The performance gap with non-Riemannian baselines (Law & Stam, 2020; Law, 2021) can be explained by the fact that those baselines compare the arc lengths of different types of geodesics (i.e., timelike or spacelike) that are not necessarily comparable. On the other hand, our framework sets non-causal distances to zero at test time, and compares only the lengths of causal geodesics. Figure 2 illustrates spacetime diagrams of the learned representations for $\mathbb{R}_1^2$ and $\mathbb{S}_1^3(r)$. The most important nodes tend to be further away from the other nodes with respect to the Lorentzian distance. Most ground truth edges are timelike geodesics and belong to the chronological past or future of an edge representation.

We report in Appendix D.3 the same kind of hierarchy extraction experiment on a dataset that describes co-authorship information from papers published at NIPS from 1988 to 2003 (Globerson et al., 2007). The number of nodes/authors is $|V| = 2715$ and the number of edges is $|E| = 4733$.

## 6 Conclusion

We have proposed a general framework to represent data, and in particular nodes of a directed graph, in a spacetime. Compared to previous work, our framework is properly endowed with the structure of a Lorentzian pre-length space, which makes it easy to optimize and applicable to a large family of spacetimes. In the case of hierarchical undirected graphs, we show that we can enforce causality between nodes connected by an edge to extract the most important nodes in the hierarchy.

## ACKNOWLEDGMENTS

We thank Yuval Atzmon, Gal Chechik and Haggai Maron for helpful discussions during the initial phase of this project. We also thank Aaron Sim for sharing his source code, Rafid Mahmood, Jos Stam and the anonymous reviewers for valuable feedback on early versions of the manuscript.

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

## A    SUMMARY OF SUPPLEMENTARY MATERIAL

The supplementary material is structured as follows:

• Section B gives the differential geometry tools that are general to pseudo-Riemannian manifolds of constant curvature (called *space forms*).

• Section C gives the differential geometry tools that are specific to our spacetimes such as the time separation function and the squared Lorentzian distance.

• Section D presents additional results and further experimental details.

• Section E is an extended related work section.

• Section F gives the details of the optimizers we use.

• Section G gives examples of directed graphs described with our model.

## B    DIFFERENTIAL GEOMETRY OF PSEUDO-RIEMANNIAN SPACE FORMS

We provide the necessary differential geometry tools to work on the pseudo-Euclidean space $\mathbb{R}_\nu^d$, the pseudo-sphere $\mathbb{S}_\nu^d(r)$ and the pseudo-hyperboloid $\mathbb{H}_\nu^d(r)$. Most of them are explained in (Law, 2021; Law & Stam, 2020). We refer the reader to (Hawking & Ellis, 1973; O'Neill, 1983; Wolf, 2011).

### B.1    PSEUDO-EUCLIDEAN SPACE

We recall that $\forall \nu, \mathbb{R}_\nu^d \approx T_\mathbf{x}\mathbb{R}_\nu^d$. Its geodesic $\gamma_{\mathbf{x}\to\mathbf{y}} : \mathbb{R} \to \mathbb{R}_\nu^d$ is $\gamma_{\mathbf{x}\to\mathbf{y}}(t) := \mathbf{x} + t\mathbf{y}$. The exponential map at $\mathbf{x}$ is defined as $\exp_\mathbf{x}(\mathbf{y}) := \gamma_{\mathbf{x}\to\mathbf{y}}(1) = \mathbf{x} + \mathbf{y}$ and its inverse is $\overrightarrow{\mathbf{xy}} := \exp_\mathbf{x}^{-1}(\mathbf{y}) = \mathbf{y} - \mathbf{x}$.

Moreover, $\forall \mathbf{x} \in \mathbb{R}_\nu^d, \mathbf{y} \in \mathbb{R}_\nu^d$, the parallel transport $\Gamma_\mathbf{x}^\mathbf{y} : T_\mathbf{x}\mathbb{R}_\nu^d \to T_\mathbf{y}\mathbb{R}_\nu^d$ can be defined as the identity function due to the isomorphism $\mathbb{R}_\nu^d \approx T_\mathbf{x}\mathbb{R}_\nu^d \approx T_\mathbf{y}\mathbb{R}_\nu^d$. See page 67 of O'Neill (1983).

### B.2    PSEUDO-SPHERE $\mathbb{S}_\nu^d(r)$

We give here the differential geometry tools specific to the pseudo-sphere which is defined as the following hypersurface: $\mathbb{S}_\nu^d(r) := \left\{ \mathbf{x} \in \mathbb{R}_\nu^{d+1} : \langle \mathbf{x}, \mathbf{x} \rangle_\nu = r^2 \right\}$. The tangent space $T_\mathbf{x}\mathbb{S}_\nu^d(r)$ of $\mathbb{S}_\nu^d(r)$ at $\mathbf{x}$ can be defined as: $T_\mathbf{x}\mathbb{S}_\nu^d(r) := \{\mathbf{u} \in \mathbb{R}_\nu^{d+1} : \langle \mathbf{u}, \mathbf{x} \rangle_\nu = 0\}$. In the case of the pseudo-sphere, we have $\forall \mathbf{u} \in T_\mathbf{x}\mathbb{S}_\nu^d(r), \forall \mathbf{v} \in T_\mathbf{x}\mathbb{S}_\nu^d(r), g_\mathbf{x}(\mathbf{u}, \mathbf{v}) = \langle \mathbf{u}, \mathbf{v} \rangle = \langle \mathbf{u}, \mathbf{v} \rangle_\nu$.

**Geodesic.** The geodesic $\gamma_{\mathbf{x}\to\mathbf{u}} : \mathbb{R} \to \mathbb{S}_\nu^d(r)$ satisfying $\gamma_{\mathbf{x}\to\mathbf{u}}(0) = \mathbf{x}$ and $\gamma'_{\mathbf{x}\to\mathbf{u}}(0) = \mathbf{u} \in T_\mathbf{x}\mathbb{S}_\nu^d(r)$ is formulated for all $t \in \mathbb{R}$:

$$\gamma_{\mathbf{x}\to\mathbf{u}}(t) := \begin{cases} \cos\left(\frac{t\sqrt{|\langle \mathbf{u},\mathbf{u}\rangle_\nu|}}{r}\right)\mathbf{x} + \frac{r}{\sqrt{|\langle \mathbf{u},\mathbf{u}\rangle_\nu|}}\sin\left(\frac{t\sqrt{|\langle \mathbf{u},\mathbf{u}\rangle_\nu|}}{r}\right)\mathbf{u} & \text{if } \langle \mathbf{u},\mathbf{u}\rangle_\nu > 0 \\ \mathbf{x} + t\mathbf{u} & \text{if } \langle \mathbf{u},\mathbf{u}\rangle_\nu = 0 \\ \cosh\left(\frac{t\sqrt{|\langle \mathbf{u},\mathbf{u}\rangle_\nu|}}{r}\right)\mathbf{x} + \frac{r}{\sqrt{|\langle \mathbf{u},\mathbf{u}\rangle_\nu|}}\sinh\left(\frac{t\sqrt{|\langle \mathbf{u},\mathbf{u}\rangle_\nu|}}{r}\right)\mathbf{u} & \text{if } \langle \mathbf{u},\mathbf{u}\rangle_\nu < 0 \end{cases} \quad (9)$$

**Exponential map.** The exponential map $\exp_\mathbf{x} : T_\mathbf{x}\mathbb{S}_\nu^d(r) \to \mathbb{S}_\nu^d(r)$ is defined such that $\forall \mathbf{u} \in T_\mathbf{x}\mathbb{S}_\nu^d(r), \exp_\mathbf{x}(\mathbf{u}) = \gamma_{\mathbf{x}\to\mathbf{u}}(1)$. We then have:

$$\exp_\mathbf{x}(\mathbf{u}) := \begin{cases} \cos\left(\frac{\sqrt{|\langle \mathbf{u},\mathbf{u}\rangle_\nu|}}{r}\right)\mathbf{x} + \frac{r}{\sqrt{|\langle \mathbf{u},\mathbf{u}\rangle_\nu|}}\sin\left(\frac{\sqrt{|\langle \mathbf{u},\mathbf{u}\rangle_\nu|}}{r}\right)\mathbf{u} & \text{if } \langle \mathbf{u},\mathbf{u}\rangle_\nu > 0 \\ \mathbf{x} + \mathbf{u} & \text{if } \langle \mathbf{u},\mathbf{u}\rangle_\nu = 0 \\ \cosh\left(\frac{\sqrt{|\langle \mathbf{u},\mathbf{u}\rangle_\nu|}}{r}\right)\mathbf{x} + \frac{r}{\sqrt{|\langle \mathbf{u},\mathbf{u}\rangle_\nu|}}\sinh\left(\frac{\sqrt{|\langle \mathbf{u},\mathbf{u}\rangle_\nu|}}{r}\right)\mathbf{u} & \text{if } \langle \mathbf{u},\mathbf{u}\rangle_\nu < 0 \end{cases} \quad (10)$$

**Logarithmic map.** The logarithmic map $\log_\mathbf{x}$ is the inverse of the exponential map $\exp_\mathbf{x}$ on a normal neighborhood of $\mathbf{x} \in \mathbb{S}_\nu^d(r)$ denoted by $\mathcal{U}_\mathbf{x} = \{\mathbf{y} \in \mathbb{S}_\nu^d(r) : \frac{\langle \mathbf{x},\mathbf{y}\rangle_\nu}{r^2} > -1\}$. It is formulated:

$$\forall \mathbf{y} \in \mathcal{U}_\mathbf{x}, \overrightarrow{\mathbf{xy}} := \log_\mathbf{x}(\mathbf{y}) := \begin{cases} \frac{\arccos\left(\frac{\langle \mathbf{x},\mathbf{y}\rangle_\nu}{r^2}\right)}{\sqrt{1-(\frac{\langle \mathbf{x},\mathbf{y}\rangle_\nu}{r^2})^2}}\left(\mathbf{y} - \frac{\langle \mathbf{x},\mathbf{y}\rangle_\nu}{r^2}\mathbf{x}\right) & \text{if } \frac{\langle \mathbf{x},\mathbf{y}\rangle_\nu}{r^2} \in (-1,1) \\ \mathbf{y} - \mathbf{x} & \text{if } \frac{\langle \mathbf{x},\mathbf{y}\rangle_\nu}{r^2} = 1 \\ \frac{\operatorname{arccosh}\left(\frac{\langle \mathbf{x},\mathbf{y}\rangle_\nu}{r^2}\right)}{\sqrt{(\frac{\langle \mathbf{x},\mathbf{y}\rangle_\nu}{r^2})^2-1}}\left(\mathbf{y} - \frac{\langle \mathbf{x},\mathbf{y}\rangle_\nu}{r^2}\mathbf{x}\right) & \text{if } \frac{\langle \mathbf{x},\mathbf{y}\rangle_\nu}{r^2} > 1 \end{cases} \quad (11)$$

where arccos is the inverse of the cosine function, and arccosh is the inverse of the hyperbolic cosine function. One can verify from equation 10 and equation 11 that $\overrightarrow{\mathbf{xy}}$ is timelike (i.e., $\langle \overrightarrow{\mathbf{xy}}, \overrightarrow{\mathbf{xy}} \rangle < 0$) iff $\langle \mathbf{x}, \mathbf{y} \rangle_\nu > r^2$.

**Geodesic "distance".** As explained in (Law & Stam, 2020) and Chapter 5 of (O'Neill, 1983), when the logarithmic map exists for some pseudo-Riemannian manifold $\mathcal{M}$, the arc length of the geodesic $\gamma_{\mathbf{x} \to \overrightarrow{\mathbf{xy}}}$ from $\mathbf{x} \in \mathcal{M}$ to $\mathbf{y} \in \mathcal{M}$ corresponds to the radius function: $\sqrt{|g_{\mathbf{x}}(\overrightarrow{\mathbf{xy}}, \overrightarrow{\mathbf{xy}})|}$ where $g_{\mathbf{x}} : T_{\mathbf{x}}\mathcal{M} \times T_{\mathbf{x}}\mathcal{M} \to \mathbb{R}$ is the metric tensor at $\mathbf{x}$ and $\overrightarrow{\mathbf{xy}} = \log_{\mathbf{x}}(\mathbf{y})$ is the logarithmic map. The geodesic distance $\mathsf{d}_\gamma : \mathbb{S}^d_\nu(r) \times \mathbb{S}^d_\nu(r) \to \mathbb{R}$ is then:

$$\mathsf{d}_\gamma(\mathbf{x}, \mathbf{y}) := \sqrt{|\langle \overrightarrow{\mathbf{xy}}, \overrightarrow{\mathbf{xy}} \rangle_\nu|} = \begin{cases} r \, \text{arccosh} \left( \frac{\langle \mathbf{x}, \mathbf{y} \rangle_\nu}{r^2} \right) & \text{if } \frac{\langle \mathbf{x}, \mathbf{y} \rangle_\nu}{r^2} \geq 1 \\ r \, \text{arccos} \left( \frac{\langle \mathbf{x}, \mathbf{y} \rangle_\nu}{r^2} \right) & \text{if } \frac{\langle \mathbf{x}, \mathbf{y} \rangle_\nu}{r^2} \in (-1, 1) \end{cases} \tag{12}$$

$\mathsf{d}_\gamma$ is not a "distance metric" but a symmetric premetric: it satisfies (i) $\mathsf{d}_\gamma(\mathbf{x}, \mathbf{y}) = \mathsf{d}_\gamma(\mathbf{y}, \mathbf{x}) \geq 0$ and (ii) $\mathsf{d}_\gamma(\mathbf{x}, \mathbf{x}) = 0$.

In (O'Neill, 1983), the "minimizing geodesic" is defined by its arc length and then also corresponds to the geodesic distance mentioned above.

Given the minimizing geodesic $\gamma$ connecting $\mathbf{x}$ to $\mathbf{y}$, the parallel transport $\Gamma^{\mathbf{y}}_{\mathbf{x}} : T_{\mathbf{x}}\mathbb{S}^d_\nu(r) \to T_{\mathbf{y}}\mathbb{S}^d_\nu(r)$ is a linear isometry such that $\forall \mathbf{u}, \mathbf{v}, \langle \mathbf{u}, \mathbf{v} \rangle_\nu = \langle \Gamma^{\mathbf{y}}_{\mathbf{x}}(\mathbf{u}), \Gamma^{\mathbf{y}}_{\mathbf{x}}(\mathbf{v}) \rangle_\nu$. The parallel transport along $\gamma$ from $\mathbf{x} = \gamma(0)$ to $\mathbf{y} = \gamma(1)$ (where $\mathbf{x}$ and $\mathbf{y}$ satisfy $\langle \mathbf{x}, \mathbf{y} \rangle_\nu > -r^2$) is:

$$\Gamma^{\mathbf{y}}_{\mathbf{x}}(\mathbf{u}) := \mathbf{u} - \frac{\langle \mathbf{y}, \mathbf{u} \rangle_\nu}{\langle \mathbf{x}, \mathbf{y} \rangle_\nu + r^2} (\mathbf{y} + \mathbf{x}) \tag{13}$$

**Theorem B.1** (Diffeomorphism (Wolf, 2011)). *There exists a diffeomorphism* $\psi : \mathbb{S}^d_\nu(r) \to \mathbb{R}^\nu \times \mathbb{S}^{d-\nu}_0(r)$. *Let us note* $\mathbf{x} = \begin{pmatrix} \mathbf{t} \\ \mathbf{s} \end{pmatrix} \in \mathbb{S}^d_\nu(r)$ *with* $\mathbf{t} \in \mathbb{R}^\nu$ *and* $\mathbf{s} \in \mathbb{R}^{d-\nu+1}_*$. *Let us note* $\mathbf{z} = \begin{pmatrix} \mathbf{t} \\ \mathbf{v} \end{pmatrix} \in \mathbb{R}^\nu \times \mathbb{S}^{d-\nu}_0(r)$ *where* $\mathbf{v} \in \mathbb{S}^{d-\nu}_0(r)$. *The mapping* $\psi$ *and its inverse* $\psi^{-1}$ *can be formulated:*

$$\psi(\mathbf{x}) = \begin{pmatrix} \mathbf{t} \\ \frac{r}{\|\mathbf{s}\|}\mathbf{s} \end{pmatrix} \qquad and \qquad \psi^{-1}(\mathbf{z}) = \begin{pmatrix} \mathbf{t} \\ \frac{\sqrt{r^2+\|\mathbf{t}\|^2}}{r}\mathbf{v} \end{pmatrix}. \tag{14}$$

*where* $\|.\|$ *is the standard Euclidean norm (i.e.,* $\|\mathbf{s}\| := \sqrt{\mathbf{s}^\top \mathbf{s}}$).

### B.3  PSEUDO-HYPERBOLOID $\mathbb{H}^d_\nu(r)$

We recall the differential geometry tools (from (Law & Stam, 2020)) specific to the pseudo-hyperboloid which is defined as the set: $\mathbb{H}^d_\nu(r) := \{\mathbf{x} \in \mathbb{R}^{d+1}_{\nu+1} : \langle \mathbf{x}, \mathbf{x} \rangle_{\nu+1} = -r^2\}$. The tangent space $T_{\mathbf{x}}\mathbb{H}^d_\nu(r)$ of $\mathbb{H}^d_\nu(r)$ at $\mathbf{x}$ can be defined as: $T_{\mathbf{x}}\mathbb{H}^d_\nu(r) := \{\mathbf{u} \in \mathbb{R}^{d+1}_{\nu+1} : \langle \mathbf{u}, \mathbf{x} \rangle_{\nu+1} = 0\}$. In the case of $\mathbb{H}^d_\nu(r)$, we have $\forall \mathbf{u} \in T_{\mathbf{x}}\mathbb{H}^d_\nu(r), \forall \mathbf{v} \in T_{\mathbf{x}}\mathbb{H}^d_\nu(r), g_{\mathbf{x}}(\mathbf{u}, \mathbf{v}) = \langle \mathbf{u}, \mathbf{v} \rangle = \langle \mathbf{u}, \mathbf{v} \rangle_{\nu+1}$.

**Geodesic.** The geodesic $\gamma_{\mathbf{x} \to \mathbf{u}} : \mathbb{R} \to \mathbb{H}^d_\nu(r)$ satisfying $\gamma_{\mathbf{x} \to \mathbf{u}}(0) = \mathbf{x}$ and $\gamma'_{\mathbf{x} \to \mathbf{u}}(0) = \mathbf{u} \in T_{\mathbf{x}}\mathbb{H}^d_\nu(r)$ is formulated for all $t \in \mathbb{R}$:

$$\gamma_{\mathbf{x} \to \mathbf{u}}(t) = \begin{cases} \cos\left(\frac{t\sqrt{|\langle \mathbf{u},\mathbf{u} \rangle_{\nu+1}|}}{r}\right)\mathbf{x} \ + \frac{r}{\sqrt{|\langle \mathbf{u},\mathbf{u} \rangle_{\nu+1}|}}\sin\left(\frac{t\sqrt{|\langle \mathbf{u},\mathbf{u} \rangle_{\nu+1}|}}{r}\right)\mathbf{u} & \text{if } \langle \mathbf{u}, \mathbf{u} \rangle_{\nu+1} < 0 \\ \mathbf{x} + t\mathbf{u} & \text{if } \langle \mathbf{u}, \mathbf{u} \rangle_{\nu+1} = 0 \\ \cosh\left(\frac{t\sqrt{|\langle \mathbf{u},\mathbf{u} \rangle_{\nu+1}|}}{r}\right)\mathbf{x} + \frac{r}{\sqrt{|\langle \mathbf{u},\mathbf{u} \rangle_{\nu+1}|}}\sinh\left(\frac{t\sqrt{|\langle \mathbf{u},\mathbf{u} \rangle_{\nu+1}|}}{r}\right)\mathbf{u} & \text{if } \langle \mathbf{u}, \mathbf{u} \rangle_{\nu+1} > 0 \end{cases} \tag{15}$$

**Exponential map.** The exponential map $\exp_{\mathbf{x}} : T_{\mathbf{x}}\mathbb{H}^d_\nu(r) \to \mathbb{H}^d_\nu(r)$ is defined such that $\forall \mathbf{u} \in T_{\mathbf{x}}\mathbb{H}^d_\nu(r), \exp_{\mathbf{x}}(\mathbf{u}) = \gamma_{\mathbf{x} \to \mathbf{u}}(1)$. We then have:

$$\exp_{\mathbf{x}}(\mathbf{u}) = \begin{cases} \cos\left(\frac{\sqrt{|\langle \mathbf{u},\mathbf{u} \rangle_{\nu+1}|}}{r}\right)\mathbf{x} \ + \frac{r}{\sqrt{|\langle \mathbf{u},\mathbf{u} \rangle_{\nu+1}|}}\sin\left(\frac{\sqrt{|\langle \mathbf{u},\mathbf{u} \rangle_{\nu+1}|}}{r}\right)\mathbf{u} & \text{if } \langle \mathbf{u}, \mathbf{u} \rangle_{\nu+1} < 0 \\ \mathbf{x} + \mathbf{u} & \text{if } \langle \mathbf{u}, \mathbf{u} \rangle_{\nu+1} = 0 \\ \cosh\left(\frac{\sqrt{|\langle \mathbf{u},\mathbf{u} \rangle_{\nu+1}|}}{r}\right)\mathbf{x} + \frac{r}{\sqrt{|\langle \mathbf{u},\mathbf{u} \rangle_{\nu+1}|}}\sinh\left(\frac{\sqrt{|\langle \mathbf{u},\mathbf{u} \rangle_{\nu+1}|}}{r}\right)\mathbf{u} & \text{if } \langle \mathbf{u}, \mathbf{u} \rangle_{\nu+1} > 0 \end{cases} \tag{16}$$

**Logarithmic map.** The logarithmic map $\log_{\mathbf{x}}$ is defined as the inverse of the exponential map $\exp_{\mathbf{x}}$ on a normal neighborhood of $\mathbf{x} \in \mathbb{H}^d_\nu(r)$ denoted by $\mathcal{U}_{\mathbf{x}} = \{\mathbf{y} \in \mathbb{H}^d_\nu(r) : \frac{\langle \mathbf{x}, \mathbf{y} \rangle_{\nu+1}}{r^2} < 1\}$. It is formulated:

$$\forall \mathbf{y} \in \mathcal{U}_{\mathbf{x}}, \; \overrightarrow{\mathbf{xy}} = \log_{\mathbf{x}}(\mathbf{y}) = \begin{cases} \frac{\operatorname{arccosh}\left(-\frac{\langle \mathbf{x}, \mathbf{y} \rangle_{\nu+1}}{r^2}\right)}{\sqrt{(\frac{\langle \mathbf{x}, \mathbf{y} \rangle_{\nu+1}}{r^2})^2 - 1}} \left(\mathbf{y} + \frac{\langle \mathbf{x}, \mathbf{y} \rangle_{\nu+1}}{r^2}\mathbf{x}\right) & \text{if } \frac{\langle \mathbf{x}, \mathbf{y} \rangle_{\nu+1}}{r^2} < -1 \\ \mathbf{y} - \mathbf{x} & \text{if } \frac{\langle \mathbf{x}, \mathbf{y} \rangle_{\nu+1}}{r^2} = -1 \\ \frac{\arccos\left(-\frac{\langle \mathbf{x}, \mathbf{y} \rangle_{\nu+1}}{r^2}\right)}{\sqrt{1 - (\frac{\langle \mathbf{x}, \mathbf{y} \rangle_{\nu+1}}{r^2})^2}} \left(\mathbf{y} + \frac{\langle \mathbf{x}, \mathbf{y} \rangle_{\nu+1}}{r^2}\mathbf{x}\right) & \text{if } \frac{\langle \mathbf{x}, \mathbf{y} \rangle_{\nu+1}}{r^2} \in (-1, 1) \end{cases}$$

(17)

One can verify that a nonconstant geodesic from $\mathbf{x}$ to $\mathbf{y}$ is timelike iff $\langle \mathbf{x}, \mathbf{y} \rangle_\nu \in (-r^2, r^2)$ or $\mathbf{y} = \pm \mathbf{x}$.

**Geodesic "distance".** The geodesic distance $\mathsf{d}_\gamma : \mathbb{H}^d_\nu(r) \times \mathbb{H}^d_\nu(r) \to \mathbb{R}$ is then:

$$\mathsf{d}_\gamma(\mathbf{x}, \mathbf{y}) = \sqrt{|\langle \overrightarrow{\mathbf{xy}}, \overrightarrow{\mathbf{xy}} \rangle_{\nu+1}|} = \begin{cases} r \operatorname{arccosh}\left(-\frac{\langle \mathbf{x}, \mathbf{y} \rangle_{\nu+1}}{r^2}\right) & \text{if } \frac{\langle \mathbf{x}, \mathbf{y} \rangle_{\nu+1}}{r^2} \leq -1 \\ r \arccos\left(-\frac{\langle \mathbf{x}, \mathbf{y} \rangle_{\nu+1}}{r^2}\right) & \text{if } \frac{\langle \mathbf{x}, \mathbf{y} \rangle_{\nu+1}}{r^2} \in (-1, 1) \end{cases}$$

(18)

**Parallel transport on $\mathbb{H}^d_\nu(r)$.** The parallel transport connecting $\mathbf{x} \in \mathbb{H}^d_\nu(r)$ to $\mathbf{y} \in \mathbb{H}^d_\nu(r)$ is:

$$\Gamma^{\mathbf{y}}_{\mathbf{x}}(\mathbf{u}) := \mathbf{u} - \frac{\langle \mathbf{y}, \mathbf{u} \rangle_{\nu+1}}{\langle \mathbf{x}, \mathbf{y} \rangle_{\nu+1} - r^2}(\mathbf{y} + \mathbf{x}) \quad \text{where} \quad \langle \mathbf{x}, \mathbf{y} \rangle_{\nu+1} < r^2$$

(19)

**Theorem B.2** (Diffeomorphism (Wolf, 2011))**.** *There exists a diffeomorphism* $\psi : \mathbb{H}^d_\nu(r) \to \mathbb{S}^\nu_0(r) \times \mathbb{R}^{d-\nu}$. *Let us note* $\mathbf{x} = \begin{pmatrix} \mathbf{t} \\ \mathbf{s} \end{pmatrix} \in \mathbb{H}^d_\nu(r)$ *with* $\mathbf{t} \in \mathbb{R}^{\nu+1}_*$ *and* $\mathbf{s} \in \mathbb{R}^{d-\nu}$. *Let us note* $\mathbf{z} = \begin{pmatrix} \mathbf{u} \\ \mathbf{v} \end{pmatrix} \in \mathbb{S}^\nu_0(r) \times \mathbb{R}^{d-\nu}$ *where* $\mathbf{u} \in \mathbb{S}^\nu_0(r)$ *and* $\mathbf{v} \in \mathbb{R}^{d-\nu}$. *The mapping* $\psi$ *and its inverse* $\psi^{-1}$ *can be formulated:*

$$\psi(\mathbf{x}) = \begin{pmatrix} \frac{r}{\|\mathbf{t}\|}\mathbf{t} \\ \mathbf{s} \end{pmatrix} \quad \text{and} \quad \psi^{-1}(\mathbf{z}) = \begin{pmatrix} \frac{\sqrt{r^2 + \|\mathbf{v}\|^2}}{r}\mathbf{u} \\ \mathbf{v} \end{pmatrix}.$$

(20)

**The anti-de Sitter spacetime** $\mathbb{H}^d_1(r)$ is non-chronological and satisfies $\mathbf{x} \ll \mathbf{y} \implies \mathbf{y} \ll \mathbf{x}$, which is convenient to represent graphs with directed cycles. It was used in Sim et al. (2021) to represent directed graphs. Nonetheless, it is worth noting that the problem formulation of Sim et al. (2021) for $\mathbb{H}^d_1(r)$ also promotes arcs (i.e., causal relation) between pairs of nodes that are not connected by any geodesic, which makes their problem hard to optimize. Following general relativity, we only consider the existence of arcs if there exists a timelike geodesic in $\mathcal{V}_{\mathbf{x}}$ joining two events $\mathbf{x}$ and $\mathbf{y}$. See Appendix C for details.

### B.4 PROJECTIVE SPACE $\mathbb{P}^d_1(r)$

The manifold $\mathbb{P}^d_1(r) := \mathbb{H}^d_1(r)/\pm 1$ used in Law (2021) is time-orientable for all $d \geq 2$ (see page 214 of O'Neill (1983)). We refer the reader to (Law, 2021) for details about $\mathbb{P}^d_1(r) := \mathbb{H}^d_1(r)/\pm 1$.

## C FUTURE DIRECTION, TIME SEPARATION FUNCTION AND LORENTZIAN DISTANCE

### C.1 FUTURE DIRECTION

We discuss in this section how to constrain future direction for our spacetimes.

#### C.1.1 MINKOWSKI SPACE

The explanation can be found in Section 3.2 when $\mathcal{M} = \mathbb{R}^d_1$. We recall that we defined: $\mathbf{t} := (1, 0, 0, 0)^\top$, $\overrightarrow{\mathbf{xy}} = \mathbf{y} - \mathbf{x}$, $\langle \overrightarrow{\mathbf{xy}}, \overrightarrow{\mathbf{xy}} \rangle = \langle \overrightarrow{\mathbf{xy}}, \overrightarrow{\mathbf{xy}} \rangle_1$ and $\alpha := -(y_0 - x_0) + \sqrt{\sum_{i=1}^{d-1}(y_i - x_i)^2}$.

If $\alpha$ is negative, then $\overrightarrow{\mathbf{xy}}$ is timelike (i.e., $\langle \overrightarrow{\mathbf{xy}}, \overrightarrow{\mathbf{xy}} \rangle < 0$) and $\overrightarrow{\mathbf{xy}} \in \mathcal{C}^+_{\mathbf{x}}(\mathbf{t})$. By definition, $\overrightarrow{\mathbf{xy}}$ is then future-directed timelike.

### C.1.2   DE SITTER SPACE

We provide the proof of Lemma 3.1 which is written: when $\mathcal{M} = \mathbb{S}^d_1(r)$ and $\overrightarrow{\mathbf{xy}}$ is timelike, $\overrightarrow{\mathbf{xy}}$ is future-directed iff $\Gamma^{\mathbf{x}}_{\mathbf{p}}(\mathbf{t}) \in \mathcal{C}^+_{\mathbf{x}}(\overrightarrow{\mathbf{xy}})$ if $\Gamma^{\mathbf{x}}_{\mathbf{p}}$ is defined (i.e., $\mathbf{p} \in \mathcal{U}_{\mathbf{x}}$), and $\Gamma^{\mathbf{x}}_{-\mathbf{p}}(\mathbf{t}) \in \mathcal{C}^+_{\mathbf{x}}(\overrightarrow{\mathbf{xy}})$ otherwise (i.e., $-\mathbf{p} \in \mathcal{U}_{\mathbf{x}}$).

We first recall a property from page 146 of (O'Neill, 1983): If $\mathcal{M}$ is a Lorentz manifold, if a piecewise smooth curve $\alpha$ is timelike, this means not only that $\alpha'(t)$ is timelike, but at each break $t_i$ of $\alpha$:

$$\langle \alpha'(t_i^-), \alpha'(t_i^+) \rangle < 0. \tag{21}$$

Here the first vector derives from $\alpha|[t_{i-1}, t_i]$ and the second from $\alpha|[t_i, t_{i+1}]$. Thus $\alpha'$ does not switch timecones at the break.

Let us note the poles $\mathbf{p} = (0, \ldots, 0, r)^\top \in \mathbb{S}^d_1(r)$, $-\mathbf{p} = (0, \ldots, 0, -r)^\top \in \mathbb{S}^d_1(r)$ and $\mathbf{t} = (1, 0 \ldots, 0)^\top$ where $\mathbf{t} \in T_{\mathbf{p}}\mathbb{S}^d_1(r)$ and $\mathbf{t} \in T_{-\mathbf{p}}\mathbb{S}^d_1(r)$. We also note $\mathbf{x} := (x_0, x_1, \ldots, x_d)^\top \in \mathbb{S}^d_1(r)$.

If $x_d \in (-r, r)$, we show below that equation 21 can be satisfied by defining $\alpha$ such that $\alpha(t_0) = \mathbf{p}$, $\alpha(t_1) = \mathbf{x}$, $\alpha(t_2) = -\mathbf{p}$, $\alpha'(t_0) = \mathbf{t}$, $\alpha'(t_1^-) = \Gamma^{\mathbf{x}}_{\mathbf{p}}(\mathbf{t})$, $\alpha'(t_1^+) = \Gamma^{\mathbf{x}}_{-\mathbf{p}}(\mathbf{t})$, $\alpha'(t_2) = \mathbf{t}$. For completeness, we also have $\forall t \in [t_0, t_1^-], \alpha'(t) = \Gamma^{\alpha(t)}_{\mathbf{p}}(\mathbf{t})$ and $\forall t \in [t_1^+, t_2], \alpha'(t) = \Gamma^{\alpha(t)}_{-\mathbf{p}}(\mathbf{t})$.

Let us arbitrarily consider that $\mathbf{t} \in T_{\mathbf{p}}\mathbb{S}^d_1(r)$ defines the future direction of the manifold. For all $\mathbf{x} \in \mathbb{S}^d_1(r)$ that satisfies $\langle \mathbf{x}, \mathbf{p} \rangle_1 > -r^2$ (i.e., $\mathbf{p} \in \mathcal{U}_{\mathbf{x}}$), we note $\Gamma^{\mathbf{x}}_{\mathbf{p}}(\mathbf{t})$ the parallel translate of $\mathbf{t} \in T_{\mathbf{p}}\mathbb{S}^d_1(r)$ to $T_{\mathbf{x}}\mathbb{S}^d_1(r)$. As explained in Appendix B.1.2 of (Law, 2021), it is formulated:

$$\Gamma^{\mathbf{x}}_{\mathbf{p}}(\mathbf{t}) := \mathbf{t} - \frac{\langle \mathbf{x}, \mathbf{t} \rangle_1}{\langle \mathbf{x}, \mathbf{p} \rangle_1 + r^2}(\mathbf{p} + \mathbf{x}) = \mathbf{t} + \frac{x_0}{r x_d + r^2}(\mathbf{p} + \mathbf{x}) \tag{22}$$

We know from page 66 of (O'Neill, 1983) that parallel transport (also called parallel translation) is a linear isometry that satisfies:

$$\forall \mathbf{u} \in T_{\mathbf{x}}\mathbb{S}^d_1(r), \forall \mathbf{v} \in T_{\mathbf{x}}\mathbb{S}^d_1(r), \langle \mathbf{u}, \mathbf{v} \rangle = \langle \Gamma^{\mathbf{p}}_{\mathbf{x}}(\mathbf{u}), \Gamma^{\mathbf{p}}_{\mathbf{x}}(\mathbf{v}) \rangle = \langle \Gamma^{\mathbf{p}}_{\mathbf{x}}(\mathbf{u}), \Gamma^{\mathbf{p}}_{\mathbf{x}}(\mathbf{v}) \rangle_1 \tag{23}$$

and $\mathbf{u} = \Gamma^{\mathbf{x}}_{\mathbf{p}}(\Gamma^{\mathbf{p}}_{\mathbf{x}}(\mathbf{u}))$. This implies

$$\Gamma^{\mathbf{x}}_{\mathbf{p}}(\mathbf{t}) \in \mathcal{C}^+_{\mathbf{x}}(\mathbf{u}) \iff \Gamma^{\mathbf{p}}_{\mathbf{x}}(\mathbf{u}) \in \mathcal{C}^+_{\mathbf{p}}(\mathbf{t}) \tag{24}$$

Similarly, if $\langle \mathbf{x}, -\mathbf{p} \rangle_1 > -r^2$ (i.e., $-\mathbf{p} \in \mathcal{U}_{\mathbf{x}}$), we have $\Gamma^{\mathbf{x}}_{-\mathbf{p}}(\mathbf{t}) \in \mathcal{C}^+_{\mathbf{x}}(\mathbf{u}) \iff \Gamma^{-\mathbf{p}}_{\mathbf{x}}(\mathbf{u}) \in \mathcal{C}^+_{-\mathbf{p}}(\mathbf{t})$.

Let us assume that $\mathbf{x}$ satisfies both $\mathbf{p} \in \mathcal{U}_{\mathbf{x}}$ (i.e., $\langle \mathbf{x}, \mathbf{p} \rangle_1 > -r^2$) and $-\mathbf{p} \in \mathcal{U}_{\mathbf{x}}$ (i.e., $\langle \mathbf{x}, \mathbf{p} \rangle_1 < r^2$), this is equivalent to $\mathbf{x}$ satisfying $x_d \in (-r, r)$.

If $x_d \in (-r, r)$, then $\Gamma^{\mathbf{x}}_{\mathbf{p}}(\mathbf{t}) \in \mathcal{C}^+_{\mathbf{x}}(\mathbf{u}), \Gamma^{\mathbf{x}}_{-\mathbf{p}}(\mathbf{t}) \in \mathcal{C}^+_{\mathbf{x}}(\mathbf{u}) \iff \langle \Gamma^{\mathbf{x}}_{\mathbf{p}}(\mathbf{t}), \Gamma^{\mathbf{x}}_{-\mathbf{p}}(\mathbf{t}) \rangle_1 < 0$. By definition, we have:

$$\Gamma^{\mathbf{x}}_{-\mathbf{p}}(\mathbf{t}) := \mathbf{t} - \frac{\langle \mathbf{x}, \mathbf{t} \rangle_1}{\langle \mathbf{x}, -\mathbf{p} \rangle_1 + r^2}(-\mathbf{p} + \mathbf{x}) = \mathbf{t} + \frac{x_0}{-r x_d + r^2}(-\mathbf{p} + \mathbf{x}) \tag{25}$$

$$\langle \Gamma^{\mathbf{x}}_{\mathbf{p}}(\mathbf{t}), \Gamma^{\mathbf{x}}_{-\mathbf{p}}(\mathbf{t}) \rangle_1 = -1 - \frac{x_0^2}{r x_d + r^2} - \frac{x_0^2}{-r x_d + r^2} = -1 - 2\frac{x_0^2}{(x_d + r)(-x_d + r)} < 0 \tag{26}$$

We have shown that $\Gamma^{\mathbf{x}}_{\mathbf{p}}(\mathbf{t})$ and $\Gamma^{\mathbf{x}}_{-\mathbf{p}}(\mathbf{t})$ have the same future direction by showing that $\langle \Gamma^{\mathbf{x}}_{\mathbf{p}}(\mathbf{t}), \Gamma^{\mathbf{x}}_{-\mathbf{p}}(\mathbf{t}) \rangle_1 < 0$. A (piecewise) smooth curve that preserves causality can then be found by using $\Gamma^{\mathbf{x}}_{\mathbf{p}}(\mathbf{t})$ or $\Gamma^{\mathbf{x}}_{-\mathbf{p}}(\mathbf{t})$. $\qquad\square$

### C.1.3   ANTI-DE SITTER SPACE

When $\mathcal{M} = \mathbb{H}^d_1(r)$, $\overrightarrow{\mathbf{xy}}$ is future-directed iff $\Gamma^{\mathbf{x}}_{\mathbf{p}}(\mathbf{t}) \in \mathcal{C}^+_{\mathbf{x}}(\overrightarrow{\mathbf{xy}})$ if $\Gamma^{\mathbf{x}}_{\mathbf{p}}$ is defined (i.e., $\mathbf{p} \in \mathcal{U}_{\mathbf{x}}$), and $\Gamma^{\mathbf{x}}_{-\mathbf{p}}(-\mathbf{t}) \in \mathcal{C}^+_{\mathbf{x}}(\overrightarrow{\mathbf{xy}})$ otherwise (i.e., $-\mathbf{p} \in \mathcal{U}_{\mathbf{x}}$). The proof is similar to the one in Section C.1.2.

Let us note the poles $\mathbf{p} = (r, 0, \ldots, 0)^\top \in \mathbb{H}^d_1(r)$, $-\mathbf{p} = (-r, 0, \ldots, 0)^\top \in \mathbb{H}^d_1(r)$ and $\mathbf{t} = (0, 1, 0 \ldots, 0)^\top$ where $\mathbf{t} \in T_{\mathbf{p}}\mathbb{H}^d_1(r)$ and $\mathbf{t} \in T_{-\mathbf{p}}\mathbb{H}^d_1(r)$. Let $\mathbf{x} = (x_{-1}, x_0, \ldots, x_{d-1})^\top \in \mathbb{H}^d_1(r)$. We recall that $\mathbf{p} \in \mathcal{U}_{\mathbf{x}}$ iff $\langle \mathbf{x}, \mathbf{p} \rangle_2 < r^2$ (i.e., $x_{-1} > -r$) and $-\mathbf{p} \in \mathcal{U}_{\mathbf{x}}$ iff $\langle \mathbf{x}, -\mathbf{p} \rangle_2 < r^2$ (i.e., $x_{-1} < r$). We define $\mathbf{x}$ such that $x_{-1} \in (-r, r)$ (i.e., $\mathbf{p} \in \mathcal{U}_{\mathbf{x}}$ and $-\mathbf{p} \in \mathcal{U}_{\mathbf{x}}$).

Let us note $\Gamma_{\mathbf{p}}^{\mathbf{x}}(\mathbf{t})$ the parallel translate of $\mathbf{t} \in T_{\mathbf{p}}\mathbb{H}_1^d(r)$ to $T_{\mathbf{x}}\mathbb{H}_1^d(r)$. As explained in AppendixB.3 of (Law, 2021), it is formulated:

$$\Gamma_{\mathbf{p}}^{\mathbf{x}}(\mathbf{t}) := \mathbf{t} - \frac{\langle \mathbf{x}, \mathbf{t} \rangle_2}{\langle \mathbf{x}, \mathbf{p} \rangle_2 - r^2}(\mathbf{p} + \mathbf{x}) = \mathbf{t} + \frac{x_0}{-rx_{-1} - r^2}(\mathbf{p} + \mathbf{x}) \tag{27}$$

We show below that $\Gamma_{\mathbf{p}}^{\mathbf{x}}(\mathbf{t})$ and $\Gamma_{-\mathbf{p}}^{\mathbf{x}}(-\mathbf{t})$ have the same future direction by showing that $\langle \Gamma_{\mathbf{p}}^{\mathbf{x}}(\mathbf{t}), \Gamma_{-\mathbf{p}}^{\mathbf{x}}(-\mathbf{t}) \rangle_2 < 0$.

$$\Gamma_{-\mathbf{p}}^{\mathbf{x}}(-\mathbf{t}) := -\mathbf{t} - \frac{\langle \mathbf{x}, -\mathbf{t} \rangle_2}{\langle \mathbf{x}, -\mathbf{p} \rangle_2 - r^2}(-\mathbf{p} + \mathbf{x}) = -\mathbf{t} - \frac{x_0}{rx_{-1} - r^2}(-\mathbf{p} + \mathbf{x}) \tag{28}$$

$$\langle \Gamma_{\mathbf{p}}^{\mathbf{x}}(\mathbf{t}), \Gamma_{-\mathbf{p}}^{\mathbf{x}}(-\mathbf{t}) \rangle_2 = 1 + \frac{x_0^2}{rx_{-1} - r^2} + \frac{x_0^2}{-rx_{-1} - r^2} = 1 - 2\frac{x_0^2}{r^2 - x_{-1}^2} \tag{29}$$

$$= \frac{r^2 - x_{-1}^2 - 2x_0^2}{r^2 - x_{-1}^2} < 0. \tag{30}$$

equation 30 is negative because we have by definition of $\mathbb{H}_1^d(r)$: $-x_{-1}^2 - x_0^2 + \sum_{i=1}^{d-1} x_i^2 = -r^2$, which implies $x_0^2 \geq r^2 - x_{-1}^2 > 0$, and the denominator is positive because we defined $x_{-1} \in (-r, r)$. $\square$

## C.2 TIME SEPARATION FUNCTION

We give the formulations we use for the time separation function $\tau(\mathbf{x}, \mathbf{y})$ that is positive if $\mathbf{y}$ is in the chronological future of $\mathbf{x}$. There is no unique way to define it. We assume here that $\nu = 1$.

**Globally hyperbolic spacetimes.** If the spacetime $\mathcal{M}$ is globally hyperbolic, there exists a Cauchy time function $c : \mathcal{M} \to \mathbb{R}$ (i.e., for any $t \in \mathbb{R}$, the set $\Sigma_t = \{\mathbf{y} \in \mathcal{M} : c(\mathbf{y}) = t\}$ is a Cauchy hypersurface) that can be used to define $\tau(\mathbf{x}, \mathbf{y}) := c(\mathbf{y}) - c(\mathbf{x})$ since it satisfies the reverse triangle inequality $\tau(\mathbf{x}, \mathbf{z}) \geq \tau(\mathbf{x}, \mathbf{y}) + \tau(\mathbf{y}, \mathbf{z})$ when $\mathbf{x} \leq \mathbf{y} \leq \mathbf{z}$. With this formulation, it actually satisfies $\tau(\mathbf{x}, \mathbf{z}) = \tau(\mathbf{x}, \mathbf{y}) + \tau(\mathbf{y}, \mathbf{z})$ when $\mathbf{x} \leq \mathbf{y} \leq \mathbf{z}$.

In theory, to have a Lorentzian pre-length space, one could define $\tau$ such that it is $0$ if $\mathbf{x} \not\leq \mathbf{y}$, but this is not easy to optimize in practice so we ignore this constraint for optimization purpose.

### C.2.1 MINKOWSKI SPACE $\mathbb{R}_1^d$

If $\mathcal{M} = \mathbb{R}_1^d$ then we arbitrarily define:

$$\tau(\mathbf{x}, \mathbf{y}) := -\langle \overrightarrow{\mathbf{x}\mathbf{y}}, \mathbf{t} \rangle = -\langle \overrightarrow{\mathbf{x}\mathbf{y}}, \mathbf{t} \rangle_1 = y_0 - x_0 \tag{31}$$

where $\mathbf{t} := (1, 0, \dots, 0)^\top$ defines the future timecone $\mathcal{C}_{\mathbf{x}}^+(\mathbf{t})$. It is worth noting that the function $c : \mathbb{R}_1^d \to \mathbb{R}$ defined such that $\forall \mathbf{x} = (x_0, \dots, x_{d-1})^\top \in \mathbb{R}_1^d, c(\mathbf{x}) := x_0$ is a Cauchy time function.

### C.2.2 DE SITTER SPACE $\mathbb{S}_1^d(r)$

Following our discussion in SectionC.1.2 and using the formulation of the parallel transport in equation 22, we can formulate the time function as:

$$\tau(\mathbf{x}, \mathbf{y}) := \begin{cases} -\langle \Gamma_{\mathbf{x}}^{\mathbf{p}}(\overrightarrow{\mathbf{x}\mathbf{y}}), \mathbf{t} \rangle_\nu & \text{if } \langle \mathbf{x}, \mathbf{p} \rangle_\nu \geq 0 \\ -\langle \Gamma_{\mathbf{x}}^{-\mathbf{p}}(\overrightarrow{\mathbf{x}\mathbf{y}}), \mathbf{t} \rangle_\nu & \text{otherwise.} \end{cases} \tag{32}$$

where $\nu = 1$. One limitation of equation 32 is that it assumes that $\overrightarrow{\mathbf{x}\mathbf{y}}$ is defined, which might not be the case (if $\mathbf{y} \notin \mathcal{U}_{\mathbf{x}}$).

From the discussion in Chapter 5.2 and Figure 16 (i) of (Hawking & Ellis, 1973) that uses Cartesian coordinates to formulate space coordinates on the pseudo-sphere, the function $c : \mathbb{S}_1^d(r) \to \mathbb{R}$ defined such that $\forall \mathbf{x} = (x_0, \dots, x_d)^\top, c(\mathbf{x}) := x_0$ is a Cauchy time function. One can then also simply define:

$$\tau(\mathbf{x}, \mathbf{y}) := y_0 - x_0. \tag{33}$$

In practice, we found that using equation 33 returns better performance than equation 32 in the experiments of Section D.1. We then also use it for the experiment of Section 5.1. See Section D.2 for details.

### C.2.3 ANTI-DE SITTER SPACE $\mathbb{H}_1^d(r)$

When $\nu > 0$, $\mathbb{H}_\nu^d(r)$ is non-chronological and satisfies $\mathbf{x} \ll \mathbf{y} \implies \mathbf{y} \ll \mathbf{x}$, which is convenient to represent graphs with directed cycles. There exists a future-directed timelike geodesic from $\mathbf{x}$ to $\mathbf{y}$ only if $\overrightarrow{\mathbf{xy}}$ is timelike or $\mathbf{y} = \pm\mathbf{x}$.

Following our discussion in Section C.1.3 and using the formulation of the parallel transport in equation 27, we can formulate the time function as:

$$\tau(\mathbf{x}, \mathbf{y}) := \begin{cases} -\langle \Gamma_{\mathbf{x}}^{\mathbf{P}}(\overrightarrow{\mathbf{xy}}), \mathbf{t} \rangle_{\nu+1} & \text{if } \langle \mathbf{x}, \mathbf{p} \rangle_{\nu+1} \leq 0 \\ -\langle \Gamma_{\mathbf{x}}^{-\mathbf{P}}(\overrightarrow{\mathbf{xy}}), -\mathbf{t} \rangle_{\nu+1} & \text{otherwise.} \end{cases} \tag{34}$$

### C.2.4 PROJECTIVE VERSION OF THE ANTI-DE SITTER SPACE $\mathbb{P}_1^d(r)$

We recall that each point of $\mathbb{P}_1^d(r)$ is an unordered pair $\{-\mathbf{x}, \mathbf{x}\} \in \mathbb{P}_1^d(r)$ where $\mathbf{x} \in \mathbb{H}_1^d(r)$. By using the formulation of the parallel transport in equation 27 and assuming $\nu = 1$, we can then define:

$$\tau(\{-\mathbf{x}, \mathbf{x}\}, \{-\mathbf{y}, \mathbf{y}\}) := \begin{cases} -\langle \Gamma_{\mathbf{x}}^{\mathbf{P}}(\overrightarrow{\mathbf{xy}}), \mathbf{t} \rangle_{\nu+1} & \text{if } \langle \mathbf{x}, \mathbf{p} \rangle_{\nu+1} \leq 0 \text{ and } \langle \mathbf{x}, \mathbf{y} \rangle_{\nu+1} \leq 0 \\ -\langle \Gamma_{\mathbf{x}}^{-\mathbf{P}}(\overrightarrow{\mathbf{xy}}), -\mathbf{t} \rangle_{\nu+1} & \text{if } \langle \mathbf{x}, \mathbf{p} \rangle_{\nu+1} > 0 \text{ and } \langle \mathbf{x}, \mathbf{y} \rangle_{\nu+1} \leq 0 \\ -\langle \Gamma_{\mathbf{x}}^{\mathbf{P}}(\overrightarrow{\mathbf{x}(-\mathbf{y})}), \mathbf{t} \rangle_{\nu+1} & \text{if } \langle \mathbf{x}, \mathbf{p} \rangle_{\nu+1} \leq 0 \text{ and } \langle \mathbf{x}, \mathbf{y} \rangle_{\nu+1} > 0 \\ -\langle \Gamma_{\mathbf{x}}^{-\mathbf{P}}(\overrightarrow{\mathbf{x}(-\mathbf{y})}), -\mathbf{t} \rangle_{\nu+1} & \text{if } \langle \mathbf{x}, \mathbf{p} \rangle_{\nu+1} > 0 \text{ and } \langle \mathbf{x}, \mathbf{y} \rangle_{\nu+1} > 0 \end{cases} \tag{35}$$

Let us assume that $\mathbf{p} = (r, 0, \ldots, 0)^\top$. We can choose $\mathbf{x}$ such that $x_{-1} \geq 0$ so that $\langle \mathbf{x}, \mathbf{p} \rangle_2 \leq 0$ is satisfied, and we can also choose $\mathbf{y}$ so that $\langle \mathbf{x}, \mathbf{y} \rangle_2 \leq 0$ is satisfied. We can define $\mathbf{t} := (0, 1, 0, \ldots, 0)^\top \in T_{\mathbf{p}} \mathbb{P}_1^d(r)$. If we define $\mathbf{u} = (u_{-1}, u_0, \ldots, u_{d-1})^\top = \Gamma_{\mathbf{x}}^{\mathbf{P}}(\overrightarrow{\mathbf{xy}})$, then equation 35 can be rewritten:

$$\tau(\{-\mathbf{x}, \mathbf{x}\}, \{-\mathbf{y}, \mathbf{y}\}) = u_0 \tag{36}$$

Another possibility is to use the following time separation function:

$$\tau(\{-\mathbf{x}, \mathbf{x}\}, \{-\mathbf{y}, \mathbf{y}\}) = u_0 - \sqrt{\sum_{i=1}^{d-1} u_i^2} \tag{37}$$

which is positive only if $\overrightarrow{\mathbf{xy}}$ is future-directed timelike.

### C.2.5 CYLINDRICAL MINKOWSKI SPACE $\mathbb{L}_1^d(C)$

As we explain in the main paper, we propose to define the chronological future $\mathcal{I}^+(\mathbf{x}, \mathcal{V}_{\mathbf{x}})$ of $\mathbf{x} \in \mathbb{L}_1^d(C)$ such that if $\overrightarrow{\mathbf{xy}}$ is timelike, we have $\mathbf{y} \in \mathcal{I}^+(\mathbf{x}, \mathcal{V}_{\mathbf{x}})$ if $\exists k \in \mathbb{Z}, y_0 + kC \in (x_0, x_0 + C/2)$. Similarly, the chronological past $\mathcal{I}^-(\mathbf{x}, \mathcal{V}_{\mathbf{x}})$ is defined such that if $\overrightarrow{\mathbf{xy}}$ is timelike, we have $\mathbf{y} \in \mathcal{I}^-(\mathbf{x}, \mathcal{V}_{\mathbf{x}})$ if $\exists k \in \mathbb{Z}, y_0 + kC \in (x_0 - C/2, x_0)$.

We define the time separation function for $\mathbb{L}_1^d(C)$ as follows:

$$\tau(\mathbf{x}, \mathbf{y}) := \left( \left( (y_0 - x_0 + \frac{C}{2}) \bmod C \right) - \frac{C}{2} \right) \in [-\frac{C}{2}, \frac{C}{2}) \tag{38}$$

where we use the modulo operation for real values which can be written as follows: $a \bmod b := a - b \cdot \lfloor \frac{a}{b} \rfloor$, and $\lfloor \cdot \rfloor$ is the floor function.

### C.3 SQUARED LORENTZIAN DISTANCE

We give the formulation of the squared Lorentzian distance for the different spacetimes that we use in the main paper depending on the nature of $\mathcal{M}$:

$$\text{If } \mathcal{M} = \mathbb{R}_1^d, \ \chi_{\mathcal{U}}^2(\mathbf{x}, \mathbf{y}) := -\langle \overrightarrow{\mathbf{xy}}, \overrightarrow{\mathbf{xy}} \rangle = -\langle \overrightarrow{\mathbf{xy}}, \overrightarrow{\mathbf{xy}} \rangle_1 := (y_0 - x_0)^2 - \sum_{j=1}^{d-1} (y_j - x_j)^2. \tag{39}$$

$$\text{If } \mathcal{M} = \mathbb{S}_1^d(r), \; \chi_{\mathcal{U}}^2(\mathbf{x}, \mathbf{y}) := \begin{cases} -\langle \overrightarrow{\mathbf{xy}}, \overrightarrow{\mathbf{xy}} \rangle = r^2 \operatorname{arccosh}^2\left(\frac{\langle \mathbf{x}, \mathbf{y} \rangle_1}{r^2}\right) & \text{if } \frac{\langle \mathbf{x}, \mathbf{y} \rangle_1}{r^2} \geq 1 \\ 2(\langle \mathbf{x}, \mathbf{y} \rangle_1 - r^2) & \text{otherwise.} \end{cases} \quad (40)$$

$$\text{If } \mathcal{M} = \mathbb{P}_1^d(r), \; \chi_{\mathcal{U}}^2(\mathbf{x}, \mathbf{y}) := \begin{cases} -\langle \overrightarrow{\mathbf{xy}}, \overrightarrow{\mathbf{xy}} \rangle = r^2 \arccos^2\left(\frac{|\langle \mathbf{x}, \mathbf{y} \rangle_2|}{r^2}\right) & \text{if } \frac{\langle \mathbf{x}, \mathbf{y} \rangle_2}{r^2} \in [-1, 1] \\ -2(|\langle \mathbf{x}, \mathbf{y} \rangle_2| - r^2) & \text{otherwise.} \end{cases} \quad (41)$$

$$\text{If } \mathcal{M} = \mathbb{L}_1^d(C), \; \chi_{\mathcal{U}}^2(\mathbf{x}, \mathbf{y}) := \left( \left( (y_0 - x_0 + \frac{C}{2}) \bmod C \right) - \frac{C}{2} \right)^2 - \sum_{j=1}^{d-1} (y_j - x_j)^2. \quad (42)$$

The above equations are equal to 0 when $\mathbf{x}$ is equivalent to $\mathbf{y}$, which is the case if $\mathbf{y} = \mathbf{x}$ in general, or if $\mathbf{y} = \pm\mathbf{x}$ when $\mathcal{M} = \mathbb{P}_1^d(r)$, or if $\mathbf{y} \sim \mathbf{x}$ when $\mathcal{M} = \mathbb{L}_1^d(C)$. Otherwise, the equations are positive iff there exists at least one timelike geodesic from $\mathbf{x}$ to $\mathbf{y}$. If the timelike geodesic from $\mathbf{x}$ to $\mathbf{y} \neq \mathbf{x}$ is uniquely defined in $\mathcal{U}_{\mathbf{x}}$, we call it $\overrightarrow{\mathbf{xy}}$, and these equations are positive whether $\overrightarrow{\mathbf{xy}}$ is future-directed timelike or past-directed timelike.

**Squared (Lorentzian) distance in Sim et al. (2021).** Sim et al. (2021) acknowledge that there exists no geodesic from $\mathbf{x}$ to $\mathbf{y}$ if $\mathcal{M} = \mathbb{H}_1^d(r)$ and $\frac{\langle \mathbf{x}, \mathbf{y} \rangle_2}{r^2} > 1$. Therefore, they consider that their squared Lorentzian distance is equal to $\pi^2$ in this case to keep their function smooth. This promotes causality between $\mathbf{x}$ and $\mathbf{y}$. However, constraining future-direction between $\mathbf{x}$ and $\mathbf{y}$ when $\frac{\langle \mathbf{x}, \mathbf{y} \rangle_2}{r^2} > 1$ becomes problematic without the explicit formulation of a timelike curve. This is why they formulate their time function based on a different criterion than ours that uses parallel transport (see Section C.2). Moreover, the fact that their distance function becomes a constant in this case makes it difficult to optimize as the gradient is zero. Instead, we propose to use the manifold $\mathcal{M} = \mathbb{P}_1^d(r)$ which can represent the same types of relationships between points as $\mathbb{H}_1^d(r)$ since $\mathbb{P}_1^d(r)$ contains elliptic and hyperbolic parts (Law, 2021), and any pair of points of $\mathbb{P}_1^d(r)$ can be joined by a geodesic.

### C.4 ABOUT THE CHOICE OF $\varepsilon$ IN EQUATION 4

We explain here why our formulation of equation 4 is general and can be extended to other spacetimes.

As explained in Theorem 2.7 of Minguzzi (2019), an open globally hyperbolic convex normal neighborhood $\mathcal{V}_{\mathbf{x}}$ can be defined for any point $\mathbf{x} \in \mathcal{M}$ where $\mathcal{M}$ is a spacetime. As we mention in Section 3, any open subset of a spacetime is a spacetime. Therefore, $\mathcal{V}_{\mathbf{x}}$ is a spacetime that can be chosen to be globally hyperbolic. We recall that a globally hyperbolic spacetime is strongly causal, and that the chronological future $\mathcal{I}^+(\mathbf{x}, \mathcal{V}_{\mathbf{x}})$ is the set of points $\mathbf{y} \in \mathcal{V}_{\mathbf{x}}$ that satisfy $\mathbf{x} \ll \mathbf{y}$ (i.e., there exists a timelike curve from $\mathbf{x}$ to $\mathbf{y}$).

Since we define $\mathcal{V}_{\mathbf{x}}$ to be a convex normal neighborhood, we know from Proposition 4.5.3 of Hawking & Ellis (1973) that any pair of its points $\mathbf{x}, \mathbf{y}$ that satisfy $\mathbf{x} \ll \mathbf{y}$ can be joined by a unique future-directed timelike geodesic $\gamma_{\mathbf{x} \to \overrightarrow{\mathbf{xy}}}$, which is also the unique longest curve joining $\mathbf{x}$ to $\mathbf{y}$. Since $\gamma_{\mathbf{x} \to \overrightarrow{\mathbf{xy}}}$ is future-directed timelike, it satisfies $\langle \overrightarrow{\mathbf{xy}}, \overrightarrow{\mathbf{xy}} \rangle < 0$ and $\langle \overrightarrow{\mathbf{xy}}, \mathbf{t} \rangle < 0$ (i.e., $\overrightarrow{\mathbf{xy}} \in \mathcal{C}_{\mathbf{x}}^+(\mathbf{t})$) where the timelike tangent vector $\mathbf{t} \in T_{\mathbf{x}}\mathcal{M}$ defines the future direction of $\mathcal{M}$. Moreover, the arc length of the geodesic $\gamma_{\mathbf{x} \to \overrightarrow{\mathbf{xy}}}$ from $\mathbf{x}$ to $\mathbf{y}$ is $\sqrt{-\langle \overrightarrow{\mathbf{xy}}, \overrightarrow{\mathbf{xy}} \rangle}$, and is called its Lorentzian distance.

From Theorem 5.6 and Lemma 5.7 of Minguzzi (2019), there exists a strongly causal open set containing $\mathbf{x}$ such that its Lorentzian distance with any other point $\mathbf{y}$ in that set has its Lorentzian distance $\sqrt{-\langle \overrightarrow{\mathbf{xy}}, \overrightarrow{\mathbf{xy}} \rangle}$ upper bounded by some constant $\varepsilon > 0$. We can then define that open set to be the convex normal neighborhood $\mathcal{V}_{\mathbf{x}}$ and we can choose $\varepsilon > 0$ such that all the points $\mathbf{y} \in \mathcal{V}_{\mathbf{x}}$ that satisfy $\mathbf{x} \ll \mathbf{y}$ also satisfy $-\varepsilon^2 < \langle \overrightarrow{\mathbf{xy}}, \overrightarrow{\mathbf{xy}} \rangle < 0$.

Since $\mathcal{V}_{\mathbf{x}}$ is a subset of the maximal normal neighborhood $\mathcal{U}_{\mathbf{x}}$, we then obtain exactly the definition of equation 4, which is: $\mathcal{I}^+(\mathbf{x}, \mathcal{V}_{\mathbf{x}}) = \{\mathbf{y} \in \mathcal{U}_{\mathbf{x}} : -\varepsilon^2 < \langle \overrightarrow{\mathbf{xy}}, \overrightarrow{\mathbf{xy}} \rangle < 0, \overrightarrow{\mathbf{xy}} \in \mathcal{C}_{\mathbf{x}}^+(\mathbf{t})\}$.

In our experiments, we consider that $\mathcal{V}_{\mathbf{x}} = \mathcal{U}_{\mathbf{x}}$. When $\mathcal{M}$ is the Minkowski space $\mathbb{R}_1^d$ or the de Sitter space $\mathbb{S}_1^d(r)$, we then have $\varepsilon = +\infty$ but we describe some case where $\varepsilon$ might be finite in Section 3.2. Similarly, we have $\varepsilon = r\pi/2$ when $\mathcal{M} = \mathbb{P}_1^d(r)$, and $\varepsilon = C/2$ when $\mathcal{M} = \mathbb{L}_1^d(C)$.

Table 3: Preservation of chronological order between pairs of articles with one citing the other.

| $|V|$ | $\mathbb{R}^2_1$ | $\mathbb{R}^5_1$ | $\mathbb{R}^9_1$ | $\mathbb{P}^2_1(r)$ | $\mathbb{P}^5_1(r)$ | $\mathbb{P}^9_1(r)$ |
|---|---|---|---|---|---|---|
| $\tau$ | equation 31 | equation 31 | equation 31 | equation 35 | equation 35 | equation 35 |
| 200 | 93.7% | 94.0% | 94.2% | 73.3% | 73.7% | 74.9% |
| 1000 | 91.5% | 91.8% | 92.0% | 72.2% | 72.4% | 72.3% |

| $|V|$ | $\mathbb{S}^3_1(r)$ | $\mathbb{S}^5_1(r)$ | $\mathbb{S}^9_1(r)$ | $\mathbb{S}^3_1(r)$ | $\mathbb{S}^5_1(r)$ | $\mathbb{S}^9_1(r)$ |
|---|---|---|---|---|---|---|
| $\tau$ | equation 32 | equation 32 | equation 32 | equation 33 | equation 33 | equation 33 |
| 200 | 93.8% | 93.8% | 93.7% | 94.4% | 94.9% | 94.9% |
| 1000 | 89.1% | 89.2% | 89.5% | 92.8% | 93.4% | 93.6% |

# D  ADDITIONAL EXPERIMENTS AND EXPERIMENTAL DETAILS

We now report additional experiments and provide experimental details.

**Setup.** We ran all our experiments on a single desktop with 64 GB of RAM, a 6-core Intel i7-7800X CPU and a NVIDIA GeForce RTX 3090 GPU.

## D.1  CHRONOLOGICAL ORDER IN DIRECTED GRAPHS

Our goal in this subsection is to represent a directed graph with spacetimes. As in Clough & Evans (2017), we select the 200 and 1000 most cited papers in the Arxiv *High-energy physics theory* (HEP-TH) citation network (Gehrke et al., 2003). HEP-TH is originally a dataset of $27,770$ papers (each represented by a node) with $352,807$ edges. The graph contains an arc from $v_i$ to $v_j$ if paper $i$ cites paper $j$.

When selecting the 200 or 1000 most cited papers, the graph is not a Directed Acyclic Graph (DAG) as there exist pairs of papers that cite each other. We ignore these pairs of arcs. To simplify the notation, we write $v_i \ll v_j$ either if there exists a path from $v_i$ to $v_j$, or the exists an arc from $v_i$ to $v_j$ but not from $v_j$ to $v_i$ (i.e., there can exist a longer path from $v_j$ to $v_i$). We also write $v_a \not\ll v_b$ if there exists no path from $v_a$ to $v_b$ or from $v_b$ to $v_a$ (i.e., $v_a \not\ll v_b \iff v_b \not\ll v_a$).

We optimize the problem:

$$\min_{\{\mathbf{x}_k \in \mathcal{M}\}^n_{k=1}} \sum_{v_a \not\ll v_b} \sigma_{\theta_1} \left( \chi^2_\mathcal{U}(\mathbf{x}_a, \mathbf{x}_b) \right) + \lambda \sum_{v_i \ll v_j} \left( \sigma_{\theta_1} \left( -\chi^2_\mathcal{U}(\mathbf{x}_i, \mathbf{x}_j) \right) + \sigma_{\theta_2} \left( -\tau \left( \mathbf{x}_i, \mathbf{x}_j \right) \right) \right) \quad (43)$$

where $\sigma_\theta(x) := 1/(1 + e^{-x/\theta})$ is the sigmoid function, $\theta_1, \theta_2 > 0$ are temperature parameters, $\lambda$ is a regularization parameter and $\tau(\mathbf{x}, \mathbf{y})$ is a time separation function that is positive (resp. negative) if $\mathbf{y}$ is in the chronological future (resp. past) of $\mathbf{x}$. For instance, if $\mathcal{M} = \mathbb{R}^d_1$ then we arbitrarily define $\tau(\mathbf{x}, \mathbf{y}) := -\langle \overrightarrow{\mathbf{xy}}, \mathbf{t} \rangle = y_0 - x_0$, where $\mathbf{t} := (1, 0, \ldots, 0)^\top$ defines the future timecone $\mathcal{C}^+_\mathbf{x}(\mathbf{t})$.

In this experiment, our temperature hyperparameter values are $\theta_1 = \theta_2 = 1$, and we fix the radius to $r = 1$. We run our experiments for $10^8$ iterations with a step size of $10^{-6}$ by using the optimization tools of (Law, 2021; Law & Stam, 2020). The regularization parameter $\lambda$ is set to $\lambda = \frac{|E^c|}{|E|}$ where $|E|$ is the number of pairs that satisfy $v_i \ll v_j$ and $|E^c|$ is the number of pairs that satisfy $v_a \not\ll v_b$.

We report in Table 3 how well the learned representations manage to preserve chronological order when we select the $|V| = 200$ or 1000 most cited papers. For instance if $\mathcal{M} = \mathbb{R}^d_1$, we report the percentage of pairs of nodes $v_i \ll v_j$ represented by $\mathbf{x}_i$ and $\mathbf{x}_j$ that satisfy $\langle \overrightarrow{\mathbf{x}_i \mathbf{x}_j}, \mathbf{t} \rangle < 0$. The chronological manifolds $\mathbb{R}^d_1$ and $\mathbb{S}^d_1(r)$ manage to predict chronological order better than the non-chronological manifold $\mathbb{P}^d_1(r)$. This suggests that chronological spacetimes are more appropriate to represent graphs that are almost DAGs. The time separation function in equation 33 returns better performance than equation 32, we then use it in the rest of our experiments.

Figure 3 illustrates the embeddings learned when $\mathcal{M} = \mathbb{R}^2_1$. Article representations tend to satisfy the chronological order along the time coordinate.

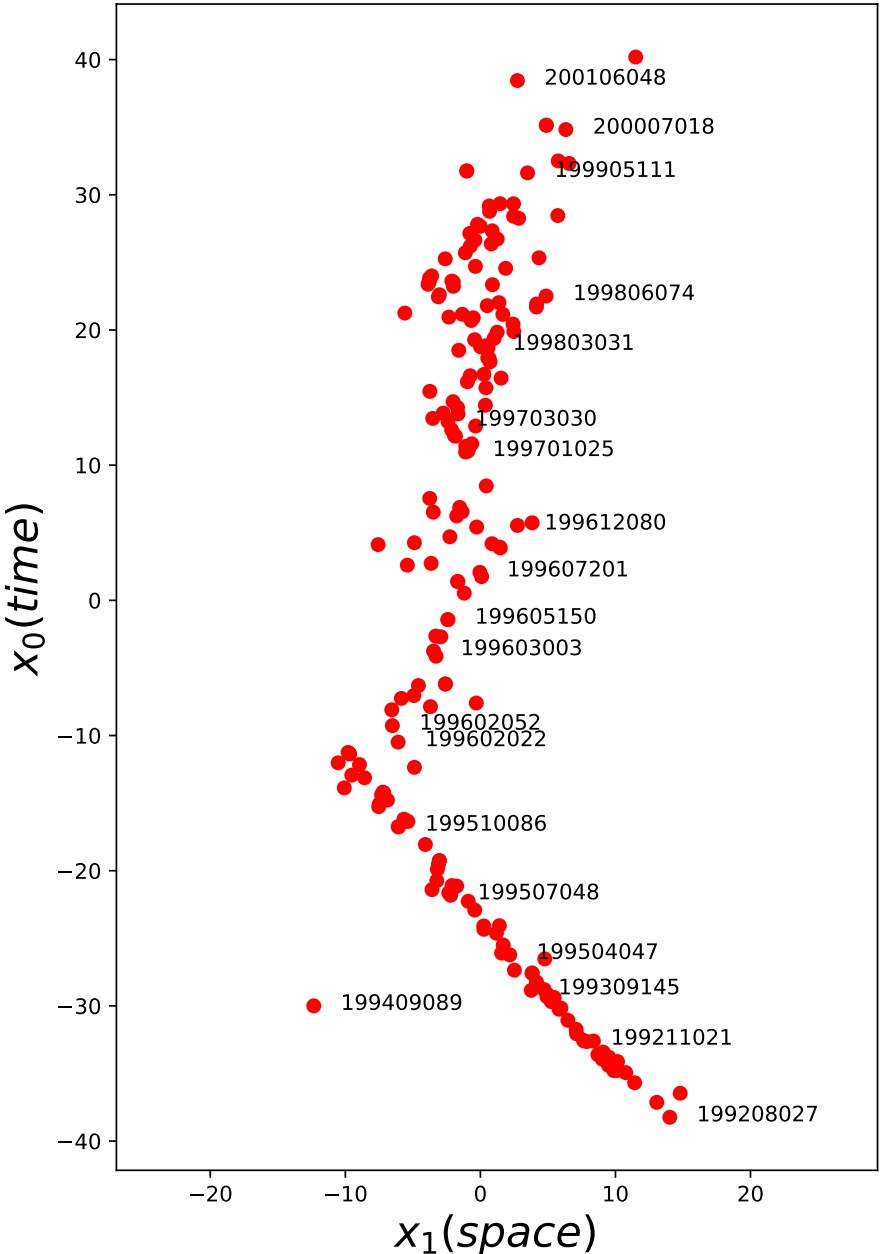

Figure 3: 2-dimensional representations of the Arxiv *High-energy physics theory* (HEP-TH) citation network (Gehrke et al., 2003) when $|V| = 200$ and $\mathcal{M} = \mathbb{R}^2_1$. The first four characters of the labels correspond to the year of submission on Arxiv, and the following two characters are the month of publication. For instance, the label "199208027" corresponds to the article submitted in August 1992 available at: `https://arxiv.org/abs/hep-th/9208027`. Similarly, "199409089" is the article available at `https://arxiv.org/abs/hep-th/9409089` and submitted in September 1994. The embeddings more or less follow the chronological order (along the time axis) in which the articles were submitted to Arxiv.

## D.2 Directed link prediction

In this section, we provide details about the experiments in Section 5.1.

We report in Table 4 the link prediction scores on the *Saccharomyces cerevisiae*, *in silico* and *Escherichia coli* DREAM5 datasets (Marbach et al., 2012). Following the evaluation protocol of (Sim et al., 2021), we report the median + standard deviation across 20 random initialization. We use the same training and test splits as (Sim et al., 2021).

We use the following hyperparameters on the DREAM5 datasets to define equation 6:

• if $\mathcal{M} = \mathbb{S}_1^d(r)$, $r = 1$, $\theta_1 = 0.15$, exponent $m = 1$, $\theta_2 = 0.03$, learning rate $= 10^{-5}$, number of epochs $= 2000$.

• if $\mathcal{M} = \mathbb{L}_1^d(C)$, $C = 8$, $\theta_1 = 0.15$, $\theta_2 = 0.03$, exponent $m = 1$, learning rate $= 10^{-3}$, number of epochs $= 2000$.

• if $\mathcal{M} = \mathbb{R}_1^d$, $\theta_1 = 0.15$, $\theta_2 = 0.03$, exponent $m = 1$, learning rate $= 10^{-3}$, number of epochs $= 2000$.

When $d = 10, 50$ or $100$, $\mathbb{S}_1^d(r)$ is not time-oriented (see explanation in Section 3.2). We report the scores obtained in these dimensionalities to be fair with baselines.

We also reran the experiments of Minkowski + TFD on the Escherichia coli DREAM 5 dataset.

It is worth noting that the DREAM5 datasets contain a relatively small number of cycles:

• The Saccharomyces cerevisiae DREAM 5 dataset contains 9 nodes that are part of at least one directed cycle (0.5% of 1,994 nodes) and 19 edges that are part of at least one directed cycle (0.5% of 3,940).

• The Escherichia coli DREAM 5 dataset contains 18 nodes that are part of at least one directed cycle (1.7% of 1,081 nodes) and 23 edges that are part of at least one directed cycle (1.1% of 2,066).

• The in silico DREAM 5 dataset contains 28 nodes that are part of at least one directed cycle (1.8% of 1,565 nodes) and 39 edges that are part of at least one directed cycle (1.0% of 4,012).

The choice of a specific manifold acts as some inductive bias. When the manifold is chosen to be chronological, the created graph does not necessarily contain directed cycles but allows their existence. On the other hand, chronological manifolds are more appropriate for Directed Acyclic Graphs as they ensure that the created graph does not contain directed cycles. From our results, it seems that the (nonchronological) Cylindrical Minkowski spacetime obtains much better performance in the low-dimensional case as its lack of causality allows some flexibility that is less beneficial in the high-dimensional case.

Sim et al. (2021) also use the synthetic "Duplication-Divergence"(Dupdiv) dataset (Ispolatov et al., 2005), but their dataset contains only 100 edges and 1,026 edges. We generated a bigger version of Dupdiv that contains 1,000 edges and 26,649 edges (22,651 for training/validation and 3,998 for test) following the same setup. More precisely, 748 nodes (74.8%) are part of at least one directed cycle and 22,409 edges are part of at least one directed cycle (84.1% of 26,649). Sim et al. (2021) obtain their best performance on their Dupdiv dataset with the Cylindrical Minkowski + Triple Fermi-Dirac (TFD). We compare it on our larger dataset with Cylindrical Minkowski + equation 6. The results are reported in Table 4 and show a consistent gain of 2% mean Average Precision by using a proper time separation function.

We use the following hyperparameters for Dupdiv: $\mathcal{M} = \mathbb{R}_1^d$, $\theta_1 = 0.4$, $\theta_2 = 0.07$, exponent $m = 0.5$, learning rate $= 0.02$, number of epochs $= 500$.

Table 4: Link prediction for directed graphs. Median average precision (AP) percentages across 20 random initializations on a held-out test set.

| Dataset | DREAM 5: Saccharomyces cerevisiae | | | | |
|---|---|---|---|---|---|
| Manifold dimensionality $d$ | 3 | 5 | 10 | 50 | 100 |
| Euclidean + FD | $33.0 \pm 2.7$ | $34.2 \pm 2.8$ | $40.2 \pm 3.3$ | $44.5 \pm 3.5$ | $49.0 \pm 2.0$ |
| Hyperboloid + FD | $29.2 \pm 2.5$ | $37.9 \pm 1.3$ | $46.5 \pm 1.6$ | $48.8 \pm 1.4$ | $47.9 \pm 1.2$ |
| Minkowski + TFD | $34.7 \pm 2.2$ | $38.6 \pm 1.9$ | $46.4 \pm 3.1$ | $52.7 \pm 3.0$ | $54.0 \pm 2.5$ |
| Anti de-Sitter + TFD | $37.2 \pm 3.2$ | $41.3 \pm 1.5$ | $44.9 \pm 2.5$ | $47.5 \pm 3.1$ | $49.4 \pm 3.3$ |
| Cylindrical Minkowski + TFD | $37.4 \pm 3.2$ | $42.7 \pm 2.3$ | $46.8 \pm 3.5$ | $53.4 \pm 2.2$ | $54.6 \pm 2.1$ |
| Minkowski + equation 6 | $47.6 \pm 1.1$ | $51.3 \pm 1.5$ | $54.4 \pm 1.1$ | $54.7 \pm 2.0$ | $54.8 \pm 1.3$ |
| Cylindrical Minkowski + equation 6 | $\mathbf{50.0 \pm 1.7}$ | $\mathbf{52.5 \pm 1.4}$ | $55.2 \pm 1.5$ | $\mathbf{56.2 \pm 1.4}$ | $\mathbf{55.7 \pm 1.7}$ |
| de Sitter + equation 6 | $44.8 \pm 2.1$ | $51.6 \pm 1.6$ | $\mathbf{55.6 \pm 1.3}$ | $55.3 \pm 1.4$ | $55.4 \pm 1.4$ |

| Dataset | DREAM5 : in silico | | | | |
|---|---|---|---|---|---|
| Manifold dimensionality $d$ | 3 | 5 | 10 | 50 | 100 |
| Euclidean + FD | $29.4 \pm 2.1$ | $32.9 \pm 2.5$ | $39.7 \pm 1.8$ | $39.8 \pm 1.6$ | $34.8 \pm 1.1$ |
| Hyperboloid + FD | $28.8 \pm 5.5$ | $46.8 \pm 4.6$ | $50.8 \pm 7.4$ | $50.9 \pm 1.5$ | $52.5 \pm 1.5$ |
| Minkowski + TFD | $36.3 \pm 2.3$ | $43.1 \pm 3.1$ | $51.2 \pm 3.0$ | $57.7 \pm 2.8$ | $58.0 \pm 2.7$ |
| Anti de-Sitter + TFD | $38.1 \pm 4.8$ | $45.2 \pm 2.3$ | $51.9 \pm 5.2$ | $55.6 \pm 4.2$ | $56.0 \pm 3.4$ |
| Cylindrical Minkowski + TFD | $41.0 \pm 3.6$ | $48.4 \pm 7.3$ | $56.3 \pm 8.4$ | $58.9 \pm 2.9$ | $61.0 \pm 1.9$ |
| Minkowski + equation 6 | $48.4 \pm 1.2$ | $49.4 \pm 1.1$ | $51.6 \pm 1.2$ | $58.1 \pm 2.1$ | $58.8 \pm 1.1$ |
| Cylindrical Minkowski + equation 6 | $\mathbf{52.5 \pm 1.9}$ | $56.5 \pm 1.6$ | $59.8 \pm 1.5$ | $60.4 \pm 1.5$ | $60.8 \pm 1.3$ |
| de Sitter + equation 6 | $48.5 \pm 1.9$ | $\mathbf{57.4 \pm 1.5}$ | $\mathbf{62.0 \pm 1.4}$ | $\mathbf{60.6 \pm 1.6}$ | $\mathbf{61.1 \pm 1.4}$ |

| Dataset | DREAM5 : Escherichia coli | | | | |
|---|---|---|---|---|---|
| Manifold dimensionality $d$ | 3 | 5 | 10 | 50 | 100 |
| Euclidean + FD | $33.0 \pm 3.9$ | $34.2 \pm 3.4$ | $40.2 \pm 4.3$ | $44.5 \pm 2.6$ | $49.0 \pm 3.2$ |
| Hyperboloid + FD | $43.4 \pm 4.1$ | $47.2 \pm 3.3$ | $52.7 \pm 1.9$ | $53.6 \pm 1.4$ | $50.6 \pm 0.7$ |
| Minkowski + TFD | $43.8 \pm 2.0$ | $50.9 \pm 2.3$ | $57.7 \pm 2.1$ | $58.4 \pm 2.3$ | $58.3 \pm 2.1$ |
| Anti de-Sitter + TFD | $42.7 \pm 3.7$ | $56.5 \pm 2.6$ | $61.8 \pm 6.8$ | $63.3 \pm 4.8$ | $63.0 \pm 7.5$ |
| Cylindrical Minkowski + TFD | $50.3 \pm 3.3$ | $56.8 \pm 3.4$ | $62.3 \pm 3.3$ | $65.8 \pm 3.4$ | $63.2 \pm 2.4$ |
| Minkowski + equation 6 | $55.9 \pm 2.1$ | $57.2 \pm 1.8$ | $58.1 \pm 1.9$ | $58.8 \pm 1.1$ | $59.1 \pm 1.2$ |
| Cylindrical Minkowski + equation 6 | $\mathbf{60.9 \pm 1.8}$ | $\mathbf{64.0 \pm 2.4}$ | $\mathbf{67.5 \pm 2.3}$ | $\mathbf{70.1 \pm 1.4}$ | $\mathbf{70.4 \pm 2.1}$ |
| de Sitter + equation 6 | $58.1 \pm 2.8$ | $62.4 \pm 2.3$ | $62.7 \pm 1.5$ | $63.4 \pm 1.3$ | $62.1 \pm 1.6$ |

| Dataset | Duplication-Divergence (1000 edges) | | | | |
|---|---|---|---|---|---|
| Manifold dimensionality $d$ | 3 | 5 | 10 | 50 | 100 |
| Cylindrical Minkowski + TFD | $55.5 \pm 0.6$ | $64.7 \pm 1.3$ | $69.8 \pm 1.4$ | $70.2 \pm 1.0$ | $70.7 \pm 0.8$ |
| Cylindrical Minkowski + equation 6 | $58.7 \pm 1.3$ | $66.9 \pm 1.1$ | $72.2 \pm 1.1$ | $72.4 \pm 1.2$ | $72.1 \pm 1.0$ |

### D.3 HIERARCHY EXTRACTION

For both equation 7 and equation 8, we set $\theta = 10^{-2}$, $r = 1$ and train the model for $10^5$ iterations/epochs. In equation 8, the regularization parameter $\lambda$ is set to $\lambda = \frac{|E^c|}{|E|}$ where $|E|$ is the number of pairs that satisfy $(v_i, v_j) \in E$ and $|E^c|$ is the number of pairs that satisfy $(v_a, v_b) \notin E$.

We report scores for the NIPS dataset in Table 5. We report the Spearmans rank correlation coefficient $\rho$ (Spearman, 1904) for all the authors (left), the authors with at least 10 coauthors (middle) and authors with at least 20 coauthors (right). Spacetimes return better performance for the subset of authors with at least 10 coauthors.

Table 5: Evaluation scores for the different learned representations (mean $\pm$ standard deviation). $\downarrow$ the lower the metric, the better. $\uparrow$ the larger the metric, the better.

| Manifold | Problem | $d(\mathbf{x}, \mathbf{y})$ | Whole dataset $\rho$ ($\uparrow$) | top $s_i \geq 10$ $\rho$ ($\uparrow$) | Top $s_i \geq 20$ $\rho$ ($\uparrow$) |
|---|---|---|---|---|---|
| $\mathbb{R}^4$ | equation 7 | $d_\gamma(\mathbf{x}, \mathbf{y})$ | 0.469 | 0.512 | 0.217 |
| $\mathbb{P}_0^4(r)$ | equation 7 | $d_\gamma(\mathbf{x}, \mathbf{y})$ | 0.460 | 0.490 | 0.292 |
| $\mathbb{P}_4^4(r)$ | equation 7 | $d_\gamma(\mathbf{x}, \mathbf{y})$ | 0.629 | 0.552 | 0.316 |
| $\mathbb{P}_1^4(r)$ | equation 7 | $d_\gamma(\mathbf{x}, \mathbf{y})$ | **0.667** | 0.493 | 0.307 |
| $\mathbb{P}_2^4(r)$ | equation 7 | $d_\gamma(\mathbf{x}, \mathbf{y})$ | 0.625 | 0.441 | 0.227 |
| $\mathbb{P}_3^4(r)$ | equation 7 | $d_\gamma(\mathbf{x}, \mathbf{y})$ | 0.437 | 0.493 | 0.387 |
| $\mathbb{S}_1^3(r)$ | equation 8 | $\chi_{\mathcal{U}}^2(\mathbf{x}, \mathbf{y})$ | 0.369 | 0.536 | **0.663** |
| $\mathbb{R}_1^4$ | equation 8 | $\chi_{\mathcal{U}}^2(\mathbf{x}, \mathbf{y})$ | 0.524 | **0.668** | 0.484 |
| $\mathbb{P}_1^4(r)$ | equation 8 | $\chi_{\mathcal{U}}^2(\mathbf{x}, \mathbf{y})$ | 0.538 | 0.326 | 0.143 |
| $\mathbb{S}_1^5(r)$ | equation 8 | $\chi_{\mathcal{U}}^2(\mathbf{x}, \mathbf{y})$ | 0.373 | 0.498 | 0.618 |
| $\mathbb{R}_1^6$ | equation 8 | $\chi_{\mathcal{U}}^2(\mathbf{x}, \mathbf{y})$ | 0.478 | 0.678 | 0.543 |
| $\mathbb{P}_1^6(r)$ | equation 8 | $\chi_{\mathcal{U}}^2(\mathbf{x}, \mathbf{y})$ | 0.576 | 0.455 | 0.219 |

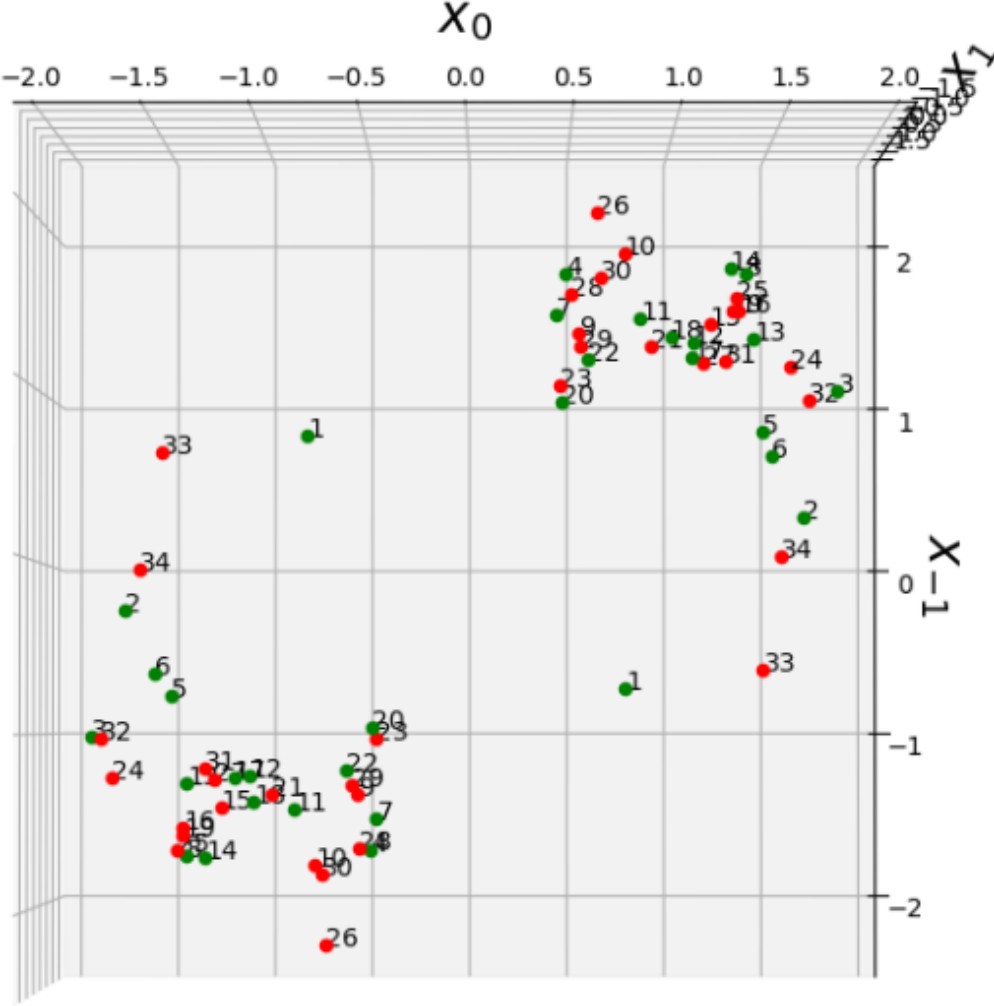

Figure 4: Coordinates of embeddings $\{-\mathbf{x}, \mathbf{x}\} \in \mathbb{P}_1^2(1) \subset \mathbb{R}_2^3$ learned with equation 8.

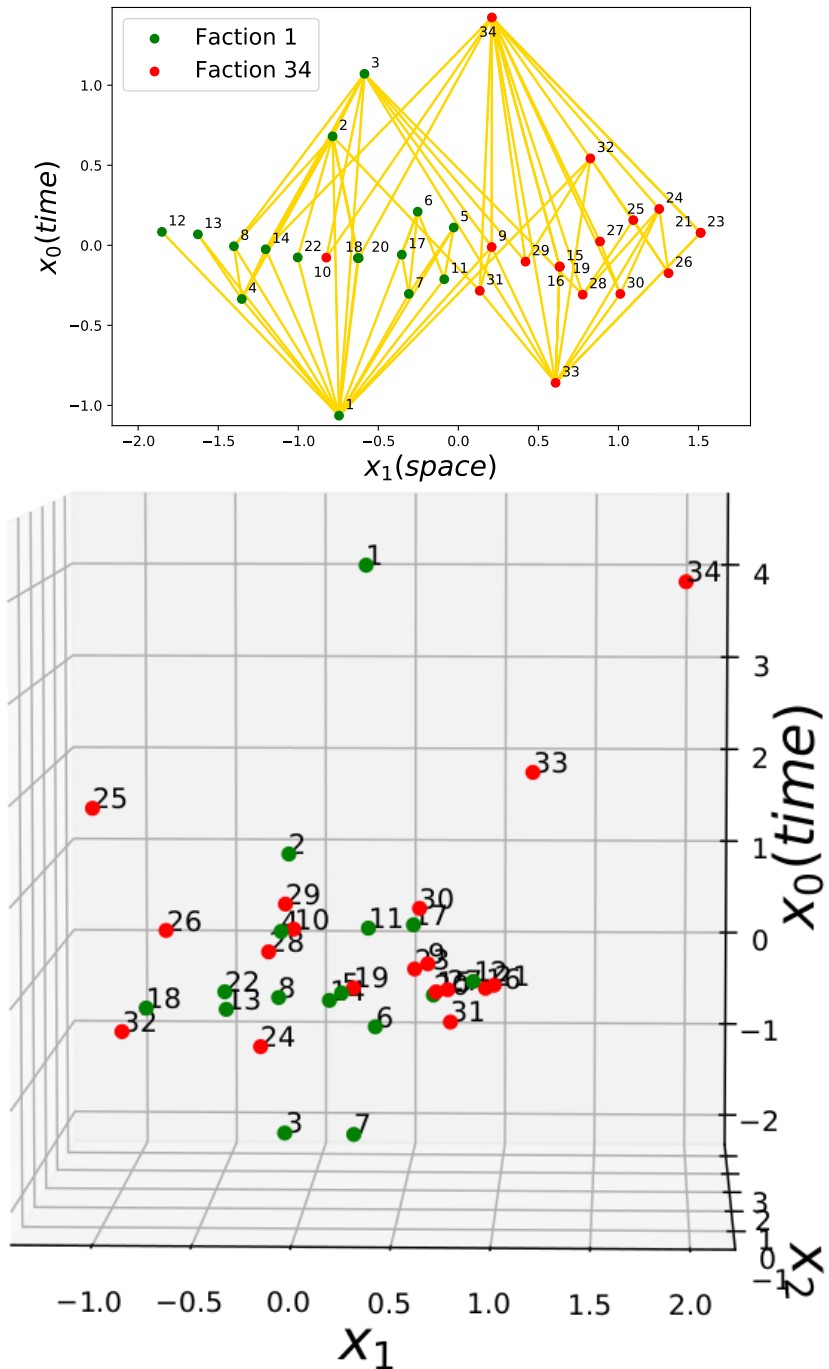

Figure 5: (top) Coordinates of 2-dimensional embeddings $\mathbf{x} = (x_0, x_1)^\top$ learned with equation 8 when $\mathcal{M} = \mathbb{R}^2_1$. (bottom) Coordinates of the first three coordinates of embeddings $\mathbf{x} = (x_0, x_1, x_2, x_3)^\top$ learned with equation 8 when $\mathcal{M} = \mathbb{S}^3_1(r)$. In Lorentz geometry, a time-like geodesic joining two points is the longest timelike curve in a given convex normal neighborhood. This translates in the high-level nodes $v_1$ and $v_{34}$ being the furthest from the rest of the nodes. The ground truth edges are plotted in yellow and the node color corresponds to the joined faction. A small number of spacelike edges are visible (those edges more than 45 degrees from vertical).

# E  EXTENDED RELATED WORK

To help explain the contributions of our work relative to prior art, we use this section to provide a more detailed comparison of our contributions to that of Sim et al. (2021).

Sim et al. (2021) extended Clough & Evans (2017) to the anti-de Sitter space and Lorentz cylinder. Although our motivation is similar, our contributions are methodological, rely on a simpler use of the intuitions of general relativity and Lorentzian pre-length spaces, and provide an easier interpretation of the learned representations as we explain below. First, Sim et al. (2021) do not address clearly the case when there is no geodesic between pairs of points, and their optimization framework leads to a distance loss term with a zero gradient in this case, which make it difficult to optimize. Moreover, in Sim et al. (2021), the prediction of an arc between a pair of nodes is determined via a *Triple Fermi-Dirac* (TFD) probability function that accounts for the squared (Lorentzian) distance between the nodes, the time coordinate difference $\Delta t$ and its opposite value $-\Delta t$. In other words, TFD accounts for both the chronological future and past (with different weights) of a given node. The major methodological difference with Sim et al. (2021) is that we restrict the representation of nodes connected by an arc to belong to $\mathcal{I}^+(\mathbf{x}, \mathcal{V}_\mathbf{x})$ where $\mathcal{V}_\mathbf{x}$ is a convex normal neighborhood. Although subtle, this difference makes the optimization and interpretation of results easier.

In general, the Lorentzian distance function from $\mathbf{x}$ to $\mathbf{y}$ on $\mathcal{M}$ is defined to be infinite (i.e., $\chi_\mathcal{M}(\mathbf{x}, \mathbf{y}) = +\infty$) if $\mathcal{M}$ is non-chronological and there exists a closed timelike curve joining $\mathbf{x}$ and $\mathbf{y}$. Moreover, the Hopf-Rinow theorem does not hold for spacetimes. We propose to work with convex normal neighborhoods, which allows us to restrict the existence of arcs between nodes to the existence of geodesics joining events. Our distance function $\chi_\mathcal{V}$ is called a *local distance function* (see Definition 4.25 of Beem et al. (1996)) when its domain is restricted to a convex normal neighborhood. Moreover, $\chi_\mathcal{V}$ is continuous and differentiable on $\mathcal{V}_\mathbf{x} \times \mathcal{I}^+(\mathbf{x}, \mathcal{V}_\mathbf{x})$ (see Lemma 4.26 of Beem et al. (1996)). In some cases, it might be easier to optimize its squared function $\chi_\mathcal{V}^2$ which is of class $C^2$ on $\mathcal{V}_\mathbf{x} \times \mathcal{V}_\mathbf{x}$ (see Theorem 2.6 of Minguzzi (2019)). It is worth noting that $\mathcal{V}_\mathbf{x}$ can be defined to be globally hyperbolic for any spacetime (see Theorem 2.7 of Minguzzi (2019)). This means that $\mathcal{V}_\mathbf{x}$ admits a Cauchy time function that can be used to define a time separation function $\tau$ (see explanation in Section C.2) whose sign defines the direction of edges. Our framework shares similarities with Sim et al. (2021) when $\mathcal{M} = \mathbb{R}_1^d = \mathcal{V}_\mathbf{x}$ because $\mathbb{R}_1^d$ is globally hyperbolic and any pair of points of $\mathbb{R}_1^d$ can be joined by a geodesic. However, the way we define the time separation $\tau$ (instead of using the same $\Delta t$) is different when $\mathcal{M} = \mathbb{L}_1^d(C)$ because we restrict it to be calculated on the maximal convex normal neighborhood. We construct $\tau$ so that it is positive if $\mathbf{x}_j \in \mathcal{I}^+(\mathbf{x}, \mathcal{V}_\mathbf{x})$ and negative if $\mathbf{x}_j \in \mathcal{I}^-(\mathbf{x}, \mathcal{V}_\mathbf{x})$. We then enforce $\tau$ to be positive if we want an arc from $v_i$ to $v_j$. The fact that we work only with an open convex set instead of the whole manifold $\mathcal{M}$ is crucial because $\mathbf{x}_j$ can belong to both the chronological future $\mathcal{I}^+(\mathbf{x}_i, \mathcal{M}) := \{\mathbf{y} \in \mathcal{M} : \mathbf{x}_i \ll \mathbf{y}\}$ and past $\mathcal{I}^-(\mathbf{x}_i, \mathcal{M}) := \{\mathbf{y} \in \mathcal{M} : \mathbf{y} \ll \mathbf{x}_i\}$ if the $\mathcal{M}$ is non-chronological. Using the entire manifold requires that the sign of $\Delta t$ is not as informative, as in Sim et al. (2021). Our approach allows us to determine the direction of the arc joining $v_i$ and $v_j$ only via the sign of $\tau$.

We also define a general way of optimizing $\tau$ via the parallel transport (see Appendix C.2) instead of working only with Cartesian coordinates, which is not meaningful for some spacetimes such as $\mathbb{P}_1^d(r)$. Moreover, since we restrict our Lorentzian distances to be calculated in the convex normal neighborhood $\mathcal{V}_\mathbf{x}$, we also have the nice interpretation that the Lorentzian distance corresponds to the length of the longest causal curve joining points.

One other contribution is the connection of our work with the theory of Lorentzian pre-length spaces (Kunzinger & Sämann, 2018) which does not require notions of differential geometry to be understood and can be applied to discrete topological spaces (see Example 2.16 of Kunzinger & Sämann (2018)). The framework proposed by Sim et al. (2021) is not a Lorentzian pre-length space due to their formulation of their time coordinate difference function that does not satisfy the properties of a time separation function (especially when $\mathcal{M}$ is non-chronological).

---

**Algorithm 1** Pseudo-Riemannian optimization

---

**input:** differentiable function $f : \mathcal{M} \to \mathbb{R}$ to be minimized, some initial value of $\mathbf{x} \in \mathcal{M}$

---

1: **while** not converge **do**
2:     Calculate $\nabla f(\mathbf{x})$       $\triangleright$ i.e., $\nabla f(\mathbf{x})$ is the Euclidean gradient of $f$ at $\mathbf{x}$ in the Euclidean ambient space
3:     $\chi \leftarrow \Pi_{\mathbf{x}} \left( \mathbf{G} \Pi_{\mathbf{x}} (\mathbf{G} \nabla f(\mathbf{x})) \right)$
4:     $\mathbf{x} \leftarrow \exp_{\mathbf{x}}(-\eta \chi)$                        $\triangleright$ where $\eta > 0$ is a step size
5: **end while**

---

# F   OPTIMIZATION

The optimizers that we use in the main paper for the different spacetimes can all be optimized as described in Algorithm 1. The goal is to minimize some function $f : \mathcal{M} \to \mathbb{R}$ by using differential geometry tools described in (Gao et al., 2018; Law, 2021; Law & Stam, 2020). $\nabla f(\mathbf{x})$ is the Euclidean gradient of $f$ at $\mathbf{x}$, $\Pi_{\mathbf{x}}(\mathbf{z})$ is the orthogonal projection of an arbitrary vector $\mathbf{z}$ onto $T_{\mathbf{x}}\mathcal{M}$, and $\mathbf{G}$ is an involutory matrix (i.e., $\mathbf{G}^{-1} = \mathbf{G}$). We consider in the following that the step size $\eta > 0$ is fixed and given. We give details for the different spacetimes that we consider in the main paper.

## F.1   MINKOWSKI SPACETIME $\mathbb{R}_1^d$

This optimizer was explained in (Gao et al., 2018). When $\mathcal{M} = \mathbb{R}_1^d$, $\Pi_{\mathbf{x}}$ is the identity function. $\mathbf{G}$ is the diagonal matrix with the first diagonal element equal to $-1$ and the remaining ones equal to 1. $\exp_{\mathbf{x}}(\mathbf{y}) := \mathbf{x} + \mathbf{y}$. Algorithm 1 corresponds to the standard Euclidean gradient descent because $\chi = \nabla f(\mathbf{x})$ in this case.

## F.2   CYLINDRICAL MINKOWSKI SPACETIME $\mathbb{L}_1^d(C)$

We recall that $\mathbb{L}_1^d(C) = \mathbb{R}_1^d / \sim$, a quotient set defined such that $\mathbf{x} \in \mathbb{R}_1^d$ and $\mathbf{y} \in \mathbb{R}_1^d$ are equivalent (i.e., $\mathbf{x} \sim \mathbf{y}$) iff $\forall i > 0, y_i = x_i$ and $\exists k \in \mathbb{Z}, y_0 = x_0 + kC$ where $C > 0$ is a circumference hyperparameter.

When $\mathcal{M} = \mathbb{L}_1^d(C)$, we use the same optimizer as in Section F.1.

Although it is optional, we also reproject the time coordinate of $\mathbf{x} = (x_0, \ldots, x_{d-1})^\top$ at the end of each iteration as follows: $x_0 \leftarrow \left( \left( (x_0 + \frac{C}{2}) \bmod C \right) - \frac{C}{2} \right) \in [-\frac{C}{2}, \frac{C}{2})$ where we use the modulo operation for real values which can be written as follows: $a \bmod b := a - b \cdot \lfloor \frac{a}{b} \rfloor$, and $\lfloor \cdot \rfloor$ is the floor function. If the initial value of $x_0$ is not in $[-\frac{C}{2}, \frac{C}{2})$, this projects $\mathbf{x}$ to an equivalent point.

## F.3   DE SITTER SPACE $\mathbb{S}_1^d(r)$

This optimizer was introduced in (Law & Stam, 2020). We recall that $\mathbb{S}_1^d(r) := \{ \mathbf{x} \in \mathbb{R}_1^{d+1} : \langle \mathbf{x}, \mathbf{x} \rangle_1 = r^2 \}$. $\mathbf{G}$ is the diagonal matrix with the first diagonal element equal to $-1$ and the remaining ones equal to 1. We have: $\Pi_{\mathbf{x}}(\mathbf{z}) := \mathbf{z} - \frac{\langle \mathbf{z}, \mathbf{x} \rangle_1}{\langle \mathbf{x}, \mathbf{x} \rangle_1} \mathbf{x} = \mathbf{z} - \frac{\langle \mathbf{z}, \mathbf{x} \rangle_1}{r^2} \mathbf{x}$. The exponential map is defined in equation 10.

## F.4   ANTI-DE SITTER SPACE $\mathbb{H}_1^d(r)$

We recall that $\mathbb{H}_1^d(r) := \{ \mathbf{x} \in \mathbb{R}_2^{d+1} : \langle \mathbf{x}, \mathbf{x} \rangle_2 = -r^2 \}$. $\mathbf{G}$ is the diagonal matrix with the first two diagonal elements equal to $-1$ and the remaining ones equal to 1. We have: $\Pi_{\mathbf{x}}(\mathbf{z}) := \mathbf{z} - \frac{\langle \mathbf{z}, \mathbf{x} \rangle_2}{\langle \mathbf{x}, \mathbf{x} \rangle_2} \mathbf{x} = \mathbf{z} + \frac{\langle \mathbf{z}, \mathbf{x} \rangle_2}{r^2} \mathbf{x}$. The exponential map is defined in equation 15.

## F.5   PROJECTIVE VERSION OF THE ANTI-DE SITTER SPACE $\mathbb{P}_1^d(r)$

The neural network optimizer is given in (Law, 2021). Otherwise, the embedding optimizer is the same as in Section F.4. The main difference is how the points are compared to calculate the distance and time separation function.

## G  EXAMPLES OF GRAPHS

Since our general framework is fairly abstract, we give some explicit examples of directed graphs with or without cycles that can be described by our framework.

### G.1  DIRECTED ACYCLIC GRAPHS (DAGs)

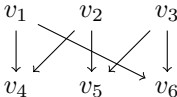

Figure 6: DAG containing an undirected cycle.

We first consider a directed acyclic graph that consists of an undirected cycle (Figure 6). The DAG $G = (V, E)$ is defined as $V = \{v_i\}_{i=1}^6$ and the set of arcs is $E = \{(v_1, v_4), (v_2, v_4), (v_2, v_5), (v_3, v_5), (v_1, v_6), (v_3, v_6)\}$. One can see that $G$ is not a directed tree since the undirected path $v_1, v_4, v_2, v_5, v_3, v_6, v_1$ is cyclic.

We recall that if $\mathcal{M}$ is globally hyperbolic, $\mathcal{M}$ is also chronological and can then only describe DAGs with our framework. Let us give an example where $\mathcal{M} = \mathbb{R}_1^{d+1}$ or $\mathcal{M} = \mathbb{S}_1^d(r) \subset \mathbb{R}_1^{d+1}$ with $d = 3$ and $r > 0$ (e.g., $r = 1$). Both $\mathbb{R}_1^{d+1}$ and $\mathbb{S}_1^d(r)$ are globally hyperbolic. Let us consider any value $\varepsilon > 0$, $a = r + \varepsilon$ and $b = \sqrt{2a^2 - r^2}$. We take the embeddings: $\mathbf{x}_1 = (0, r, 0, 0)^\top$, $\mathbf{x}_2 = (0, 0, r, 0)^\top$, $\mathbf{x}_3 = (0, 0, 0, r)^\top$, $\mathbf{x}_4 = (b, a, a, 0)^\top$, $\mathbf{x}_5 = (b, 0, a, a)^\top$, $\mathbf{x}_6 = (b, a, 0, a)^\top$.

When $\mathcal{M} = \mathbb{S}_1^d(r)$, we know from Section B.2 that $\overrightarrow{\mathbf{x}_i \mathbf{x}_j}$ is timelike (i.e., $\langle \overrightarrow{\mathbf{x}_i \mathbf{x}_j}, \overrightarrow{\mathbf{x}_i \mathbf{x}_j} \rangle < 0$) iff $\langle \mathbf{x}_i, \mathbf{x}_j \rangle_1 > r^2$. By using the time separation function in equation 33 and assuming $\overrightarrow{\mathbf{x}_i \mathbf{x}_j}$ is timelike, we only need to compare the first coordinate of $\mathbf{x}_i$ and $\mathbf{x}_j$ to determine the direction of the edge between $v_i$ and $v_j$.

The case $\mathcal{M} = \mathbb{R}_1^{d+1}$ is similar except that $\overrightarrow{\mathbf{x}_i \mathbf{x}_j} := \mathbf{x}_j - \mathbf{x}_i$ is timelike iff $\langle \overrightarrow{\mathbf{x}_i \mathbf{x}_j}, \overrightarrow{\mathbf{x}_i \mathbf{x}_j} \rangle_1 < 0$.

One can verify that $\forall i, j, (v_i, v_j) \in E \iff \mathbf{x}_j \in \mathcal{U}_{\mathbf{x}_i}$ and the tangent vector $\overrightarrow{\mathbf{x}_i \mathbf{x}_j}$ is future-directed timelike.

### G.2  GRAPHS WITH DIRECTED CYCLES

#### G.2.1  MINKOWSKI CYLINDER $\mathbb{L}_1^d(C)$

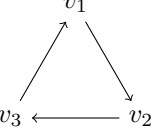

Figure 7: A simple directed cycle.

We now illustrate a simple example of graph with directed cycle that can be drawn with $\mathbb{L}_1^d(C)$ (Figure 7). The graph $G = (V, E)$ is defined as $V = \{v_i\}_{i=1}^3$ and $E = \{(v_1, v_2), (v_2, v_3), (v_3, v_1)\}$, which is a graph with directed cycle.

Let us consider that $C = 3$ and $d = 2$. The maximal normal neighborhood of every point $\mathbf{x} \in \mathbb{R}_1^d$ is $\mathcal{U}_{\mathbf{x}} = \{\mathbf{y} = (y_0, \ldots, y_{d-1})^\top \in \mathbb{R}_1^d : y_0 \in (x_0 - 1.5, x_0 + 1.5)\}$. We also consider that $\mathcal{V}_{\mathbf{x}} = \mathcal{U}_{\mathbf{x}}$.

Since $\mathbb{L}_1^d(C)$ is a quotient set, its points are equivalence classes. We consider three equivalence classes $[\mathbf{x}_i] := \{(i + 3k, 0)^\top : k \in \mathbb{Z}\}$ where $i \in \{1, 2, 3\}$ and we define $\mathbf{x}_i := (i, 0)^\top \in \mathbb{R}_1^d$. In other words, the five points $\mathbf{x}_0 = (0, 0)^\top$, $\mathbf{x}_1 = (1, 0)^\top$, $\mathbf{x}_2 = (2, 0)^\top$, $\mathbf{x}_3 = (3, 0)^\top$, $\mathbf{x}_4 = (4, 0)^\top$ in $\mathbb{R}_1^d$ actually correspond to three points in $\mathbb{L}_1^d(C)$ due to the equivalence relation (i.e., $\mathbf{x}_3 \sim \mathbf{x}_0$ and $\mathbf{x}_4 \sim \mathbf{x}_1$). We can then compare those three equivalence classes by comparing $\mathbf{x}_i$ with $\mathbf{x}_{i-1}$ and $\mathbf{x}_{i+1}$ only. One can verify that $\forall i \in \{1, 2, 3\}$, we have $\mathbf{x}_{i+1} \in \mathcal{U}_{\mathbf{x}_i}$, $\mathbf{x}_{i+1} \in \mathcal{I}^+(\mathbf{x}_i, \mathcal{U}_{\mathbf{x}_i})$, and $\mathbf{x}_{i-1} \in \mathcal{U}_{\mathbf{x}_i}$. However, we also have $\mathbf{x}_{i-1} \notin \mathcal{I}^+(\mathbf{x}_i, \mathcal{U}_{\mathbf{x}_i})$. We then obtain the graph illustrated in Figure 7.

### G.2.2   PROJECTIVE VERSION OF THE ANTI-DE SITTER SPACE $\mathbb{P}_1^d(r)$

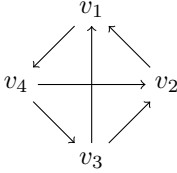

Figure 8: A graph with directed cycles.

We now consider the graph $G = (V, E)$ defined as $V = \{v_i\}_{i=1}^4$ and $E = \{(v_1, v_4), (v_2, v_1), (v_3, v_1), (v_3, v_2), (v_4, v_2), (v_4, v_3)\}$. This is a graph with directed cycles (e.g., $v_1 \to v_4 \to v_2 \to v_1$, see Figure 8).

We recall that every point of $\mathbb{P}_1^d(r)$ can be written as the unordered pair $\{-\mathbf{x}, \mathbf{x}\}$ where $\mathbf{x} \in \mathbb{H}_1^d(r) \subset \mathbb{R}_2^{d+1}$. Taking $d = 2$, the maximal normal neighborhood of every point $\{-\mathbf{x}, \mathbf{x}\}$ can be written $\mathcal{U}_{\{-\mathbf{x}, \mathbf{x}\}} = \{\{-\mathbf{y}, \mathbf{y}\} : \mathbf{y} \in \mathbb{H}_1^d(r), \langle \mathbf{x}, \mathbf{y} \rangle_2 < 0\}$.

Let us consider $r = 1, \varepsilon_1 = -0.1, \varepsilon_2 = \varepsilon_3 = -0.5, a = -1, b = -\sqrt{(r + \varepsilon_1)^2 - r^2 + a^2}, c = -2, e = -\sqrt{(r + \varepsilon_2)^2 - r^2 + c^2}, g = 1, h = \sqrt{(r + \varepsilon_3)^2 - r^2 + g^2}$. We construct the four following points: $\mathbf{x}_1 = \mathbf{p} = (r, 0, 0)^\top, \mathbf{x}_2 = (r + \varepsilon_1, a, b)^\top, \mathbf{x}_3 = (r + \varepsilon_2, c, e)^\top, \mathbf{x}_4 = (r + \varepsilon_3, g, h)^\top$. To define the future direction, we consider the timelike tangent vector $\mathbf{t} = (0, 1, 0)^\top \in T_\mathbf{p}\mathbb{H}_1^d(r)$.

In our framework, there exists an edge between $\mathbf{x}_i$ and $\mathbf{x}_j$ iff $|\langle \mathbf{x}_i, \mathbf{x}_j \rangle_2| \in (0, r^2)$. The direction of the edge is determined by using Section C.2.4.

