# OpenReview forum: "Spacetime Representation Learning"
_ICLR.cc/2023/Conference — ICLR 2023 poster_

### Official Review · Reviewer_WSKC · 2022-10-24

**Confidence:** 3
**Correctness:** 4
**Technical Novelty And Significance:** 3
**Empirical Novelty And Significance:** 3
**Recommendation:** 8

**Clarity, Quality, Novelty And Reproducibility:**

The paper is clear in motivation, model description as well as related mathematical concepts and proofs.

 The paper offers a novel and general framework for directed graph representation learning based on time-orientated Lorentz manifolds and the related concepts and theories are described in high quality.

As for Reproducibility, the authors describe the details of the optimizers they used,  as well as some examples of directed graphs with their model.
Although many mathematical contents are described, it is possible to reproduce the results if following the authors’ ideas carefully.

**Strength And Weaknesses:**

Strength of the paper:
1. Inspired by the Lorentzian causality theory,  the paper proposes a general framework for directed graph representation learning based on pseudo-Riemannian manifolds.
2. The authors make a comprehensive summary of related work and the motivation is clear.
3. Sufficient experiments are conducted to evaluate the effectiveness of the proposed model on related graph datasets.
4. Additive tools are presented to account for time orientation and exploit distances specific to Lorentz geometry.

**Summary Of The Paper:**

As for data represented as a directed graph, in order to explore the causal structure within the manifold,  the authors introduce a general family of representations based on time-oriented Lorentz manifolds.

**Summary Of The Review:**

The paper proposes the model with clear motivation and describes the proposed model clearly. Meanwhile, the experimental results show the proposed model is effective. So it is suited for ICLR.

---

> ### Author Response · Authors · 2022-11-15
> **To Reviewer WSKC**
>
> Thank you for your positive review.

---

### Official Review · Reviewer_SZRp · 2022-10-24

**Confidence:** 3
**Correctness:** 3
**Technical Novelty And Significance:** 3
**Empirical Novelty And Significance:** 3
**Recommendation:** 6

**Clarity, Quality, Novelty And Reproducibility:**

The clarity and reproducibility of this work could be improved. Currently, the definition and method are all mixed up. I think this work is novel as it proposes a new data representation framework.

**Strength And Weaknesses:**

- Strengths:
    1. The authors present the definition of basic concepts in spacetime differential geometry with detailed explanations of the mathematical notations, which helps the readers to better understand the paper.
    2. Experiments on commonly-used graph learning benchmarks show the superiority of the proposed spacetime graph.
- Weakness:
    1. I feel that this work focuses more on the construction of a spacetime graph instead of the representation learning (especially learning) on it.
    2. The method is not presented clearly enough. It simply refers to several previous works and theories for many crucial steps in the proposed method, without enough explanations.
    3. Are the assumptions of Eq. 4 mild enough for the generality of the proposed framework?

**Summary Of The Paper:**

This work proposes a novel data representation framework, namely, the spacetime graph. Both theoretical and experimental analyses are provided to show the effectiveness and superiority of the proposed framework.

**Summary Of The Review:**

The proposed spacetime graph framework is interesting and novel to me. But currently, the writing and organization of the paper could be improved for better readability.

---

> ### Author Response · Authors · 2022-11-15
> **To Reviewer SZRp (1/2)**
>
> $\blacksquare$   I feel that this work focuses more on the construction of a spacetime graph instead of the representation learning (especially learning) on it.
>
> $\bullet$ Our paper focuses on two parts:
>
>   1) The first part is indeed the construction of spacetime graphs (as described in Section 3.1) by constraining each node representation $\textbf{x}$ to be connected to a subset of nodes lying in some convex normal neighborhood $V_{\textbf{x}}$. We explain how our framework fits into the theory of Lorentzian pre-length spaces which are a general framework used to describe directed graphs. In the rest of Section 3, we illustrate differential geometry tools that allow us to define differentiable functions over representations that are consistent with Lorentzian pre-length spaces.
>
> 2)   Once we have introduced the context and motivation of the above differentiable tools, our representations can be learned with pseudo-Riemannian optimization techniques (see Appendix F) and our representation learning scheme then depends on the task that is solved.
>
> 2a)  Our representation learning scheme in the directed link prediction task is explained in Section 5.1. In Eq. (6), pairs of nodes that are supposed connected by an arc are learned so that they are the only points that are joined by a future-directed timelike geodesic.
>
> 2b)  In the undirected graph case, we learn representations so that only pairs joined by an edge are connected by a timelike geodesic as formulated in Eq. (8).
>
> $\blacksquare$  The method is not presented clearly enough. It simply refers to several previous works and theories for many crucial steps in the proposed method, without enough explanations.
>
> $\bullet$ Our method is presented in Eq. (6) and Eq. (8). These equations require the formulation of the squared Lorenzian distance (whose details are given in Appendix C.3) and the time separation function (given in Appendix C.2). We have added a reference to these sections of the appendix in the main paper after Eq. (6) and Eq. (8).
> We motivate the use of these differentiable functions in Section 3 via the framework of Lorentzian pre-length spaces.

---

> > ### Author Response · Authors · 2022-11-15
> > **To Reviewer SZRp (2/2)**
> >
> > $\blacksquare$ Are the assumptions of Eq. 4 mild enough for the generality of the proposed framework?
> >
> > $\bullet$ The short answer is yes.
> >
> > The detailed explanation is as follows. As explained in Theorem 2.7 of Minguzzi (2019), an open globally hyperbolic convex normal neighborhood $V_{\textbf{x}}$ can be defined for any point $\textbf{x} \in \mathcal{M}$ where $\mathcal{M}$ is a spacetime. As we mention in Section 3.1, any open subset of a spacetime is a spacetime. Therefore, $V_{\textbf{x}}$ is a spacetime that can be chosen to be globally hyperbolic. We recall that a globally hyperbolic spacetime is strongly causal, and that the chronological future $\mathcal{I}^+ (\textbf{x}, V_{\textbf{x}})$ is the set of points $\textbf{y} \in V_{\textbf{x}}$ that satisfy $\textbf{x} \ll \textbf{y}$ (i.e., there exists a future-directed timelike curve from $\textbf{x}$ to $\textbf{y}$).
> >
> > Since we define $V_{\textbf{x}}$ to be a convex normal neighborhood, we know from Proposition 4.5.3 of Hawking and Ellis (1973) that any pair of its points $\textbf{x}, \textbf{y}$ that satisfy $\textbf{x} \ll \textbf{y}$ can be joined by a unique future-directed timelike geodesic $\overrightarrow{\textbf{x}\textbf{y}}$, which is also the unique longest such curve joining $\textbf{x}$ to $\textbf{y}$.
> >
> > Since $\overrightarrow{\textbf{x}\textbf{y}}$ is future-directed timelike, it satisfies $\langle \overrightarrow{\textbf{x}\textbf{y}}, \overrightarrow{\textbf{x}\textbf{y}} \rangle < 0$ by definition of timelike geodesics, it also satisfies $\langle \overrightarrow{\textbf{x}\textbf{y}}, \boldsymbol{\mathsf{t}} \rangle < 0$ (i.e., $\overrightarrow{\textbf{x}\textbf{y}} \in \mathcal{C}_{\textbf{x}}^{+}(\boldsymbol{\mathsf{t}})$)
> >
> > where the timelike tangent vector $\boldsymbol{\mathsf{t}} \in T_{\textbf{x}} \mathcal{M}$ defines the future direction of $\mathcal{M}$. Moreover, the arc length of $\overrightarrow{\textbf{x}\textbf{y}}$ is $\sqrt{- \langle \overrightarrow{\textbf{x}\textbf{y}}, \overrightarrow{\textbf{x}\textbf{y}} \rangle}$, and is called its Lorentzian distance.
> >
> > From Theorem 5.6 and Lemma 5.7 of Minguzzi (2019), there exists a strongly causal open set containing $\textbf{x}$ such that its Lorentzian distance with any other point $\textbf{y}$ in that set has its Lorentzian distance $\sqrt{- \langle \overrightarrow{\textbf{x}\textbf{y}}, \overrightarrow{\textbf{x}\textbf{y}} \rangle}$ finite and upper bounded by some constant $\varepsilon > 0$.
> >
> > Since $V_{\textbf{x}}$ is a subset of the maximal normal neighborhood $U_{\textbf{x}}$, there exists some value $\varepsilon > 0$ such that we can define the convex normal neighborhood $V_{\textbf{x}}$ to be a strongly causal open set and all the points $\textbf{y} \in U_{\textbf{x}}$ that satisfy $\textbf{x} \ll \textbf{y}$ and $-\varepsilon^2 < \langle \overrightarrow{\textbf{x}\textbf{y}}, \overrightarrow{\textbf{x}\textbf{y}} \rangle < 0$ belong to $V_{\textbf{x}}$.
> >
> > We then obtain exactly the definition of Eq. (4), which is: $\mathcal{I}^+ (\textbf{x}, V_{\textbf{x}}) = \{ \textbf{y} \in U_{\textbf{x}} : -\varepsilon^2 < \langle \overrightarrow{\textbf{x}\textbf{y}}, \overrightarrow{\textbf{x}\textbf{y}} \rangle < 0, \overrightarrow{\textbf{x}\textbf{y}} \in \mathcal{C}_{\textbf{x}}^{+}(\boldsymbol{\mathsf{t}}) \}$.
> >
> >
> > In our experiments, we consider that $V_{\textbf{x}} = U_{\textbf{x}}$.
> >
> > When $\mathcal{M}$ is the Minkowski space $\mathbb{R}_1^d$ or the de Sitter space $\mathbb{S}^d_1(r)$, we then have $\varepsilon = + \infty$ but we describe some case where $\varepsilon$ might be finite in Section 3.2.
> >
> > Similarly, we have $\varepsilon = r\pi/2$ when $\mathcal{M} = \mathbb{P}^d_{1}(r)$, and $\varepsilon = C/2$  when $\mathcal{M} = \mathbb{L}^d_{1}(C)$.

---

> > > ### Comment · Reviewer_SZRp · 2022-11-25
> > > **Response**
> > >
> > > I appreciate the authors' detailed and thorough explanation. My major concerns have been addressed.

---

### Official Review · Reviewer_bMme · 2022-10-24

**Confidence:** 4
**Correctness:** 4
**Technical Novelty And Significance:** 2
**Empirical Novelty And Significance:** Not applicable
**Recommendation:** 3

**Clarity, Quality, Novelty And Reproducibility:**

This is a competent work which I found readable though not very well written.  Much of it seems to this reviewer like a direct sequel or follow-on to Sim et al (2021).  The most significant innovation I saw was the use of the time separation function in the objective function to distinguish link orientations.  It may be significant that this function breaks Lorentz invariance and singles out a preferred direction, much as is done in the TransE embedding of DAGs (Bordes et al 2013).  It would be interesting to understand the geometry behind this.
If this were the first paper in this direction, I would be more willing to accept it, but a follow-on paper should be more innovative, go into more depth, or both.

**Strength And Weaknesses:**

Strengths: ideas are simple and have some interest, presentation is relatively clear for a reader with a background in the relevant geometry.  Known results on spacetimes developed in the study of general relativity are brought into the discussion.

Weaknesses: the results are incremental, especially compared to Sim et al (2021).  The new ideas are not explored in any depth.  The experiments do not go farther than proof of principle.  Comparison with other work outside of this very specific direction is weak, for example no mention is made of the very popular and standard methods for embedding directed graphs such as TransE (Bordes et al 2013).

**Summary Of The Paper:**

This paper studies embedding of directed graphs into Lorentzian spacetimes.  The idea is very natural as such spacetimes have causal structure, and a pair of points can be embedded with timelike separation if there is an edge and spacelike if there is not.  It was proposed by Clough and Evans (2017) and studied by Sim et al (2021) and others.  The present work generalizes this in various ways: use spacetimes other than Minkowski which can have closed timelike curves and thus represent graphs with directed cycles, and exclude part of the future lightcone to represent nontransitive graphs.  Probably the most significant improvement over previous models is the use of a time separation function in the objective function to distinguish link orientations.  Several experiments are done to choose hyperparameters (spacetime topology, dimension, norm) and provide proof of principle.  It is found that a cylindrical Minkowski space is better than Minkowski space for low dimensional embeddings.

**Summary Of The Review:**

Not a bad work but without sufficient novelty or depth to meet the standards of this conference.

---

> ### Author Response · Authors · 2022-11-15
> **To Reviewer bMme (1/2)**
>
> $\blacksquare$ The results are incremental, especially compared to Sim et al (2021).
>
>    $\bullet$ Please see our response to all reviewers.
>
> $\blacksquare$ The new ideas are not explored in any depth.
>
>
> $\bullet$ We disagree with this statement. Unlike Sim et al. (2021), our framework is properly endowed with the structure of a Lorentzian pre-length space, which generalizes previous frameworks to describe directed graphs through the Lorentzian causality theory (e.g., Bombelli et al. (1987) for causal and chronological spacetimes). We provide the necessary mathematical properties to justify the use of convex normal neighborhoods to draw directed edges, which is not done in Sim et al. (2021). The use of convex normal neighborhoods ensures the existence of geodesics (see Proposition 4.5.1 of (Hawking and Ellis, 1973)) and time separation functions (as explained in Section 3.3) to define Lorentzian pre-length spaces.
> In Section 3.1, we also discuss the type of graph that our framework is able to represent depending on the causality conditions of the chosen spacetime. These directly derive from the causality theory of spacetimes (see Chapter 3 of Beem et al. (1996)).
>
> $\blacksquare$ The experiments do not go farther than proof of principle.
>
> $\bullet$ In terms of experiments, we provide in Section 5.1 directed link prediction results based on standard benchmarks on three genome graphs with directed cycles, which are the largest datasets studied in Sim et al. (2021). These datasets are the Saccharomyces Cerevisiae, in silico and Escherichia
> coli DREAM5 datasets (Marbach et al., Nature 2012). These datasets contain about 1,000 nodes each and from 2,066 to 4,012 edges. In Appendix D.3 , we also provide experiments on an undirected graph with 2,715 nodes and 4,733 edges.
>
> We do provide some proofs of concept by illustrating some examples in Appendix G and learning citation networks in Appendix D.1.
>
> We feel that our empirical evaluation is thorough and comparable to prior art. If you disagree, we would appreciate specific pointers to appropriate datasets and tasks that you recommend we explore --- we would be happy to do so.

---

> > ### Author Response · Authors · 2022-11-15
> > **To Reviewer bMme (2/2)**
> >
> > $\blacksquare$ Comparison with other work outside of this very specific direction is weak, for example no mention is made of the very popular and standard methods for embedding directed graphs such as TransE (Bordes et al., NeurIPS 2013).
> >
> > $\bullet$ We thank the reviewer for raising this as a concern. In the last paragraph of the related work section (Section 4), we mention approaches that have considered using Riemannian geometry to Directed Acyclic Graphs (DAGs). We explain in the following how TransE (Bordes et al., NeurIPS 2013) falls into that family of approaches and how our motivations are different. We have added a citation to TransE in Section 4.
> >
> >
> > Given a single relation $R$, TransE cannot represent graphs with directed cycles. The goal of TransE is to learn embeddings as follows: given some asymmetric (i.e., irreflexive and antisymmetric) relation $R$ over some set $\mathcal{E}$, the relation $\textbf{h}R\textbf{t}$ implies that the distance between $\textbf{h} + \textbf{r}$ and $\textbf{t}$ is smaller than (1) the distance between $\textbf{h} + \textbf{r}$ and any other point $\textbf{t}' \in \mathcal{E}$ that does not satisfy $\textbf{h}R\textbf{t}'$ and (2) the distance between $\textbf{t}$ and $\textbf{h}' + \textbf{r}$ where $\textbf{h}' \in \mathcal{E}$ is any other points  that does not satisfy $\textbf{h}'R\textbf{t}$.
> > Following the notation of TransE (Bordes et al., NeurIPS 2013), we then write $\textbf{h} + \textbf{r} \approx \textbf{t}$. We assume in the following that an arc is drawn from $\textbf{h}$ to $\textbf{t}$ if $\textbf{h} + \textbf{r} \approx \textbf{t}$.
> >
> > Whether the embeddings are $\ell_2$-normalized after each iteration or not, there cannot exist a sequence that satisfies:
> >
> > $\textbf{h} + \textbf{r} \approx \textbf{t}_1$
> >
> > $\textbf{t}_1 + \textbf{r} \approx \textbf{t}_2$
> >
> > $\dots$
> >
> > $\textbf{t}_{k}  + \textbf{r} \approx \textbf{h}$.
> >
> > Therefore, TransE cannot represent graphs with directed cycles with a single relation $R$. It can also be shown the formed DAG is a directed tree as its underlying undirected graph does not contain cycles.
> >
> > Nonetheless, assuming an arc is drawn if $\textbf{h} + \textbf{r} \approx \textbf{t}$, TransE is able to contain cycles by considering multiple types of relations (e.g., one relation could represent "is parent of" and another would be "is child of", which would result in directed cycles between any pair of nodes). We mention in Section 4 how to adapt our approach to multiple relations (i.e., multiple types of arcs) but this is not the goal of our paper, and this would require the different types of relations to be provided during training.
> >
> > Our goal is to learn node representations of directed graphs based on a partial set of edge relations provided during training (or equivalently a partial adjacency matrix).
> > In our submission, we denote our relation by $\ll$, and our method can describe different types of relations depending on the choice of manifold. For instance, non-chronological spacetimes (e.g., the Cylindrical Minkowski spacetime) can describe reflexive relations (i.e., $\textbf{x} \ll \textbf{x}$) hence graphs with directed cycles. We give such examples in Section G.2 and we evaluate our approach in the directed link prediction task on graphs with directed cycles in Section 5.1.
> > Moreover, globally hyperbolic spacetimes can describe strict partial orders (i.e. irreflexive, antisymmetric, and transitive), hence DAGs that can contain undirected cycles (see example in Section G.1). In Section 5, our algorithm is provided with a set of pairs of nodes that should or should not be connected by an arc, and our approach learns node representations that follow the geometry of spacetimes, especially the Lorentzian causality theory.
> > Our framework can handle different types of relations and is well-defined for any spacetime.

---

> > > ### Comment · Reviewer_bMme · 2022-11-25
> > > **Reply to authors**
> > >
> > > Thanks to the authors for replying to my review.
> > > I find the revised version of the paper is better, and I now agree that the experiments go beyond "proof of principle".
> > > I also see the authors' point that their method can handle directed graphs with cycles,
> > > and appreciate the evidence that performance is improved by using spacetimes with closed timelike loops, as in table 1 and 4.
> > >
> > > But in practice many directed graphs have only a few cycles.
> > > In Sim et al (2021) section 4.1 it is even stated that "These networks [DREAM5] contain a relatively small number of cycles."
> > > In these cases the inability to represent cycles does not lead to significant degradation.
> > > And the evidence in tables 1 and 4 conflates multiple contrasts.  Especially, in table 4 the improvement in cylindrical Minkowski from the change "TFD -> equation 6" is large, but this is not related to cycles or closed timelike loops.  Why no results for Minkowski and the other spacetimes using the loss in equation 6 or other variations on the loss?
> > > Also it is not clear under their interpretation why the performance improvement would decrease for large embedding dimension.
> > > So, I am not yet convinced that this is what is going on.   More evidence would be welcome, ranging from easy (how many edges would need to be deleted to eliminate the cycles, and are errors associated with cycles) to the additional contrasts requested above.
> > >
> > > Still, in light of the revisions and the other referees' comments I am willing to upgrade to "6: marginally above the acceptance threshold"
> > > if the discussion based on tables 1 and 4 can be clarified.

---

> > > > ### Author Response · Authors · 2022-12-07
> > > > **Answer to new comments**
> > > >
> > > >
> > > > Thank you for your comments.
> > > >
> > > > $\bullet$ The main reason the performance is sometimes lower when the dimensionality is 100 than 50 or 10 is that 15\% of the edges are used for test, and the rest for training and validation. The embeddings that return the best performance on the validation set are then evaluated on the test set and reported over multiple runs. A dimensionality of embeddings that is too large might lead to overfitting. The DREAM5 datasets that we use for evaluation contain from 2,066 to 4,012 edges. Therefore a dimensionality of 10 is sometimes enough to obtain the best performance. It is worth noting that the performance never drops significantly (less than 2\%) and the performance of our 10-dimensional embeddings outperforms baselines of any dimensionality.
> > > >
> > > > $\bullet$  We checked the number of nodes and edges in directed cycles after your comment. The exact numbers are as follows:
> > > >
> > > > - The Saccharomyces Cerevisiae DREAM 5 dataset contains 9 nodes that are part of at least one directed cycle (0.5\% of 1,994 nodes) and 19 edges that are part of at least one directed cycle (0.5\% of 3,940).
> > > >
> > > > - The Escherichia Coli DREAM 5 dataset contains 18 nodes that are part of at least one directed cycle (1.7\% of 1,081 nodes) and 23 edges that are part of at least one directed cycle (1.1\% of 2,066).
> > > >
> > > > - The in Silico DREAM 5 dataset contains 28 nodes that are part of at least one directed cycle (1.8\% of 1,565 nodes) and 39 edges that are part of at least one directed cycle (1.0\% of 4,012).
> > > >
> > > > These numbers are indeed low. Nonetheless, we would like to emphasize that the choice of a specific manifold acts as some inductive bias. When the manifold is chosen to be nonchronological, the created graph does not necessarily contain directed cycles but allows their existence. In other words, nonchronological manifolds allow us to represent both graphs with or without directed cycles. On the other hand, it is not possible to represent directed cycles when the manifold is chosen to be chronological, and the inferred graph can then only be a DAG.
> > > > From our results, it seems that the (nonchronological) Cylindrical Minkowski spacetime obtains much better performance in the low-dimensional case as its lack of causality allows some flexibility that is less beneficial in the high-dimensional case.
> > > >
> > > > Sim et al. (2021) also use the synthetic "Duplication Divergence" (Dupdiv) dataset, but their dataset contains only 100 edges and 1,026 edges. Therefore, we have generated a larger version of Dupdiv that contains 1,000 edges and 26,649 edges (22,651 for training/validation and 3,998 for test) following the same setup. More precisely, 748 nodes (74.8\%) are part of at least one directed cycle and 22,409 edges are part of at least one directed cycle (84.1\% of 26,649).
> > > > Sim et al. obtain their best performance on their Dupdiv dataset with the Cylindrical Minkowski + Triple Fermi-Dirac (TFD). We compare it on our larger dataset with Cylindrical Minkowski + our Equation (6). The results are as follows for the dimensionalities 3, 5, 10, 50, 100 respectively:
> > > >
> > > > Cylindrical Minkowski + TFD --- 55.5 $\pm$ 0.6 --- 64.7 $\pm$ 1.3 --- 69.8 $\pm$ 1.4 --- 70.2 $\pm$ 1.0 --- 70.7  $\pm$ 0.8
> > > >
> > > > Cylindrical Minkowski + Eq.(6) ---  58.7 $\pm$ 1.3 --- 66.9 $\pm$ 1.1 ---  72.2 $\pm$ 1.1 --- 72.4 $\pm$ 1.2 ---  72.1 $\pm$ 1.0
> > > >
> > > > This shows a consistent gain of 2\% mean Average Precision by using a proper time separation function.
> > > >
> > > >
> > > > $\bullet$ We will also add the performance of Minkowski + Eq. (6). Its omission was unintentional and the results are provided below. We report the gain of Minkowski + Eq. (6) over  Minkowski + TFD for the dimensionalities 3, 5, 10, 50, 100 respectively:
> > > >
> > > > Saccharomyces Cerevisiae DREAM 5: 12.9 --- 12.7 --- 8.0 --- 2.0 --- 0.8
> > > >
> > > > Escherichia Coli DREAM 5: 12.1 --- 6.3 --- 0.4 --- 0.4 --- 0.8
> > > >
> > > > in Silico DREAM 5: 2.7 --- 1.8 --- 2.3 --- 2.6 --- 2.8
> > > >
> > > > Enforcing geodesics to future-directed timelike if an arc exists is then more beneficial in the low-dimensional case.

---

### Official Review · Reviewer_cPyY · 2022-10-26

**Confidence:** 3
**Correctness:** 4
**Technical Novelty And Significance:** 3
**Empirical Novelty And Significance:** Not applicable
**Recommendation:** 6

**Clarity, Quality, Novelty And Reproducibility:**

The paper offers a general framework for directed graph representation learning based on time-orientated Lorentz manifolds. However,
in my opinion this paper is very difficult to reproduce.

**Strength And Weaknesses:**

Strength
- The experiments results  show the superiority of the proposed spacetime graph.
- Using  Lorentz manifolds in the context of the paper is very interesting.

 Weaknesses

- I am familliar with Riemannian geometry but I found this paper difficult to read. The writing and organization of the paper could be improved for better readability.
- The results are incremental, especially compared to Sim et al (2021).
- The authors should explain in more details their contributions instead of explications  of basic differential geometry.




**Summary Of The Paper:**

This paper proposes a framework inspired by Laurentzian audacity theory like learn directed graph representation. This framework is evaluted in different tasks and show how this framework can represent graphs with directed cycles.

**Summary Of The Review:**

The proposed spacetime graph framework is interesting. But the writing and organization of the paper make very difficult to understand the novelty and the contributions of the paper.

---

> ### Author Response · Authors · 2022-11-15
> **Riemannian vs Lorentzian geometry**
>
> $\blacksquare$ I am familliar with Riemannian geometry but I found this paper difficult to read. The writing and organization of the paper could be improved for better readability.
>
>    $\bullet$ We have reorganized the paper and Section 3.1 now only contains our methodological contributions. We follow the notation and definitions used in standard general relativity textbooks (i.e.,  Hawking and Ellis (1973), O'Neill (1983), Beem et al. (1996)). It is worth noting that many properties that are crucial in Riemannian geometry, such as the Hopf-Rinow theorem, are not valid in Lorentzian geometry.
>    We are also aware that the machine learning audience is familiar with Riemannian geometry but less so with Lorentzian geometry. This is why we introduce the concepts of Lorentz manifolds: causal character of tangent vectors and curves, future/past timecones, time-orientability, and chronological future. These concepts do not exist in Riemannian geometry, were not all introduced in Sim et al. (2021), and are essential to understand our approach. For instance, we draw an arc between two nodes only if they are joined by a future-directed timelike geodesic. This type of geodesic does not exist in Riemannian geometry, and is not necessarily defined between any pair of points because the Hopf-Rinow theorem does not hold.
>
> $\blacksquare$ The results are incremental, especially compared to Sim et al (2021). The authors should explain in more details their contributions instead of explications of basic differential geometry.
>
> $\bullet$ Please see our response to all reviewers.

---

### Author Response · Authors · 2022-11-15
**To all reviewers**

We thank the reviewers for their positive comments. We recall that the goal of our approach is to exploit differential geometry properties of time-oriented Lorentz manifolds, called "spacetimes", to learn the representation of directed graphs. This has been done previously in the machine learning literature by Clough and Evans (2017) and Sim et al., (ICML 2021).

$\bullet$ One of the common concerns of the reviewers is that "the results are incremental, especially compared to Sim et al (2021)" and we should make the contributions clearer.

Clough and Evans (2017) only consider the Minkowski spacetime which is the simplest spacetime. Sim et al. (2021) extend their framework to other spacetimes. However, when the spacetimes are non-chronological (which is the case for two of the three spacetimes they exploit, the third one being the Minkowski spacetime), their approach does not fit in with the Lorentzian causality theories that describe directed graphs as proposed in Bombelli et al. (1987) and its generalization in Kunzinger and Sämann (2018). In particular, the distance function of Sim et al. (2021) is a constant when there is no geodesic joining pairs of points and their "time coordinate difference" does not correspond to a time separation function, which is an essential tool to define Lorentzian pre-length spaces (Kunzinger and Sämann, 2018).

As a major contribution, we propose to represent directed graphs by considering a union of well-defined Lorentzian pre-length spaces. To this end, we assign to each node representation $\textbf{x}$ some open convex normal neighborhood $V_{\textbf{x}}$. Such a neighborhood can always be defined  (see Theorem 2.7 of Minguzzi (2019)). An arc is then drawn from $\textbf{x}$ to another node $\textbf{y}$ iff $\textbf{y} \in V_{\textbf{x}}$ and the geodesic from $\textbf{x}$ to $\textbf{y}$ is future-directed timelike.


$\bullet$ Another concern of the reviewers is clarity. Following the reviewers' recommendations, we have removed the definitions of causal spaces and Lorentzian pre-length spaces from Section 3.1 to make our contributions clearer. Section 3.1 now only contains our construction of directed graph as described in the paragraph above. We justify the structure of our paper in the following.

Although differential geometry is not necessary to define and understand Lorentzian pre-length spaces, it is essential in our framework to define a differentiable loss function that can be optimized and learn meaningful representations. This is why we dedicate an entire section to general definitions and high-level notation that can be adapted to any spacetime. We provide the exact definitions and low-level details for the different spacetimes that we use in the appendix (see Appendix C.2 and C.3).
Our framework unifies the recent and general framework of Lorentzian pre-length spaces to machine learning, which is not the case of Sim et al. (2021). Our approach can then be applied to a larger family of spacetimes.
We discuss in detail the contributions with respect to Sim et al. (2021) in Section 4 and Appendix E.

---

### Decision · Program_Chairs · 2023-01-20

**Decision:**

Accept: poster

**Justification For Why Not Higher Score:**

the presentation of this paper can be improved

**Justification For Why Not Lower Score:**

the paper makes novel technical contributions

**Metareview: Summary, Strengths And Weaknesses:**

This paper introduces general space-time representations for directed graphs through connected time-oriented Lorentz manifolds. The spacetimes contain a causal structure that indicates causal/chronological order between points on the manifold. The experiments demonstrate the effectiveness of the proposed method.

The proposed method extends Sim et al in a non-trivial way. In particular, the proposed method can handle directed graphs with cycles, which is useful in real applications that have cyclic causal structures. The presentation of the paper needs some improvement, as the paper introduces some mathematical terms that are not often seen in machine learning. Overall, I think it is an interesting paper and recommend acceptance. To make the method more reproducible, I would suggest the authors put the code online after publishing.


**Note From Pc:**

if the above contains the word "oral" or "spotlight" please see: "oral" presentation means -> notable-top-5% and "spotlight" means -> notable-top-25%. As stated in our emails, we are disassociating presentation type from AC recommendations